# On the Exploration of Local Significant Differences For Two-Sample Test

**Zhi-Jian Zhou,  Jie Ni,  Jia-He Yao,  Wei Gao**
National Key Laboratory for Novel Software Technology, Nanjing University, China
School of Artificial Intelligence, Nanjing University, China
{zhouzj, nij, yaojh, gaow}@lamda.nju.edu.cn

## Abstract

Recent years have witnessed increasing attentions on two-sample test with diverse real applications, while this work takes one more step on the exploration of local significant differences for two-sample test. We propose the $\text{ME}_{\text{MaBiD}}$, an effective test for two-sample testing, and the basic idea is to exploit local information by multiple Mahalanobis kernels and introduce bi-directional hypothesis for testing. On the exploration of local significant differences, we first partition the embedding space into several rectangle regions via a new splitting criterion, which is relevant to test power and data correlation. We then explore local significant differences based on our bi-directional masked $p$-value together with the $\text{ME}_{\text{MaBiD}}$ test. Theoretically, we present the asymptotic distribution and lower bounds of test power for our $\text{ME}_{\text{MaBiD}}$ test, and control the familywise error rate on the exploration of local significant differences. We finally conduct extensive experiments to validate the effectiveness of our proposed methods on two-sample test and the exploration of local significant differences.

## 1  Introduction

Two-sample test has attracted much attention with diverse applications such as cancer detection [1], distribution-shift detection [2], generative modeling [3, 4], etc. The basic problem is to assess whether two i.i.d. samples are drawn from the same distribution. Various kernel-based methods have been developed for two-sample test such as Maximum Mean Discrepancy (MMD) [5–8] and Mean Embedding (ME) [9–11]. Another relevant approach is to construct a binary classifier and assess two samples according to classification performance [12–20]. For an overview of two-sample test, we refer to a survey [21,  and references therein].

In many real applications, however, it is necessary to take one more step to explore and understand local significant differences, rather than only two-sample test. For example, a scientific problem in galaxy morphology is to identify some local regions of significant differences between two kinds of galaxies, which is important to discover galaxy formation and evolution history [22]. On the analysis of mass cytometry data in cell biology, researchers are always interested in finding local regions of significantly different abundance between disease and healthy samples [23].

Several attempts have been made to explore local significant differences in the past years. A feasible solution is to partition space into several regions, and identify significantly different regions according to their cardinalities of samples [24–27]. Another relevant work is to identify local significant differences by estimating kernel densities [28, 29] and conditional probabilities [30]. Generally, it is not easy to deal with complex data by simply counting cardinalities of samples, regardless of data intrinsic correlations, and it is also difficult to make accurate estimation of kernel densities and conditional probabilities without sufficient data, especially for many regions with finite samples.

37th Conference on Neural Information Processing Systems (NeurIPS 2023).

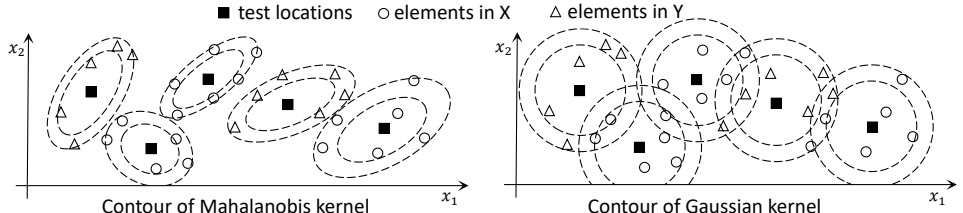

**Figure 1:** An illustration of different contours of Mahalnobis and Gaussian kernels for two-sample test.

This work presents a new two-sample test from local and directional information, and further explore local significant differences. The main contributions can be summarized as follows:

- We propose the effective $\text{ME}_{\text{MaBiD}}$ test for two-sample testing, and the basic idea is to exploit local information by multiple Mahalanobis kernels and introduce bi-directional hypothesis for testing. Intuitively, Mahalanobis kernels are more flexible to exploit local differences from neighborhoods and feature maps, and the bi-directional hypothesis is beneficial to improve the sensitivity of two-sample test with proper parameter adaptation.

- We partition the embedding space into several rectangle regions based on a new splitting criterion, which is relevant to test power and data correlation. We introduce the bi-directional masked $p$-value for each rectangle region, and finally explore local regions with significant difference based on our bi-directional masked $p$-value together with the $\text{ME}_{\text{MaBiD}}$ test.

- We present theoretical guarantees for our $\text{ME}_{\text{MaBiD}}$ test via the asymptotic distribution, as well as the lower bounds on the test power for test. We also present the upper bounds on familywise error rate for our exploration of local significant differences.

- We conduct extensive experiments to validate the effectiveness and efficiency of our methods. Specifically, our methods achieve better performance on most datasets for two-sample test and exploring local significant differences, along with comparable or smaller running time.

The rest of this work is organized as follows: Section 2 presents our $\text{ME}_{\text{MaBiD}}$ test. Section 3 explores local significant differences. Section 4 conducts extensive experiments, and Section 5 concludes with future work. All technical proofs are given in Appendix A.

## 2  Our $\text{ME}_{\text{MaBiD}}$ Test for Two-Sample Testing

Let $\mathbb{P}$ and $\mathbb{Q}$ denote two (unknown) Borel probability measures over an instance space $\mathcal{X} \subseteq \mathbb{R}^d$, and $X = \{\boldsymbol{x}_i\}_{i=1}^m$ and $Y = \{\boldsymbol{y}_j\}_{j=1}^n$ are two i.i.d. samples from $\mathbb{P}$ and $\mathbb{Q}$, respectively. The goal of two-sample test is to assess whether $X$ and $Y$ are drawn from the same distribution; in other words, we aim to assess whether $\mathbb{P} = \mathbb{Q}$ from two samples $X$ and $Y$.

We introduce some necessary notations used in this work. Write $[\tau] = \{1, 2, \cdots, \tau\}$ for integer $\tau \geq 2$, and $|A|$ denotes the cardinality of set $A$. Let $\mathbf{I}_d$ be the identity matrix of size $d \times d$. For a vector $\boldsymbol{a} = [a_1, a_2, \cdots, a_d]$, denote by $\text{sgn}(\boldsymbol{a}) = [\text{sgn}(a_1), \text{sgn}(a_2), \cdots, \text{sgn}(a_d)]$ with $\text{sgn}(a_i) = a_i/|a_i|$ for $a_i \neq 0$; otherwise, $\text{sgn}(a_i) = 0$. Let $\chi_\ell^2$ be the $\chi^2$ distribution with $\ell$ degree of freedom, as well as the $p$-value function $\chi_\ell^2(\cdot)$. Denote by $\chi_{\ell,\alpha}^2$ the $\alpha$-quantile of distribution $\chi_\ell^2$ for $\alpha \in (0, 1)$.

**Learning multiple Mahalanobis kernels via maximizing test power in training**

Following ME test [9, 10], we begin with a set of test locations $\mathcal{V} = \{\boldsymbol{v}_1, \boldsymbol{v}_2, \ldots, \boldsymbol{v}_\ell\} \subset \mathcal{X}$ to construct discriminative features. For every $\boldsymbol{v}_i \in \mathcal{V}$, we introduce a Mahalanobis kernel as follows:

$$\kappa_i(\boldsymbol{x}, \boldsymbol{v}_i) = \exp\left(-(\boldsymbol{x} - \boldsymbol{v}_i)^\top M_i (\boldsymbol{x} - \boldsymbol{v}_i)/2\gamma_i^2\right) \text{ for } \gamma_i > 0 \text{ and positive definite matrix } M_i. \quad (1)$$

Here, we propose multiple Mahalanobis kernels for two-sample test, which is motivated from multiple kernel learning [31, 32] and Mahalanobis distance [33–35]. The advantage of multiple Mahalanobis kernels is to exploit intrinsic structures and correlations from different directions and regions, and adjust geometrical distribution of data so as to enlarge the distance between different samples [36, 37].

This is different from previous Gaussian kernel $\kappa_i(\boldsymbol{x}, \boldsymbol{v}_i) = \exp(-\|\boldsymbol{x} - \boldsymbol{v}_i\|^2 / 2\gamma^2)$ [10, 38], which deals with every direction isotropically without difference. Figure 1 presents an illustration of different contours of Mahalnobis and Gaussian kernels for two-sample test. As we can see, Mahalanobis kernels are more flexible to exploit different directional information than Gaussian kernels. Our work is also different from previous deep kernel approaches [4, 39], which train single one deep neural network combined with Gaussian kernel for variations in distribution smoothness and shape.

We then embed each element in $X = \{\boldsymbol{x}_i\}_{i=1}^m$ and $Y = \{\boldsymbol{y}_j\}_{j=1}^n$ into an $\ell$-dimensional space as

$$\hat{\boldsymbol{x}}_i = (\kappa_1(\boldsymbol{x}_i, \boldsymbol{v}_1), \cdots, \kappa_\ell(\boldsymbol{x}_i, \boldsymbol{v}_\ell))^\top \ \text{ and } \ \hat{\boldsymbol{y}}_j = (\kappa_1(\boldsymbol{y}_j, \boldsymbol{v}_1), \cdots, \kappa_\ell(\boldsymbol{y}_j, \boldsymbol{v}_\ell))^\top, \text{ respectively . } \quad (2)$$

Denote by $\hat{X} = \{\hat{\boldsymbol{x}}_i\}_{i=1}^m$ and $\hat{Y} = \{\hat{\boldsymbol{y}}_j\}_{j=1}^n$. We define the *pooled covariance matrix* as

$$\Sigma_{\hat{X}, \hat{Y}} = \sum_{i=1}^m \frac{(\hat{\boldsymbol{x}}_i - \boldsymbol{c}_{\hat{X}})(\hat{\boldsymbol{x}}_i - \boldsymbol{c}_{\hat{X}})^\top}{m + n - 2} + \sum_{i=1}^n \frac{(\hat{\boldsymbol{y}}_i - \boldsymbol{c}_{\hat{Y}})(\hat{\boldsymbol{y}}_i - \boldsymbol{c}_{\hat{Y}})^\top}{m + n - 2} + \epsilon \mathbf{I}_d , \quad (3)$$

where $\boldsymbol{c}_{\hat{X}} = \sum_{i=1}^m \hat{\boldsymbol{x}}_i / m$ and $\boldsymbol{c}_{\hat{Y}} = \sum_{j=1}^n \hat{\boldsymbol{y}}_i / n$, and $\epsilon \mathbf{I}_d$ is introduced to guarantee the positive definiteness for small constant $\epsilon > 0$. We consider the *Hotelling $T^2$ statistic*, as in [40–42],

$$\mathcal{T}(\hat{X}, \hat{Y}) = mn(\boldsymbol{c}_{\hat{X}} - \boldsymbol{c}_{\hat{Y}})^\top \Sigma_{\hat{X}, \hat{Y}}^{-1} (\boldsymbol{c}_{\hat{X}} - \boldsymbol{c}_{\hat{Y}}) / (m + n) . \quad (4)$$

Test power is the probability of correctly identifying two different samples. Maximizing $\mathcal{T}(\hat{X}, \hat{Y})$ is essentially equivalent to maximizing a lower bound of test power [10, 11], and we learn test locations and Mahalanobis kernels as follows:

$$\{\mathcal{V}, M_1, \cdots, M_\ell, \gamma_1, \cdots, \gamma_\ell\} \in \arg\max\{\mathcal{T}(\hat{X}, \hat{Y})\} . \quad (5)$$

We take gradient method [43] to solve the above optimization, as done by Jitkrittum et al. [10], and the details are presented in Appendix B.

We decompose $\Sigma_{\hat{X}, \hat{Y}} = \boldsymbol{L}\boldsymbol{L}$ via the Schur method [44] to remove feature correlations, and it follows

$$\mathcal{T}(\hat{X}, \hat{Y}) = \frac{mn}{m + n}(\boldsymbol{c}_{\hat{X}} - \boldsymbol{c}_{\hat{Y}})^\top \Sigma_{\hat{X}, \hat{Y}}^{-1} (\boldsymbol{c}_{\hat{X}} - \boldsymbol{c}_{\hat{Y}}) = \frac{mn}{m + n} \left\| \boldsymbol{L}^{-1}\boldsymbol{c}_{\hat{X}} - \boldsymbol{L}^{-1}\boldsymbol{c}_{\hat{Y}} \right\|_2^2 .$$

Hence, $\mathcal{T}(\hat{X}, \hat{Y})$ essentially measures the difference between two samples via the $L_2$-norm of vector $\boldsymbol{L}^{-1}\boldsymbol{c}_{\hat{X}} - \boldsymbol{L}^{-1}\boldsymbol{c}_{\hat{Y}}$. We further exploit their *inference direction*, defined by

$$\boldsymbol{F} = \text{sgn}\left(\boldsymbol{L}^{-1}\boldsymbol{c}_{\hat{X}} - \boldsymbol{L}^{-1}\boldsymbol{c}_{\hat{Y}}\right) \in \{-1, 0, +1\}^\ell . \quad (6)$$

**Bi-directional hypothesis for testing**

Let $\hat{\mathbb{P}}$ and $\hat{\mathbb{Q}}$ be the corresponding embedding distributions from the original $\mathbb{P}$ and $\mathbb{Q}$, respectively. Denote by $\boldsymbol{\mu}_{\hat{\mathbb{P}}} = E_{\hat{\boldsymbol{x}}' \sim \hat{\mathbb{P}}}[\hat{\boldsymbol{x}}']$ and $\boldsymbol{\mu}_{\hat{\mathbb{Q}}} = E_{\hat{\boldsymbol{y}}' \sim \hat{\mathbb{Q}}}[\hat{\boldsymbol{y}}']$. We consider the following null hypothesis

$$H_0 \colon \boldsymbol{\mu}_{\hat{\mathbb{P}}} = \boldsymbol{\mu}_{\hat{\mathbb{Q}}} .$$

The null hypothesis $H_0$ can be used to test whether $\mathbb{P} = \mathbb{Q}$ by the following lemma:

**Lemma 1.** *We have $\boldsymbol{\mu}_{\hat{\mathbb{P}}} = \boldsymbol{\mu}_{\hat{\mathbb{Q}}}$ iff $\mathbb{P} = \mathbb{Q}$, for bounded Mahalanobis kernels $\{\kappa_j\}_{j=1}^\ell$ and for test locations $\{\boldsymbol{v}_j\}_{j=1}^\ell$ drawn i.i.d. from a absolutely-continuous distribution w.r.t. Lebesgue measure.*

Let $\hat{X}' = \{\hat{\boldsymbol{x}}_i'\}_{i=1}^{m'}$ and $\hat{Y}' = \{\hat{\boldsymbol{y}}_j'\}_{j=1}^{n'}$ denote two embedding testing samples. We make similar Schur decomposition $\Sigma_{\hat{X}', \hat{Y}'} = \boldsymbol{L}'\boldsymbol{L}'$, and calculate testing statistic $\mathcal{T}(\hat{X}', \hat{Y}')$ according to Eqn. (4). Based on Lemma 1, we can present the asymptotic distribution of statistic $\mathcal{T}(\hat{X}', \hat{Y}')$ as follows:

**Theorem 2.** *The testing statistic $\mathcal{T}(\hat{X}', \hat{Y}')$ is almost surely asymptotically distributed as $\chi_\ell^2$ if $\mathbb{P} = \mathbb{Q}$; otherwise, $\chi_\ell^2(\mathcal{T}(\hat{X}', \hat{Y}')) \to 0$ as $m'n'/(m' + n') \to \infty$.*

From Theorem 2, we propose the *bi-directional hypothesis*, by considering inference direction,

$$h(\hat{X}', \hat{Y}') = \begin{cases} \mathbb{I}\left[\chi_\ell^2\big(\mathcal{T}(\hat{X}', \hat{Y}')\big) \leq \beta\alpha\right] & \text{for} \quad \boldsymbol{F}^\top \boldsymbol{L}'^{-1}(\boldsymbol{c}_{\hat{X}'} - \boldsymbol{c}_{\hat{Y}'}) \geq 0 \\ \mathbb{I}\left[\chi_\ell^2\big(\mathcal{T}(\hat{X}', \hat{Y}')\big) \leq (2 - \beta)\alpha\right] & \text{for} \quad \boldsymbol{F}^\top \boldsymbol{L}'^{-1}(\boldsymbol{c}_{\hat{X}'} - \boldsymbol{c}_{\hat{Y}'}) < 0, \end{cases} \quad (7)$$

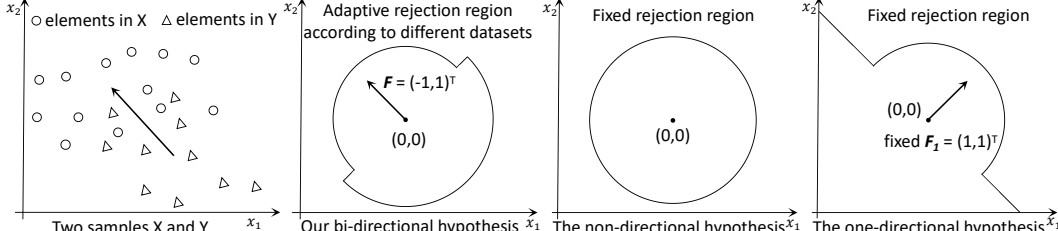

**Figure 2:** An illustration of different rejection regions on two samples for our bi-directional hypothesis and previous non/one-directional hypothesis. Our rejection region can be adaptive according to different datasets.

where $\alpha \in (0,1)$ is the significance level of hypothesis test, and $\beta \in [1,2]$ is an adaptive parameter.

Our bi-directional hypothesis is essentially about designing a rejection region of null hypothesis [45]. How to design an efficient rejection region is an interesting problem from the early work [46], and some techniques has been developed for selecting rejection regions [47–49]. We consider the most discriminative directions $\boldsymbol{F}$ and $-\boldsymbol{F}$ in our bi-directional hypothesis, which could improve the sensitivity for two-sample test by selecting appropriate parameters according to different datasets.

Our bi-directional hypothesis can be viewed as a generalization of previous hypotheses, that is,

- By setting $\beta = 1$, our test has been the non-directional hypothesis [9, 11, 50] regardless of direction information, which is also referred to as two-sided/tailed hypothesis [49, 51, 52];
- By setting $\beta = 2$ and $\boldsymbol{F} = (1, 1, \cdots, 1)^{\top}$, our test has become one-directional hypothesis [53–55], which is also referred to as one-sided/tailed hypothesis [52, 56].

Notice that previous non/one-directional hypotheses fix the structures of rejection region for a given significance level $\alpha$, whereas our bi-directional hypothesis could adjust rejection region according to inference direction $\boldsymbol{F}$ w.r.t. different datasets, which can be illustrated in Figure 2. Here, we consider an illustrative dataset, and our bi-directional hypothesis gives the rejection region adaptive to dataset, which could yield higher test power in two-sample test, as shown in Figure 5 (in Section 4).

We can also present some distribution and probability information for $\boldsymbol{F}^{\top}\boldsymbol{L}'^{-1}(\boldsymbol{c}_{\hat{X}'} - \boldsymbol{c}_{\hat{Y}'}) \geq 0$ in Eqn. (7). Denote by $\boldsymbol{L}_{\hat{\mathbb{P}},\hat{\mathbb{Q}}} = E_{\hat{X}'\sim\hat{\mathbb{P}}^{m'},\hat{Y}'\sim\hat{\mathbb{Q}}^{n'}}[\boldsymbol{L}']$ and $\xi = \Pr[\boldsymbol{F}^{\top}\boldsymbol{L}'^{-1}(\boldsymbol{c}_{\hat{X}'} - \boldsymbol{c}_{\hat{Y}'}) \geq 0]$. We have

**Lemma 3.** *For inference direction $\boldsymbol{F}$ in Eqn. (6) and for embedding samples $\hat{X}'$ and $\hat{Y}'$, we have*

$$\boldsymbol{L}'^{-1}(\boldsymbol{c}_{\hat{X}'} - \boldsymbol{c}_{\hat{Y}'}) \sim \mathcal{N}\big(\boldsymbol{L}_{\hat{\mathbb{P}},\hat{\mathbb{Q}}}^{-1}(\boldsymbol{\mu}_{\hat{\mathbb{P}}} - \boldsymbol{\mu}_{\hat{\mathbb{Q}}}), \omega^{-1}\boldsymbol{I}_{\ell}\big) \ \ and \ \ sgn(\boldsymbol{F}^{\top}\boldsymbol{L}'^{-1}(\boldsymbol{c}_{\hat{X}'} - \boldsymbol{c}_{\hat{Y}'})) \sim \mathcal{TP}(\xi)$$

*with $\omega = m'n'/(m'+n')$ and $\xi = 1 - \Phi(-\sqrt{\omega/\ell}\,\boldsymbol{F}^{\top}\boldsymbol{L}_{\hat{\mathbb{P}},\hat{\mathbb{Q}}}^{-1}(\boldsymbol{\mu}_{\hat{\mathbb{P}}} - \boldsymbol{\mu}_{\hat{\mathbb{Q}}}))$. Here, $\Phi(\cdot)$ is the distribution function of standard Gaussian, and $\mathcal{TP}(\xi)$ denotes a distribution over $\{-1, +1\}$ with probability $\xi$ on the selection of $+1$.*

The selection of parameter $\beta$ is highly positive-relevant to the probability $\xi$. This is because a larger $\xi$ implies larger difference between two samples in the inference direction $\boldsymbol{F}$, and we should select a larger $\beta$ to enlarge the rejection region and improve the sensitivity of dataset. Figure 6 (in Section 4) shows such positive relevance between the optimal parameter $\beta$ and probability $\xi$ empirically.

We finally present theoretical analysis on test power and type-I error of our bi-directional hypothesis. Let $f(x: n_1, n_2, \lambda)$ be the density function of noncentral $F$-distribution with $n_1$ and $n_2$ degrees of freedom and non-centrality parameter $\lambda$, and denote by $F_{n_1,n_2,\alpha}$ the $\alpha$-quantile of central $F$-distribution with $n_1$ and $n_2$ degrees of freedom for $\alpha \in (0,1)$. We define the following probability, from the work of [57],

$$q(n_1, n_2, \lambda, \alpha) = \int_{F_{n_1,n_2,\alpha}}^{\infty} f(x \mid n_1, n_2, \lambda)\, dx \,. \tag{8}$$

**Theorem 4.** *For our bi-directional hypothesis $h(\hat{X}', \hat{Y}')$, the test power can be lower bounded by*

$$q(\ell, \nu - \ell, \lambda, \beta\alpha) \cdot \xi + \Phi\left(-(\chi^2_{\ell,(2-\beta)\alpha})^{1/2} - (\omega/\ell)^{1/2}\boldsymbol{F}^{\top}\boldsymbol{L}_{\hat{\mathbb{P}},\hat{\mathbb{Q}}}^{-1}(\boldsymbol{\mu}_{\hat{\mathbb{P}}} - \boldsymbol{\mu}_{\hat{\mathbb{Q}}})\right)$$

---

**Algorithm 1** Construction of partition tree

---

**Input**: Two embeddding samples $\hat{X}$ and $\hat{Y}$
**Output**: Partition tree $T$
**Initialize**: Tree $T$ with only a root $[0,1]^\ell$

1: **for** $t = 1, \ldots, s - 1$ **do**
2:     Randomly select one of the leaf node $\mathcal{B}$ of the largest $|\hat{X} \cap \mathcal{B}|$
3:     Initialize the best splitting feature $j^* = 0$
4:     **for** $j = 1, \ldots, \ell$ **do**
5:         Calculate the median value in $j$-th dimension $\tau_j = \text{median}\{\hat{\boldsymbol{x}}_{i,j} : \hat{\boldsymbol{x}}_i \in \hat{X} \cap \mathcal{B}\}$
6:         Split two samples $\hat{X}_{\mathcal{B}}$ and $\hat{Y}_{\mathcal{B}}$ into $\hat{X}_{\mathcal{B}_l^j}, \hat{Y}_{\mathcal{B}_l^j}$ and $\hat{X}_{\mathcal{B}_r^j}, \hat{Y}_{\mathcal{B}_r^j}$, according to $\tau_j$
7:         Calculate $\mathcal{T}(\hat{X}_{\mathcal{B}_l^j}, \hat{Y}_{\mathcal{B}_l^j})$ and $\mathcal{T}(\hat{X}_{\mathcal{B}_r^j}, \hat{Y}_{\mathcal{B}_r^j})$ based on Eqn. (4)
8:         Update $j^* = j$ if $\mathcal{T}(\hat{X}_{\mathcal{B}_l^j}, \hat{Y}_{\mathcal{B}_l^j}) \mathcal{T}(\hat{X}_{\mathcal{B}_r^j}, \hat{Y}_{\mathcal{B}_r^j}) > \mathcal{T}(\hat{X}_{\mathcal{B}_l^{j^*}}, \hat{Y}_{\mathcal{B}_l^{j^*}}) \mathcal{T}(\hat{X}_{\mathcal{B}_r^{j^*}}, \hat{Y}_{\mathcal{B}_r^{j^*}})$
9:     **end for**
10:    Update node $\mathcal{B}$ with two children $\mathcal{B}_l^{j^*}$ and $\mathcal{B}_r^{j^*}$ w.r.t the splitting feature $j^*$ and position $\tau_{j^*}$
11: **end for**

---

if $\boldsymbol{F}^\top \boldsymbol{L}_{\hat{\mathbb{P}},\hat{\mathbb{Q}}}^{-1} \boldsymbol{\mu}_{\hat{\mathbb{P}}} > \boldsymbol{F}^\top \boldsymbol{L}_{\hat{\mathbb{P}},\hat{\mathbb{Q}}}^{-1} \boldsymbol{\mu}_{\hat{\mathbb{Q}}}$; and the test power can also be lower bounded by

$$q(\ell, \nu - \ell, \lambda, (2 - \beta)\alpha)(1 - \xi) + 1 - \Phi\left((\chi_{\ell,\beta\alpha}^2)^{1/2} - (\omega/\ell)^{1/2} \boldsymbol{F}^\top \boldsymbol{L}_{\hat{\mathbb{P}},\hat{\mathbb{Q}}}^{-1}(\boldsymbol{\mu}_{\hat{\mathbb{P}}} - \boldsymbol{\mu}_{\hat{\mathbb{Q}}})\right)$$

if $\boldsymbol{F}^\top \boldsymbol{L}_{\hat{\mathbb{P}},\hat{\mathbb{Q}}}^{-1} \boldsymbol{\mu}_{\hat{\mathbb{P}}} < \boldsymbol{F}^\top \boldsymbol{L}_{\hat{\mathbb{P}},\hat{\mathbb{Q}}}^{-1} \boldsymbol{\mu}_{\hat{\mathbb{Q}}}$; and the type-I error rate is equal to $\alpha$ if $\boldsymbol{\mu}_{\hat{\mathbb{P}}} = \boldsymbol{\mu}_{\hat{\mathbb{Q}}}$. Here, $\omega = m'n'/(m' + n')$, $\nu = m' + n' - 1$ and $\lambda = \omega\|\boldsymbol{L}_{\hat{\mathbb{P}},\hat{\mathbb{Q}}}^{-1}(\boldsymbol{\mu}_{\hat{\mathbb{P}}} - \boldsymbol{\mu}_{\hat{\mathbb{Q}}})\|_2^2$.

This theorem presents lower bounds on test power, and the performance of the statistical test is maintained under the general condition. Notice that the type-I error in our hypothesis test is controlled only by the significant level $\alpha$, regardless of different $\xi$ and $\beta$.

We call our test as $ME_{MaBiD}$ test because of multiple Mahalanobis kernels in training and our bi-directional hypothesis in testing.

## 3 Explore Local Significant Differences for Two-sample Test

On the exploration of local significant difference, most previous studies [24–28] partition the instance space into several regions, and then exploit the difference on each region from two samples. Motivated from pólya tree method [26, 58], we first partition the embedding instance space with a new splitting criterion, which is relevant to test power and data correlations. We then exploit local regions (i.e., leaves nodes of partition tree) with significant difference.

**Partition of the embedding instance space**

Our partition tree is constructed iteratively as follows: We initiate the tree root with embedding space $(0,1]^\ell$. In each iteration, each node is associated with a rectangle region, and all leaves constitute a partition of embedding instance space. The following procedure is repeated $s - 1$ iterations ($s \geq 2$):

- Randomly select a leaf node, denote by $\mathcal{B}$, uniformly over leaf nodes of the largest $|\hat{X} \cap \mathcal{B}|$.
- Let $\tau_j = \text{median}\{\hat{\boldsymbol{x}}_{i,j} : \hat{\boldsymbol{x}}_i \in \hat{X} \cap \mathcal{B}\}$ for $j \in [\ell]$. We select the best splitting feature

$$j^* \in \arg\max_{j \in [\ell]} \left\{\mathcal{T}(\hat{X}_{\mathcal{B}_l^j}, \hat{Y}_{\mathcal{B}_l^j}) \times \mathcal{T}(\hat{X}_{\mathcal{B}_r^j}, \hat{Y}_{\mathcal{B}_r^j})\right\},$$

with $\hat{X}_{\mathcal{B}_l^j} = \mathcal{B}_l^j \cap \hat{X}$, $\hat{X}_{\mathcal{B}_r^j} = \mathcal{B}_r^j \cap \hat{X}$, $\hat{Y}_{\mathcal{B}_l^j} = \mathcal{B}_l^j \cap \hat{Y}$ and $\hat{Y}_{\mathcal{B}_r^j} = \mathcal{B}_r^j \cap \hat{Y}$. Here, $\mathcal{B}_l^j$ and $\mathcal{B}_r^j$ are left and right children of $\mathcal{B}$ w.r.t. the $j$-th splitting feature and splitting position $\tau_j$, respectively, and $\mathcal{T}(\cdot, \cdot)$ is defined by Eqn. (4).

- Select the splitting position $\tau_{j^*} = \text{median}\{\hat{\boldsymbol{x}}_{i,j^*} : \hat{\boldsymbol{x}}_i \in \hat{X} \cap \mathcal{B}\}$.

**Table 1:** Datasets

| Dataset | # Inst. | # Feat. | Dataset | # Inst. | # Feat. | Dataset | # Inst. | # Feat. | Dataset | # Inst. | # Feat. |
|---------|---------|---------|---------|---------|---------|---------|---------|---------|---------|---------|---------|
| dna | 3,186 | 180 | kropt | 27,705 | 6 | santan | 200,000 | 200 | adult | 1,000,000 | 14 |
| agnos | 3,468 | 970 | diamon | 53,940 | 9 | codrna | 487,867 | 8 | labor | 1,000,000 | 16 |
| topo21 | 8,885 | 266 | cifar10 | 60,000 | 3072 | blob | 1,000,000 | 2 | poker | 1,025,010 | 10 |
| har | 10,299 | 561 | mnist | 70,000 | 784 | sea50 | 1,000,000 | 3 | higgs | 11,000,000 | 4 |

We finally get the partition tree with $s$ leaf nodes, associated with $s$ rectangle regions $\mathcal{B}_1, \mathcal{B}_2, \ldots, \mathcal{B}_s$. Algorithm 1 presents the detailed description on tree construction and rectangle region splitting.

We take the statistic $\mathcal{T}(\cdot, \cdot)$ as a splitting criterion, relevant to test power and data correlations, and it is helpful to exploit local significant difference directly. We also adopt the median splitting position with equal probabilities on partitioned regions, i.e., balanced examples for each partition region, and this could yield better performance than regular grids, as shown empirically in [59, 60].

Our partition tree is different from previous $p$-value histogram based on Chi-square test [61, 62], where the difference is measured by cardinalities of elements in two samples over a local rectangle region. Our splitting criterion is also different from that of previous decision trees [63–66], which consider some information-theoretic criterions such as entropy, Gini index, information gain, etc. In comparisons, our statistic $\mathcal{T}(\cdot, \cdot)$ is more essential to reflect the test power for two-sample test.

For each rectangle region $\mathcal{B}_i$, let $\boldsymbol{c}_{\hat{X}_{\mathcal{B}_i}}$ and $\boldsymbol{c}_{\hat{Y}_{\mathcal{B}_i}}$ be the means of $\hat{X}_{\mathcal{B}_i} = \mathcal{B}_i \cap \hat{X}$ and $\hat{Y}_{\mathcal{B}_i} = \mathcal{B}_i \cap \hat{Y}$, respectively. We make similar Schur decomposition $\boldsymbol{L}_{\mathcal{B}_i} \boldsymbol{L}_{\mathcal{B}_i} = \Sigma_{\hat{X}_{\mathcal{B}_i}, \hat{Y}_{\mathcal{B}_i}}$ for covariance matrix $\Sigma_{\hat{X}_{\mathcal{B}_i}, \hat{Y}_{\mathcal{B}_i}}$, and introduce the local inference direction for each $\mathcal{B}_i$ as follow:

$$\boldsymbol{F}_{\mathcal{B}_i} = \mathrm{sgn}\left(\boldsymbol{L}_{\mathcal{B}_i}^{-1} \boldsymbol{c}_{\hat{X}_{\mathcal{B}_i}} - \boldsymbol{L}_{\mathcal{B}_i}^{-1} \boldsymbol{c}_{\hat{Y}_{\mathcal{B}_i}}\right) \in \{-1, 0, +1\}^\ell . \tag{9}$$

For different rectangle regions, we could have different or even contrary inference directions, which is helpful to exploit local differences from distributional shapes of two samples.

**Exploration of local significant differences**

For testing embedding samples $\hat{X}'$ and $\hat{Y}'$, we denote by $\hat{X}'_{\mathcal{B}_i} = \mathcal{B}_i \cap \hat{X}'$ and $\hat{Y}'_{\mathcal{B}_i} = \mathcal{B}_i \cap \hat{Y}'$ with their respective means $\boldsymbol{c}_{\hat{X}'_{\mathcal{B}_i}}$ and $\boldsymbol{c}_{\hat{Y}'_{\mathcal{B}_i}}$ for each rectangle region $\mathcal{B}_i$, and calculate testing statistic $\mathcal{T}_{\mathcal{B}_i} = \mathcal{T}(\hat{X}'_{\mathcal{B}_i}, \hat{Y}'_{\mathcal{B}_i})$ by Eqn. (4).

We propose the new *bi-directional masked p-value* for each rectangle region $\mathcal{B}_i$ as follows:

$$g(\hat{X}'_{\mathcal{B}_i}, \hat{Y}'_{\mathcal{B}_i}) = \begin{cases} \min\left\{\frac{\chi_\ell^2(\mathcal{T}_{\mathcal{B}_i})}{\beta}, \frac{p_*\left(1 - \chi_\ell^2(\mathcal{T}_{\mathcal{B}_i})\right)}{1 - \beta p_*}\right\} & \text{for } \boldsymbol{F}_{\mathcal{B}_i}^\top \boldsymbol{L}_{\mathcal{B}_i}'^{-1}(\boldsymbol{c}_{\hat{X}'_{\mathcal{B}_i}} - \boldsymbol{c}_{\hat{Y}'_{\mathcal{B}_i}}) \geq 0 \\ \min\left\{\frac{\chi_\ell^2(\mathcal{T}_{\mathcal{B}_i})}{2 - \beta}, \frac{p_*\left(1 - \chi_\ell^2(\mathcal{T}_{\mathcal{B}_i})\right)}{1 - (2 - \beta)p_*}\right\} & \text{for } \boldsymbol{F}_{\mathcal{B}_i}^\top \boldsymbol{L}_{\mathcal{B}_i}'^{-1}(\boldsymbol{c}_{\hat{X}'_{\mathcal{B}_i}} - \boldsymbol{c}_{\hat{Y}'_{\mathcal{B}_i}}) < 0 , \end{cases} \tag{10}$$

where $\boldsymbol{L}_{\mathcal{B}_i}'^{-1}$ is from Schur decomposition $\Sigma_{\hat{X}'_{\mathcal{B}_i}, \hat{Y}'_{\mathcal{B}_i}} = \boldsymbol{L}_{\mathcal{B}_i}'^{-1} \boldsymbol{L}_{\mathcal{B}_i}'^{-1}$, $p_* \in (0, 1)$ is a parameter on significance level, and $\beta$ is an adaptive parameter. Here, we also consider two discriminative directions $\boldsymbol{F}_{\mathcal{B}_i}$ and $-\boldsymbol{F}_{\mathcal{B}_i}$ on each rectangle region $\mathcal{B}_i$, which is different from previous masked $p$-value [67] without directional information.

Our bi-directional masked $p$-value directly reflects the significant level of local difference when there is a significant difference in local $\mathcal{B}_i$, similarly to [67]. The smaller the bi-directional masked $p$-value, the more significant the local difference. On the other hand, the bi-directional masked $p$-value is a random number with uniform distribution over $(0, p_*)$ when there is no significant difference in $\mathcal{B}_i$, since $\chi_\ell^2(\mathcal{T}_{\mathcal{B}_i})$ follows a uniform distribution in such case [68].

Based on such recognition, we resort rectangle regions as $\mathcal{B}_{\langle 1 \rangle}, \mathcal{B}_{\langle 2 \rangle}, \ldots, \mathcal{B}_{\langle s \rangle}$ according to their bi-directional masked $p$-value, i.e.,

$$g(\hat{X}'_{\mathcal{B}_{\langle 1 \rangle}}, \hat{Y}'_{\mathcal{B}_{\langle 1 \rangle}}) \leq g(\hat{X}'_{\mathcal{B}_{\langle 2 \rangle}}, \hat{Y}'_{\mathcal{B}_{\langle 2 \rangle}}) \leq \cdots \leq g(\hat{X}'_{\mathcal{B}_{\langle s \rangle}}, \hat{Y}'_{\mathcal{B}_{\langle s \rangle}}) .$$

**Table 2:** Comparisons of test powers (mean±std) on two-sample test. Bold denotes the highest mean in per row.

| Dataset | Our ME$_{\text{MaBiD}}$ | ME | MMDAgg | MMD-D | C2ST-L | C2ST-S | AutoMLTST |
|---|---|---|---|---|---|---|---|
| blob | **.985**±**.009** | .823±.000 | .935±.012 | .963±.010 | .972±.078 | .946±.037 | .980±.029 |
| dna | **.717**±**.068** | .536±.059 | .659±.070 | .628±.006 | .699±.028 | .505±.044 | .603±.085 |
| agnos | **.812**±**.018** | .602±.033 | .779±.046 | .734±.006 | .742±.012 | .679±.051 | .632±.077 |
| topo21 | **.692**±**.006** | .526±.058 | .605±.077 | .633±.062 | .679±.046 | .517±.046 | .591±.006 |
| har | **.858**±**.065** | .816±.015 | .814±.026 | .728±.064 | .761±.093 | .738±.063 | .740±.058 |
| kropt | **.992**±**.012** | .875±.027 | .971±.024 | .916±.066 | .946±.013 | .929±.031 | .971±.026 |
| diamon | **.837**±**.066** | .697±.068 | .676±.047 | .755±.056 | .747±.086 | .727±.076 | .831±.062 |
| cifar | **.893**±**.022** | .859±.075 | .866±.091 | .878±.090 | .834±.099 | .798±.019 | .882±.086 |
| mnist | **.985**±**.017** | .926±.056 | .932±.068 | .972±.051 | .969±.042 | .930±.029 | .963±.074 |
| santan | **1.00**±**.000** | .896±.060 | **1.00**±**.000** | .887±.021 | .911±.084 | .850±.021 | .954±.012 |
| codrna | **1.00**±**.000** | .946±.085 | .926±.037 | .914±.076 | **1.00**±**.000** | **1.00**±**.000** | .876±.067 |
| sea50 | **.993**±**.018** | **.993**±**.018** | .982±.012 | **.993**±**.018** | **.993**±**.018** | .970±.053 | .989±.029 |
| adult | **.996**±**.002** | .875±.034 | .967±.029 | .908±.072 | .761±.091 | .854±.058 | .992±.006 |
| labor | .992±.012 | .807±.078 | .988±.010 | .930±.093 | .756±.059 | .791±.031 | **1.00**±**.000** |
| poker | .821±.079 | .719±.096 | .712±.033 | .701±.056 | .743±.039 | .731±.052 | **.832**±**.048** |
| higgs | **.979**±**.024** | .818±.090 | .938±.047 | .953±.055 | .968±.043 | .933±.013 | .969±.030 |
| Average | **.909**±.026 | .795±.053 | .859±.039 | .843±.050 | .842±.052 | .806±.039 | .863±.043 |

We then take our bi-directional hypothesis $h(\hat{X}'_{\mathcal{B}_i}, \hat{Y}'_{\mathcal{B}_i})$ with parameter $\beta$ and $\alpha = p_*$ as in Eqn. (7), and finally get the local regions with significant differences as

$$\left\{ \mathcal{B}_{\langle i \rangle}: \quad i \leq t^* \text{ and } h(\hat{X}'_{\mathcal{B}_{\langle i \rangle}}, \hat{Y}'_{\mathcal{B}_{\langle i \rangle}}) = 1 \right\} ,$$

where

$$t^* = \underset{t \in [s]}{\arg\max} \left\{ t - \left| \{ i \in [t]: h(\hat{X}'_{\mathcal{B}_{\langle i \rangle}}, \hat{Y}'_{\mathcal{B}_{\langle i \rangle}}) = 1 \} \right| + 1 \leq \frac{\ln(1 - \alpha_*)}{\ln(1 - p_*)} \right\} . \quad (11)$$

Here, $p_*$ is selected as in Eqn. (10), and $\alpha_* \in [p_*, 1)$ is a parameter to control the probability of mis-identifying at least one rectangle region without significant difference, also called *familywise error rate* [69–72]. We present theoretical analysis for familywise error rate as follows:

**Theorem 5.** *For our exploration, the familywise error rate is upper bounded by $\alpha_*$, if 1) the $p$-values of local regions without differences are mutually independent; and 2) the $p$-values of local regions with differences are independent to those $p$-values of local regions without differences.*

Our method is different from previous space partition methods of trees or clusters [24–27, 60], where the splitting criterion is taken as the cardinalities of samples in each region. Duong [28] partitioned and searched local regions from the estimated density function, and Kim et al. [30] identified local regions by clustering data samples from the estimated conditional probabilities. It is not easy to make accurate estimation for density and conditional probabilities without sufficient data, particularly for multiple small regions. Other relevant studies detected local differences implicitly based on interactive rank test [73] or a learned classifier [74].

## 4 Experiments

We conduct experiments on 16 datasets[1] as summarized in Table 1. Most dataset have been studied in previous two-sample test, and features have been scaled to $[0, 1]$ for all datasets. All experiments are performed with Python on nodes of a computational cluster with a single CPU (Intel Core i9-10900X 3.7GHz) and a single GPU (GeForce RTX 2080 Ti), running Ubuntu with 128GB main memory.

**Experimental comparisons for two-sample test**

We compare our ME$_{\text{MaBiD}}$ with the state-of-the-art approaches on two-sample test as follows:

---

[1]Dataset blob is downloaded from *github.com/fengliu90/DK-for-TST*,
and other datasets are downloaded from *www.openml.org*.

**Table 3:** Comparisons of density differences (mean±std) on the exploration of local significant differences, and the bold denotes the highest mean in per row.

| Dataset | Our method | FDG | KPRIM | MRS | MMDT | BTLDD | TEAM |
|---------|-----------|-----|-------|-----|------|-------|------|
| blob | **.945**±**.082** | .902±.075 | .879±.045 | .849±.075 | .909±.091 | .932±.046 | .877±.067 |
| diamon | **.974**±**.054** | .852±.010 | .895±.089 | .876±.066 | .867±.104 | .951±.070 | .947±.022 |
| codrna | **.969**±**.026** | .936±.037 | .876±.061 | .966±.045 | .884±.105 | .905±.050 | .863±.036 |
| sea50 | .977±.056 | .975±.062 | .933±.074 | .928±.058 | **.985**±**.045** | .892±.059 | .944±.065 |
| adult | **.953**±**.048** | .862±.044 | .838±.109 | .911±.085 | .927±.101 | .875±.074 | .880±.070 |
| labor | **.959**±**.062** | .894±.080 | .905±.016 | .900±.037 | .911±.093 | .922±.048 | .932±.067 |
| poker | **.945**±**.030** | .901±.030 | .882±.027 | .925±.023 | .927±.057 | .894±.028 | .884±.064 |
| higgs | **.946**±**.016** | .932±.011 | .918±.001 | .940±.000 | .927±.026 | .926±.027 | .937±.002 |
| Average | **.959**±.047 | .907±.044 | .891±.053 | .912±.049 | .917±.078 | .912±.050 | .908±.049 |

- ME: Mean Embeddings over multiple test locations and a single Gaussian kernel [9, 10];
- MMD-D: Maximum Mean Discrepancy based on a Deep kernel [39];
- MMDAgg: Maximum Mean Discrepancy with Aggregating of multiple Gaussian kernels [75];
- C2ST-S: Train a binary classification network and test its accuracy on a hold-out set [13];
- C2ST-L: Train a binary classification network with a statistic about class probabilities [14, 18];
- AutoMLTST: Train a binary classifier based on AutoML method with a statistic as C2ST-L [19].

Following [39, 76], we train on a subset of each available data, and test on 100 random subsets from the remaining dataset, and the ratio is set as $4:1$ for training and testing. We repeat such process 10 times for each dataset. More details are given in Appendix C. For our $ME_{MaBiD}$, we set $\alpha = 0.05$ and take 5-fold cross validation to select $\beta \in [1:0.2:2]$. We limit the cardinality of test locations within 20 for ME and $ME_{MaBiD}$ as in [9–11], and optimization parameters of Eqn. (5) is presented in Appendix C. We take parameter settings for other methods as in their respective inferences.

Table 2 summarizes the average of test powers and standard deviations. It is evident that our $ME_{MaBiD}$ takes better performance than ME and MMDAgg, because they both take Gaussian kernels with isotropic scale, and ignore the distributional differences from different directions. Our method is still better than MMD-D with a deep kernel, and a reason is that multiple Mahalanobis kernels are more flexible than a deep kernel to capture local difference from multiple neighborhoods and directions.

From Table 2, it is also observed that our $ME_{MaBiD}$ outperforms three classifier-based methods C2ST-S, C2ST-L and AutoML expect for datasets labor and poker, since those methods focus merely on the prediction information from outputs of classifiers, rather than local and directional information among data samples. For datasets labor and poker, AutoML generates new features automatically from the original mixture of continuous and symbolic features, and thus achieves better performance.

We further compare the average running time (in seconds) for different methods on two-sample test, as shown in Figure 3. As expected, ME takes the least running time since it considers only one Gaussian kernel, yet with the smallest average of test powers in Table 2. Our $ME_{MaBiD}$ method takes smaller and comparable running time in contrast to other methods since our method takes relatively smaller time on training Mahalanobis kernels without permutation test in the testing process.

**Experiments on the exploration of local significant differences**

We compare with the state-of-the-art approaches on exploring local significant difference as follows:

- FDG: Partition space by probability binning and compare cardinalities of two samples [24];
- K-PRIM: Partition space by patient rule induction and estimate kernel density differences [28];
- MRS: Partition space by pólya tree and measure difference via Binomial distributions [26];
- TEAM: Partition space by data variance and measure difference via Binomial distributions [27];
- BTLDD: Estimate conditional probabilities of two samples and cluster data via difference [77];
- MMDT: Partition space into equal grids and test density difference via Welch's statistic [29].

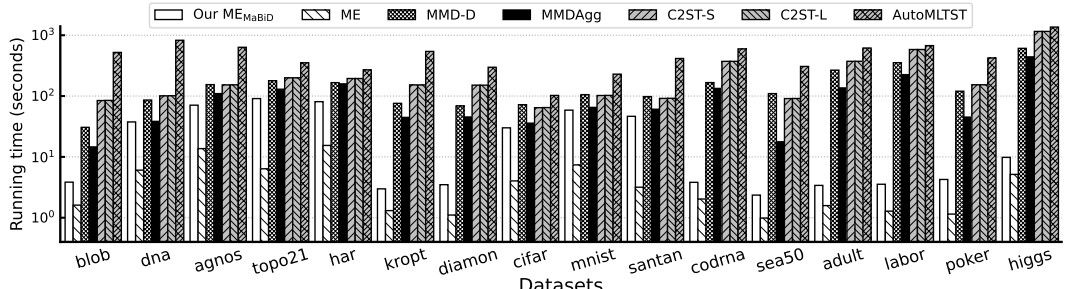

**Figure 3:** Comparisons of running time for different methods on two-sample test. Note that y-axis is in log-scale.

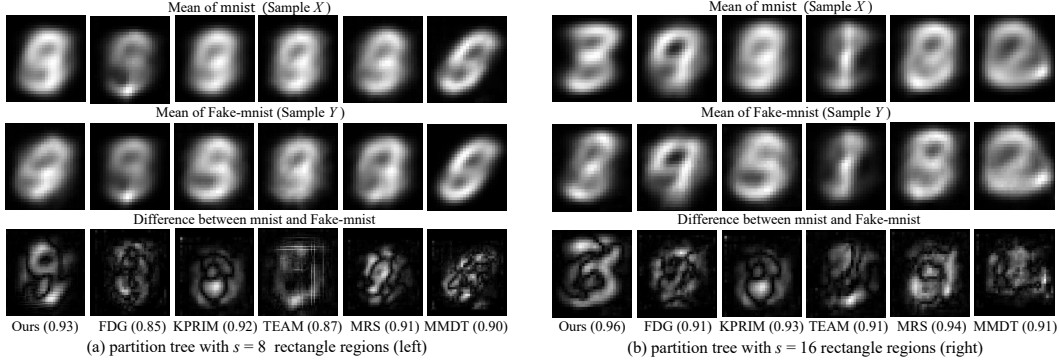

(a) partition tree with $s = 8$ rectangle regions (left)     (b) partition tree with $s = 16$ rectangle regions (right)

**Figure 4:** Visualization of the local significant differences with partition tree of $s = 8$ (left) and $s = 16$ (right) rectangle regions. The bigger the values, the larger the density difference.

For fair comparisons with some methods of density estimations, we select eight large datasets: diamond, codrna, blob, sea50, adult, labor, poker and higgs, whose instance numbers are more than 50,000 and feature numbers are smaller than 20. We randomly select 10,000 instances during the exploration of local significant differences. For the BTLDD method, we set the percentage of selected samples as 6.25% for local regions in a cluster, while for other methods, we partition the space into 64 rectangle regions, and identify 4 rectangle regions with the most local differences.

We take density differences [78] between two samples in a local region as an evaluation measure for local significant differences, and follow the works of [79, 80] based on $k$-NN density estimator with $k = 20$. We calculate the difference between two estimated density functions in an identified local region, and normalize the returned values into $[0, 1]$.

Table 3 summarizes the average of density differences and standard deviations. As can be seen, our method takes better explorations on local significant difference than FDG, MRS and TEAM, since those methods simply take cardinalities of samples to measure significant difference. Our exploration is still better than K-PRIM, BTLDD and MMDT except for dataset sea50, because those methods estimate density or conditional probabilities from two samples, and it is not easy to make accurate estimation without sufficient data, especially for small regions with finite samples. MMDT could estimate density function well on dataset sea50, and achieves better exploration via Welch's statistic.

We further visualize the local significant differences on mnist (sample $X$) and Fake-mnist (sample $Y$) in Figure 4. Here, we take partition trees with $s = 8$ (left) and $s = 16$ (right) rectangle regions. As can be seen, our method achieves the largest local differences via new splitting criterion for partition tree, and our proposed ME$_{\text{MaBiD}}$ test and bi-directional masked $p$-values.

## Parameter analysis

We now present some parameter influence for our ME$_{\text{MaBiD}}$ test and the exploration of local significant difference. We only present the results on four datasets due to pages limit, but the trends are similar on other datasets, and more results can be found in Appendix D.

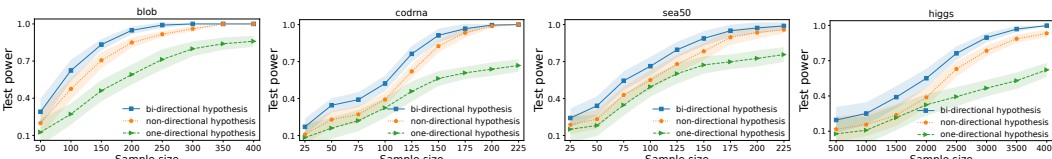

**Figure 5:** The comparisons of test power vs sample size for our bi-directional hypothesis and previous one/non-directional hypothesis.

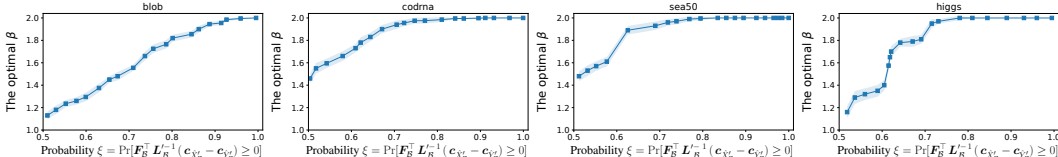

**Figure 6:** The relationship between the optimal adaptive parameter $\beta$ and probability $\xi$ for our $\text{ME}_{\text{MaBiD}}$.

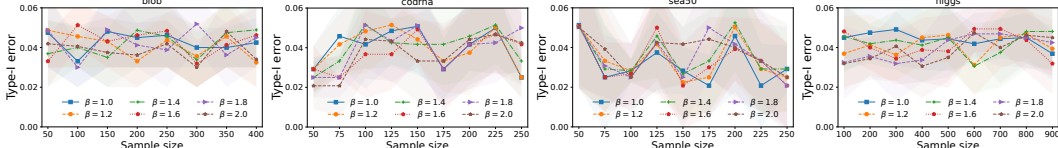

**Figure 7:** The type-I error is limited about $\alpha = 0.05$ w.r.t different $\beta$ for our $\text{ME}_{\text{MaBiD}}$.

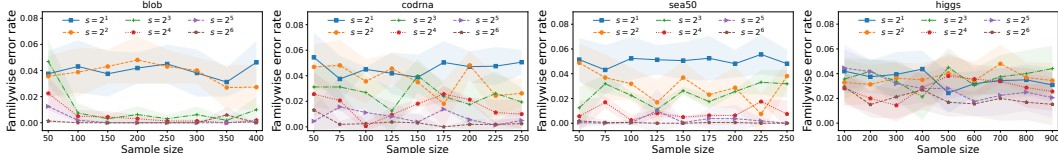

**Figure 8:** The FWER is limited about $\alpha_* = 0.05$ w.r.t. different $s$ for exploring local significant difference.

Figure 5 illustrates the test power versus sample size for our bi-directional hypothesis, non-directional hypothesis and one-directional hypothesis. As can be seen, our bi-directional hypothesis achieves higher test power by considering the inference and its contrary direction and adaptive parameter selection. Figure 6 exploits the relationship between the optimal parameter $\beta$ and the probability $\xi = \Pr[\boldsymbol{F}^\top \boldsymbol{L}'^{-1}(\boldsymbol{c}_{\hat{X}'} - \boldsymbol{c}_{\hat{Y}'}) \geq 0]$ for our $\text{ME}_{\text{MaBiD}}$ method. We can easily find the positive relevance between $\beta$ and $\xi$: the larger the probability $\xi$, the larger the optimal parameter $\beta$.

Figure 7 indicates that the type-I error is limited about $\alpha = 0.05$ for different $\beta$ in our experiments, as shown in Theorem 4, and thus our method could effectively control the rate of falsely reject the null hypothesis, which empirically verify the trustworthiness of our $\text{ME}_{\text{MaBiD}}$ test. Figure 8 empirically shows the familywise error rate is limited about $\alpha_* = 0.05$ for different number of local regions $s$; therefore, our exploring method could control the rate of incorrectly exploiting the local regions with significant difference, and this is nicely in accordance with Theorem 5.

## 5 Conclusion

This work takes one more step on the exploration of local significant differences. We propose the $\text{ME}_{\text{MaBiD}}$ test by exploiting local information from multiple Mahalanobis kernels and introducing bi-directional hypothesis for testing. We partition embedding space via a new splitting criterion, and then identify local significant differences based on our bi-directional masked $p$-value and $\text{ME}_{\text{MaBiD}}$ test. We verify the effectiveness of our proposed methods both theoretically and empirically. An interesting work is to explore other local and directional information for local significant differences.

## Acknowledgments and Disclosure of Funding

The authors want to thank the reviewers for their helpful comments and suggestions. This research was supported by National Key R&D Program of China (2021ZD0112802), NSFC (61921006, 62376119) and CAAI-Huawei MindSpore Open Fund. W. Gao is corresponding author of this paper.

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

# A  Detailed Proofs for Our Theoretical Results

## A.1  Proof of Lemma 1

We begin with some useful definitions and lemmas as follows:

**Definition 6.** (Random Metric [9]). We say that $\rho$ is a random metric with values in $\mathbb{R}$, indexed with pairs from the set of probability measures $\mathcal{M}$, i.e., $\rho = \{\rho(\mathbb{P}, \mathbb{Q}) \colon \mathbb{P}, \mathbb{Q} \in \mathcal{M}\}$, if it satisfies the conditions for a metric with qualification 'almost surely'. Formally, for every $\mathbb{P}, \mathbb{Q}, \mathbb{U} \in \mathcal{M}$, random variables $\rho(\mathbb{P}, \mathbb{Q})$, $\rho(\mathbb{P}, \mathbb{U})$ and $\rho(\mathbb{Q}, \mathbb{U})$ satisfy

1. $\rho(\mathbb{P}, \mathbb{Q}) \geq 0$ a.s.

2. if $\mathbb{P} = \mathbb{Q}$, then $\rho(\mathbb{P}, \mathbb{Q}) = 0$ a.s, if $\mathbb{P} \neq \mathbb{Q}$ then $\rho(\mathbb{P}, \mathbb{Q}) \neq 0$ a.s.

3. $\rho(\mathbb{P}, \mathbb{Q}) = \rho(\mathbb{Q}, \mathbb{P})$ a.s.

4. $\rho(\mathbb{P}, \mathbb{Q}) \leq \rho(\mathbb{P}, \mathbb{U}) + \rho(\mathbb{U}, \mathbb{Q})$ a.s.

**Lemma 7.** *[11] Let $\mu$ be a absolutely continuous Lebesgue measure on $\mathbb{R}^d$. A non-zero analytic function $f$ can be zero at most in the set of measure $0$ w.r.t. $\mu$.*

**Lemma 8.** *[9] If $\kappa$ is a bounded and analytic kernel on $\mathbb{R}^d \times \mathbb{R}^d$, then it is analytic for every function in the RKHS associated with this kernel.*

**Definition 9.** For test locations $\mathcal{V} = \{\boldsymbol{v}_1, \boldsymbol{v}_2, \cdots, \boldsymbol{v}_\ell\}$ and Mahalanobis kernels $\kappa_1, \kappa_2, \cdots, \kappa_\ell$ given by Eqn. (1), we define

$$\rho_{\mu_1,\mu_2,\cdots,\mu_\ell}(\mathbb{P}, \mathbb{Q}) = \sum_{j=1}^{\ell} \left( [\mu_j \mathbb{P}](\boldsymbol{v}_j) - [\mu_j \mathbb{Q}](\boldsymbol{v}_j) \right)^2, \tag{12}$$

where $[\mu_j \mathbb{P}](\boldsymbol{v}_j) = E_{\boldsymbol{x} \sim \mathbb{P}}[\kappa_j(\boldsymbol{x}, \boldsymbol{v}_j)]$ and $[\mu_j \mathbb{Q}](\boldsymbol{v}_j) = E_{\boldsymbol{y} \sim \mathbb{Q}}[\kappa_j(\boldsymbol{y}, \boldsymbol{v}_j)]$.

We show that the above distance metric for two probability measures is a random metric as follows:

**Lemma 10.** *If $\boldsymbol{v}_1, \boldsymbol{v}_2, \cdots, \boldsymbol{v}_\ell$ are drawn i.i.d. from a absolutely continuous distribution $\mathcal{G}$, then $\rho_{\mu_1,\mu_2,\cdots,\mu_\ell}(\cdot, \cdot)$ is a random metric for bounded kernels $\{\kappa_1, \kappa_2, \cdots, \kappa_\ell\}$.*

*Proof.* For $j \in [\ell]$, we first introduce a function

$$\rho_{\mu_j}(\mathbb{P}, \mathbb{Q}) = \left( [\mu_j \mathbb{P}](\boldsymbol{v}_j) - [\mu_j \mathbb{Q}](\boldsymbol{v}_j) \right)^2,$$

and it is sufficient to prove that $\rho_{\mu_j}(\mathbb{P}, \mathbb{Q})$ is a random metric for each $j \in [\ell]$ from Eqn. (12).

It is well-known that Mahalanobis kernels in Eqn. (1) are characteristic and analytic from [81], and the corresponding mapping

$$\mu_j \colon \mathbb{P} \to \mu_j \mathbb{P} \quad \text{and} \quad \mu_j \colon \mathbb{Q} \to \mu_j \mathbb{Q}$$

are injective for $\kappa_j$, where $\mu_j \mathbb{P}$ and $\mu_j \mathbb{Q}$ denote the images of measures $\mathbb{P}$ and $\mathbb{Q}$, respectively. Hence, the image of $\mu_j$ is a subset of analytic functions for analytic and bounded $\kappa_j$, according to Lemma 8. If $\mathbb{P} = \mathbb{Q}$, then we have

$$f_j = \mu_j \mathbb{P} - \mu_j \mathbb{Q} = 0 \quad \text{and} \quad \rho_{\mu_j}(\mathbb{P}, \mathbb{Q}) = ([\mu_j \mathbb{P}](\boldsymbol{v}_j) - [\mu_j \mathbb{Q}](\boldsymbol{v}_j))^2 = 0.$$

We now prove that if $\mathbb{P} \neq \mathbb{Q}$ then $f_j \neq 0$ almost surely, by applying Lemma 7 to analytic function $f_j = \mu_j \mathbb{P} - \mu_j \mathbb{Q}$ with distribution $\mathcal{G}$. For injective map $\mu_j$, there exists at least one point $a$ such that $f_j(a) \neq 0$, and there exists a ball around $a$ with non-zero $f_j$ from the continuity of $f_j$. Hence, $f$ is almost everywhere nonzero based on Lemma 7, and this follows that

$$\rho_{\mu_j}(\mathbb{P}, \mathbb{Q}) = ([\mu_j \mathbb{P}](\boldsymbol{v}_j) - [\mu_j \mathbb{Q}](\boldsymbol{v}_j))^2 > 0 \quad \text{a.s.} \quad \text{for} \quad \mathbb{P} \neq \mathbb{Q}.$$

Hence, $\rho_{\mu_j}$ is random metric from Definition 6 from the symmetry and triangle inequality of $\rho_{\mu_j}$. $\square$

**Proof of Lemma 1.** We first have

$$\begin{aligned}
\boldsymbol{\mu}_{\hat{\mathbb{P}}} &= E_{\hat{\boldsymbol{x}} \sim \mathbb{P}}[\hat{\boldsymbol{x}}] = ([\mu_1 \mathbb{P}](\boldsymbol{v}_1), [\mu_2 \mathbb{P}](\boldsymbol{v}_2), \cdots, [\mu_\ell \mathbb{P}](\boldsymbol{v}_\ell)), \\
\boldsymbol{\mu}_{\hat{\mathbb{Q}}} &= E_{\hat{\boldsymbol{y}} \sim \mathbb{Q}}[\hat{\boldsymbol{y}}] = ([\mu_1 \mathbb{Q}](\boldsymbol{v}_1), [\mu_2 \mathbb{Q}](\boldsymbol{v}_2), \cdots, [\mu_\ell \mathbb{Q}](\boldsymbol{v}_\ell)),
\end{aligned}$$

and further rewrite Eqn. (12) as

$$\rho_{\mu_1,\mu_2,\cdots,\mu_\ell}(\mathbb{P},\mathbb{Q}) = \sum_{j=1}^{\ell}\left([\mu_j\mathbb{P}]\,(\boldsymbol{v}_j) - [\mu_j\mathbb{Q}]\,(\boldsymbol{v}_j)\right)^2 = \sum_{j=1}^{\ell}\left(\boldsymbol{\mu}_{\hat{\mathbb{P}},j} - \boldsymbol{\mu}_{\hat{\mathbb{Q}},j}\right)^2 = \left\|\boldsymbol{\mu}_{\hat{\mathbb{P}}} - \boldsymbol{\mu}_{\hat{\mathbb{Q}}}\right\|_2^2.$$

From Lemma 10, we can see that $\rho_{\mu_1,\mu_2,\cdots,\mu_\ell}(\mathbb{P},\mathbb{Q}) = 0$ if and only if $\mathbb{P} = \mathbb{Q}$, and this implies that $\boldsymbol{\mu}_{\hat{\mathbb{P}}} = \boldsymbol{\mu}_{\hat{\mathbb{Q}}}$ if and only if $\mathbb{P} = \mathbb{Q}$. This completes the proof of Lemma 1. □

## A.2 Proof of Theorem 2

We begin with a useful lemma from [11] as follows.

**Lemma 11.** *For symmetric and positive definite matrix* $\Sigma$, *function* $h(\Sigma) = \Sigma^{-1/2}$ *is continuous and well-defined on the positive definite space.*

Based on this lemma, we present the detailed proof of Theorem 2 as follows:

**Proof of Theorem 2.** For embedding testing sample $\hat{X}' = \{\hat{\boldsymbol{x}}'_1, \cdots, \hat{\boldsymbol{x}}'_{m'}\}$ and $\hat{Y}' = \{\hat{\boldsymbol{y}}'_1, \cdots, \hat{\boldsymbol{y}}'_{n'}\}$, recall the pooled covariance matrix in Eqn. (3) as

$$\Sigma_{\hat{X}',\hat{Y}'} = \frac{(m'-1)\Sigma_{\hat{X}} + (n'-1)\Sigma_{\hat{Y}}}{m'+n'-2} + \epsilon\boldsymbol{I}_d\,,$$

and we also define

$$\Sigma_{\hat{\mathbb{P}},\hat{\mathbb{Q}}} = E_{\hat{X}'\sim\hat{\mathbb{P}}^{m'},\hat{Y}'\sim\hat{\mathbb{Q}}^{n'}}\left[\Sigma_{\hat{X}',\hat{Y}'}\right]\,.$$

We first prove that testing statistic $\mathcal{T}(\hat{X}',\hat{Y}')$ is almost surely asymptotically distributed as $\chi_\ell^2$ with $\ell$ degrees of freedom under the condition $\boldsymbol{\mu}_{\hat{\mathbb{P}}} = \boldsymbol{\mu}_{\hat{\mathbb{Q}}}$ for small $\epsilon$.

For i.i.d. samples $\hat{X}'$ and $\hat{Y}'$, $\boldsymbol{c}_{\hat{\boldsymbol{x}}}$ is independent to $\boldsymbol{c}_{\hat{\boldsymbol{y}}}$. If $\boldsymbol{\mu}_{\hat{\mathbb{P}}} = \boldsymbol{\mu}_{\hat{\mathbb{Q}}}$, then $\bar{\boldsymbol{z}} = \boldsymbol{c}_{\hat{X}} - \boldsymbol{c}_{\hat{Y}}$ follows a multivariate normal distribution with mean $\boldsymbol{\mu} = \boldsymbol{\mu}_{\hat{\mathbb{P}}} - \boldsymbol{\mu}_{\hat{\mathbb{Q}}} = \boldsymbol{0}$ and covariance matrix $(m'+n')\Sigma_{\hat{\mathbb{P}},\hat{\mathbb{Q}}}/m'n'$, from [41, 82] and the Slutsky's theorem [83], that is

$$\bar{\boldsymbol{z}} \xrightarrow{d} \mathcal{N}(\boldsymbol{0}, (m'+n')\Sigma_{\hat{\mathbb{P}},\hat{\mathbb{Q}}}/m'n')\,,$$

where $\xrightarrow{d}$ denotes convergence in distribution.

The matrix $\boldsymbol{L}$ is symmetric and invertible in the Schur decomposition $\Sigma_{\hat{X}',\hat{Y}'} = \boldsymbol{L}\boldsymbol{L}$. Our statistic can be formalized as

$$\begin{aligned}
\mathcal{T}(\hat{X}',\hat{Y}') &= m'n'\,\bar{\boldsymbol{z}}^\top\Sigma_{\hat{X},\hat{Y}}^{-1}\bar{\boldsymbol{z}}/(m'+n') \\
&= m'n'\,\bar{\boldsymbol{z}}^\top\boldsymbol{L}'^{-1}\boldsymbol{L}'^{-1}\bar{\boldsymbol{z}}/(m'+n') \\
&= \left(\sqrt{m'n'}\left\|\boldsymbol{L}'^{-1}\bar{\boldsymbol{z}}\right\|_2\big/\sqrt{m'+n'}\right)^2,
\end{aligned}$$

and by applying the Slutsky's theorem, we have

$$\sqrt{m'n'}\boldsymbol{L}'^{-1}\bar{\boldsymbol{z}}\big/\sqrt{m'+n'} \xrightarrow{d} \mathcal{N}(\boldsymbol{0},\boldsymbol{I}_\ell)\,.$$

This follows that

$$\mathcal{T}(\hat{X}',\hat{Y}') = \frac{m'n'}{m'+n'}\sum_{j=1}^{\ell}(\boldsymbol{L}'^{-1}\bar{\boldsymbol{z}})_j^2\,,$$

where $(\boldsymbol{L}'^{-1}\bar{\boldsymbol{z}})_j$ is the $j$-th dimension value of $\boldsymbol{L}'^{-1}\bar{\boldsymbol{z}}$, and $\sqrt{m'n'/(m'+n')}(\boldsymbol{L}'^{-1}\bar{\boldsymbol{z}})_j$ follows the standard normal distribution. This proves the $\chi_\ell^2$ distribution for $\mathcal{T}(\hat{X}',\hat{Y}')$ from the sum of $\ell$ squares of standard (i.i.d.) normal random variables.

We then prove that $\chi_\ell^2(\mathcal{T}(\hat{X}',\hat{Y}')) \to 0$ as $m'n'/(m'+n') \to \infty$ if $\boldsymbol{\mu}_{\hat{\mathbb{P}}} \neq \boldsymbol{\mu}_{\hat{\mathbb{Q}}}$, and hence our test rejects $H_0$ almost surely.

From Lemma 11, we have

$$\lim_{m'n'/(m'+n')\to\infty} \boldsymbol{L}'^{-1} = \lim_{m'n'/(m'+n')\to\infty} \Sigma_{\hat{X}',\hat{Y}'}^{-1/2} = \Sigma_{\hat{\mathbb{P}},\hat{\mathbb{Q}}}^{-1/2} = \boldsymbol{L}_{\hat{\mathbb{P}},\hat{\mathbb{Q}}}^{-1} \ .$$

In a similar manner, $\bar{z} = \boldsymbol{c}_{\hat{X}} - \boldsymbol{c}_{\hat{Y}}$ converges to $\boldsymbol{\mu}_{\hat{\mathbb{P}}} - \boldsymbol{\mu}_{\hat{\mathbb{Q}}}$ in probability. For $\boldsymbol{\mu}_{\hat{\mathbb{P}}} \neq \boldsymbol{\mu}_{\hat{\mathbb{Q}}}$, we have

$$\left\| \Sigma_{\hat{\mathbb{P}},\hat{\mathbb{Q}}}^{-\frac{1}{2}} (\boldsymbol{\mu}_{\hat{\mathbb{P}}} - \boldsymbol{\mu}_{\hat{\mathbb{Q}}}) \right\|_2^2 > 0 \ .$$

Then, $\|\boldsymbol{L}'^{-1}\bar{z}\|_2^2$ is a continuous function with entries $\bar{z}$ and $\boldsymbol{L}'^{-1}$, and it is convergent to some positive constant. We have $(m' + n')/(m'n')\chi_{\ell,\alpha}^2 \to 0$, and

$$\Pr\left[ \frac{m'n'}{m'+n'} \left\| \boldsymbol{L}'^{-1}\bar{z} \right\|_2^2 > \chi_{\ell,\alpha}^2 \right] = P\left( \left\| \boldsymbol{L}'^{-1}\bar{z} \right\|_2^2 > \frac{m'+n'}{m'n'}\chi_{\ell,\alpha}^2 \right) \to 1 \ .$$

This follows that

$$\chi_\ell^2 \left( \mathcal{T}(\hat{X}', \hat{Y}') \right) = \chi_\ell^2 \left( \frac{m'n'}{m'+n'} \left\| \boldsymbol{L}'^{-1}\bar{z} \right\|_2^2 \right) \to 0 \ ,$$

and hence $\chi_\ell^2(\mathcal{T}(\hat{X}', \hat{Y}')) \to 0$ as $m'n'/(m'+n') \to \infty$ if $\boldsymbol{\mu}_{\hat{\mathbb{P}}} \neq \boldsymbol{\mu}_{\hat{\mathbb{Q}}}$. $\qquad\square$

## A.3  Proof of Lemma 3

Recall the pooled covariance matrix $\Sigma_{\hat{X}',\hat{Y}'}$ and $\Sigma_{\hat{\mathbb{P}},\hat{\mathbb{Q}}} = E_{\hat{X}'\sim\hat{\mathbb{P}}^{m'}, \hat{Y}'\sim\hat{\mathbb{Q}}^{n'}}[\Sigma_{\hat{X}',\hat{Y}'}]$ in the proof of Theorem 2, and $\bar{z} = (\boldsymbol{c}_{\hat{X}'} - \boldsymbol{c}_{\hat{Y}'})$ follows a multivariate normal distribution with mean $\boldsymbol{\mu}_{\hat{\mathbb{P}}} - \boldsymbol{\mu}_{\hat{\mathbb{Q}}}$ and covariance matrix $(m' + n')\Sigma_{\hat{\mathbb{P}},\hat{\mathbb{Q}}}/m'n'$. We first observe

$$\boldsymbol{L}_{\hat{\mathbb{P}},\hat{\mathbb{Q}}} = E_{\hat{X}'\sim\hat{\mathbb{P}}^{m'}, \hat{Y}'\sim\hat{\mathbb{Q}}^{n'}}[\boldsymbol{L}'] = \Sigma_{\hat{\mathbb{P}},\hat{\mathbb{Q}}}^{1/2} \ ,$$

and this follows that

$$\boldsymbol{L}'^{-1}\bar{z} \sim \mathcal{N}\left( \boldsymbol{L}_{\hat{\mathbb{P}},\hat{\mathbb{Q}}}^{-1}(\boldsymbol{\mu}_{\hat{\mathbb{P}}} - \boldsymbol{\mu}_{\hat{\mathbb{Q}}}), \frac{m'+n'}{m'n'}\boldsymbol{I}_\ell \right) \ .$$

This is because $E[\boldsymbol{L}'^{-1}\bar{z}] = \boldsymbol{L}_{\hat{\mathbb{P}},\hat{\mathbb{Q}}}^{-1}(\boldsymbol{\mu}_{\hat{\mathbb{P}}} - \boldsymbol{\mu}_{\hat{\mathbb{Q}}})$ and the covariance matrix is given by

$$\begin{aligned}
\text{Cov}(\boldsymbol{L}'^{-1}\bar{z}, \boldsymbol{L}'^{-1}\bar{z}) &= \boldsymbol{L}_{\hat{\mathbb{P}},\hat{\mathbb{Q}}}^{-1}\left( \frac{m'+n'}{m'n'}\Sigma_{\hat{\mathbb{P}},\hat{\mathbb{Q}}} \right)\boldsymbol{L}_{\hat{\mathbb{P}},\hat{\mathbb{Q}}}^{-1} \\
&= \boldsymbol{L}_{\hat{\mathbb{P}},\hat{\mathbb{Q}}}^{-1}\left( \frac{m'+n'}{m'n'}\boldsymbol{L}_{\hat{\mathbb{P}},\hat{\mathbb{Q}}}\boldsymbol{L}_{\hat{\mathbb{P}},\hat{\mathbb{Q}}} \right)\boldsymbol{L}_{\hat{\mathbb{P}},\hat{\mathbb{Q}}}^{-1} = \frac{m'+n'}{m'n'}\boldsymbol{I}_\ell \ .
\end{aligned}$$

Write $\boldsymbol{B} = \sqrt{m'n'/(m'+n')}\boldsymbol{L}'^{-1}\text{diag}(\boldsymbol{F})\bar{z}$, and we have

$$\begin{aligned}
\boldsymbol{B}^\top \boldsymbol{B} &= \left( \sqrt{\frac{m'n'}{m'+n'}}\boldsymbol{L}'^{-1}\text{diag}(\boldsymbol{F})\bar{z} \right)^\top \left( \sqrt{\frac{m'n'}{m'+n'}}\boldsymbol{L}'^{-1}\text{diag}(\boldsymbol{F})\bar{z} \right) \\
&= \frac{m'n'}{m'+n'}\bar{z}^\top \text{diag}(\boldsymbol{F})(\boldsymbol{L}'^{-1})^\top \boldsymbol{L}'^{-1}\text{diag}(\boldsymbol{F})\bar{z} = \frac{m'n'}{m'+n'}\bar{z}^\top \text{diag}(\boldsymbol{F})\Sigma_{\hat{X}',\hat{Y}'}^{-1}\text{diag}(\boldsymbol{F})\bar{z} \\
&= \frac{m'n'}{m'+n'}\bar{z}^\top \text{diag}(\boldsymbol{F})\text{diag}(\boldsymbol{F})\Sigma_{\hat{X}',\hat{Y}'}^{-1}\bar{z} = \frac{m'n'}{m'+n'}\bar{z}^\top \text{diag}(\boldsymbol{1})\Sigma_{\hat{X}',\hat{Y}'}^{-1}\bar{z} \\
&= \frac{m'n'}{m'+n'}\bar{z}^\top \Sigma_{\hat{X}',\hat{Y}'}^{-1}\bar{z}
\end{aligned}$$

by using the symmetry of $\Sigma_{\hat{X}',\hat{Y}'}^{-1}$. It is easy to get

$$E[\boldsymbol{B}] = \sqrt{m'n'/(m'+n')}\boldsymbol{L}_{\hat{\mathbb{P}},\hat{\mathbb{Q}}}^{-1}\text{diag}(\boldsymbol{F})(\boldsymbol{\mu}_{\hat{\mathbb{P}}} - \boldsymbol{\mu}_{\hat{\mathbb{Q}}})$$

and covariance matrix is given by

$$\begin{aligned}
\text{Cov}(\boldsymbol{B}, \boldsymbol{B}) &= \sqrt{\frac{m'n'}{m'+n'}}\boldsymbol{L}_{\hat{\mathbb{P}},\hat{\mathbb{Q}}}^{-1}\text{diag}(\boldsymbol{F})\big(\frac{m'+n'}{m'n'}\Sigma_{\hat{\mathbb{P}},\hat{\mathbb{Q}}}\big)\sqrt{\frac{m'n'}{m'+n'}}\text{diag}(\boldsymbol{F})\boldsymbol{L}_{\hat{\mathbb{P}},\hat{\mathbb{Q}}}^{-1} \\
&= \sqrt{\frac{m'n'}{m'+n'}}\text{diag}(\boldsymbol{F})\boldsymbol{L}_{\hat{\mathbb{P}},\hat{\mathbb{Q}}}^{-1}\big(\frac{m'+n'}{m'n'}\boldsymbol{L}_{\hat{\mathbb{P}},\hat{\mathbb{Q}}}\boldsymbol{L}_{\hat{\mathbb{P}},\hat{\mathbb{Q}}}\big)\sqrt{\frac{m'n'}{m'+n'}}\text{diag}(\boldsymbol{F})\boldsymbol{L}_{\hat{\mathbb{P}},\hat{\mathbb{Q}}}^{-1} \\
&= \boldsymbol{I}_\ell \ .
\end{aligned}$$

This yields that

$$\boldsymbol{B} \sim \mathcal{N}\left(\sqrt{m'n'/(m'+n')}\boldsymbol{L}_{\hat{\mathbb{P}},\hat{\mathbb{Q}}}^{-1}\mathrm{diag}(\boldsymbol{F})(\boldsymbol{\mu}_{\hat{\mathbb{P}}}-\boldsymbol{\mu}_{\hat{\mathbb{Q}}}),\boldsymbol{I}_\ell\right) ,$$

and all random variables in $\boldsymbol{B}$ are mutually independent. Define

$$\bar{B} = \mathbf{1}^\top \boldsymbol{B}/\ell = \sum_{i=1}^{\ell} \boldsymbol{B}_i/\ell \quad\text{and}\quad S^2 = \sum_{i=1}^{\ell}(\boldsymbol{B}_i - \bar{B})^2 ,$$

and $\bar{B}$ is normally distributed with mean $\sqrt{m'n'/(m'+n')}\boldsymbol{F}^\top \boldsymbol{L}_{\hat{\mathbb{P}},\hat{\mathbb{Q}}}^{-1}(\boldsymbol{\mu}_{\hat{\mathbb{P}}}-\boldsymbol{\mu}_{\hat{\mathbb{Q}}})/\ell$ and variance $1/\ell$. It is easy to see that

$$\boldsymbol{F}^\top \boldsymbol{L}'^{-1}\left(\boldsymbol{c}_{\hat{X}'} - \boldsymbol{c}_{\hat{Y}'}\right) = \sqrt{(m'+n')/m'n'}\mathbf{1}^\top \boldsymbol{B} ,$$

which yields that

$$\Pr\left[\mathrm{sgn}(\boldsymbol{F}^\top \boldsymbol{L}'^{-1}\left(\boldsymbol{c}_{\hat{X}'} - \boldsymbol{c}_{\hat{Y}'}\right)) = 1\right] = \Pr\left[\sqrt{\frac{m'+n'}{m'n'}}\mathbf{1}^\top \boldsymbol{B} > 0\right] = \Pr\left[\bar{B} > 0\right] . \tag{13}$$

We further have

$$\Pr\left[\mathrm{sgn}(\boldsymbol{F}^\top \boldsymbol{L}'^{-1}\left(\boldsymbol{c}_{\hat{X}'} - \boldsymbol{c}_{\hat{Y}'}\right)) = 1\right] = 1 - \Phi\left(-\sqrt{\frac{m'n'}{(m'+n')\ell}}\boldsymbol{F}^\top \boldsymbol{L}_{\hat{\mathbb{P}},\hat{\mathbb{Q}}}^{-1}(\boldsymbol{\mu}_{\hat{\mathbb{P}}}-\boldsymbol{\mu}_{\hat{\mathbb{Q}}})\right) ,$$

where $\Phi(\cdot)$ is the cumulative distribution function of standard Gaussian distribution. For continuous normal distribution, we have

$$\Pr\left[\mathrm{sgn}(\boldsymbol{F}^\top \boldsymbol{L}'^{-1}\left(\boldsymbol{c}_{\hat{X}'} - \boldsymbol{c}_{\hat{Y}'}\right)) = 0\right] = \Pr\left[\bar{B} = 0\right] = 0 ,$$

and this follows that

$$\Pr\left[\mathrm{sgn}(\boldsymbol{F}^\top \boldsymbol{L}'^{-1}\left(\boldsymbol{c}_{\hat{X}'} - \boldsymbol{c}_{\hat{Y}'}\right)) = -1\right] = \Pr\left[\bar{B} < 0\right] = \Phi\left(-\sqrt{\frac{m'n'}{(m'+n')\ell}}\boldsymbol{F}^\top \boldsymbol{L}_{\hat{\mathbb{P}},\hat{\mathbb{Q}}}^{-1}(\boldsymbol{\mu}_{\hat{\mathbb{P}}}-\boldsymbol{\mu}_{\hat{\mathbb{Q}}})\right) .$$

Hence, the $\mathrm{sgn}(\boldsymbol{F}^\top \boldsymbol{L}'^{-1}\left(\boldsymbol{c}_{\hat{X}'} - \boldsymbol{c}_{\hat{Y}'}\right))$ follows a two-point distribution $\mathcal{TP}(\xi)$ with parameter

$$\xi = 1 - \Phi\left(-\sqrt{\frac{m'n'}{(m'+n')\ell}}\boldsymbol{F}^\top \boldsymbol{L}_{\hat{\mathbb{P}},\hat{\mathbb{Q}}}^{-1}(\boldsymbol{\mu}_{\hat{\mathbb{P}}}-\boldsymbol{\mu}_{\hat{\mathbb{Q}}})\right) .$$

This completes the proof. $\qquad\square$

### A.4 Proof of Theorem 4

Recall the pooled covariance matrix $\Sigma_{\hat{X}',\hat{Y}'}$ and $\Sigma_{\hat{\mathbb{P}},\hat{\mathbb{Q}}} = E_{\hat{X}'\sim\hat{\mathbb{P}}^{m'},\hat{Y}'\sim\hat{\mathbb{Q}}^{n'}}[\Sigma_{\hat{X}',\hat{Y}'}]$ in the proof of Theorem 2, and $\bar{z} = (\boldsymbol{c}_{\hat{X}'} - \boldsymbol{c}_{\hat{Y}'})$ follows a multivariate normal distribution with mean $\boldsymbol{\mu}_{\hat{\mathbb{P}}} - \boldsymbol{\mu}_{\hat{\mathbb{Q}}}$ and covariance matrix $(m'+n')\Sigma_{\hat{\mathbb{P}},\hat{\mathbb{Q}}}/m'n'$. We have $\boldsymbol{L}_{\hat{\mathbb{P}},\hat{\mathbb{Q}}}\boldsymbol{L}_{\hat{\mathbb{P}},\hat{\mathbb{Q}}} = \Sigma_{\hat{\mathbb{P}},\hat{\mathbb{Q}}}$.

We begin with a useful lemma and corollary as follows.

**Lemma 12.** *If* $\boldsymbol{F}^\top \boldsymbol{L}_{\hat{\mathbb{P}},\hat{\mathbb{Q}}}^{-1}\boldsymbol{\mu}_{\hat{\mathbb{P}}} > \boldsymbol{F}^\top \boldsymbol{L}_{\hat{\mathbb{P}},\hat{\mathbb{Q}}}^{-1}\boldsymbol{\mu}_{\hat{\mathbb{Q}}}$ *or null hypothesis* $H_0\colon \boldsymbol{\mu}_{\hat{\mathbb{P}}} = \boldsymbol{\mu}_{\hat{\mathbb{Q}}}$, *we have*

$$\Pr\left[\frac{m'n'}{m'+n'}\bar{z}^\top \Sigma_{\hat{X}',\hat{Y}'}^{-1}\bar{z} \geq c \mid \boldsymbol{F}^\top \boldsymbol{L}'^{-1}\bar{z} \geq 0\right] \geq \Pr\left[\frac{m'n'}{m'+n'}\bar{z}^\top \Sigma_{\hat{X}',\hat{Y}'}^{-1}\bar{z} \geq c\right] . \tag{14}$$

*Proof.* Recall $\boldsymbol{B} = \sqrt{m'n'/(m'+n')}\boldsymbol{L}'^{-1}\mathrm{diag}(\boldsymbol{F})\bar{z}$ and $\boldsymbol{F}^\top \boldsymbol{L}'^{-1}\bar{z} = \sqrt{(m'+n')/m'n'}\mathbf{1}^\top \boldsymbol{B}$ in the proof of Lemma 3, and Eqn. (14) is equivalent to

$$\Pr\left[\boldsymbol{B}^\top \boldsymbol{B} \geq c \mid \mathbf{1}^\top \boldsymbol{B} \geq 0\right] \geq \Pr\left[\boldsymbol{B}^\top \boldsymbol{B} \geq c\right] . \tag{15}$$

Recall that

$$\bar{B} = \mathbf{1}^\top \boldsymbol{B}/\ell = \sum_{i=1}^{\ell} \boldsymbol{B}_i/\ell \quad\text{and}\quad S^2 = \sum_{i=1}^{\ell}(\boldsymbol{B}_i - \bar{B})^2 ,$$

and from Eqn. (15), we have

$$\Pr\left[\bar{B}^2 \geq (c - S^2)/\ell \mid \bar{B} \geq 0\right] \geq \Pr\left[\bar{B}^2 \geq (c - S^2)/\ell\right] .$$

From the independence of $S^2$ and $\bar{B}$, it is sufficient to prove that, for every $\delta \geq 0$,

$$\Pr\left[\bar{B}^2 \geq \delta \mid \bar{B} \geq 0\right] \geq \Pr\left[\bar{B}^2 \geq \delta\right] . \tag{16}$$

It's easy to see that $\bar{B}$ is normally distributed with mean $\sqrt{m'n'/(m'+n')}\boldsymbol{F}^\top \boldsymbol{L}_{\hat{\mathbb{P}},\hat{\mathbb{Q}}}^{-1}(\boldsymbol{\mu}_{\hat{\mathbb{P}}} - \boldsymbol{\mu}_{\hat{\mathbb{Q}}})/\ell$ and variance $1/\ell$. We define

$$a = \sqrt{\ell\delta} \quad \text{and} \quad b = \sqrt{\frac{m'n'}{(m'+n')\ell}}\boldsymbol{F}^\top \boldsymbol{L}_{\hat{\mathbb{P}},\hat{\mathbb{Q}}}^{-1}(\boldsymbol{\mu}_{\hat{\mathbb{P}}} - \boldsymbol{\mu}_{\hat{\mathbb{Q}}}) ,$$

and from Eqn. (16), we have

$$\frac{\Phi(-b)}{\Phi(-b-a)} \geq \frac{\Phi(b)}{\Phi(b-a)} ,$$

where the equality holds from $b = 0$, i.e., $\boldsymbol{\mu}_{\hat{\mathbb{P}}} = \boldsymbol{\mu}_{\hat{\mathbb{Q}}}$. This completes the proof. $\qquad\square$

**Corollary 13.** *If $\boldsymbol{F}^\top \boldsymbol{L}_{\hat{\mathbb{P}},\hat{\mathbb{Q}}}^{-1}\boldsymbol{\mu}_{\hat{\mathbb{P}}} < \boldsymbol{F}^\top \boldsymbol{L}_{\hat{\mathbb{P}},\hat{\mathbb{Q}}}^{-1}\boldsymbol{\mu}_{\hat{\mathbb{Q}}}$, we have*

$$\Pr\left[\frac{m'n'}{m'+n'}\bar{\boldsymbol{z}}^\top \Sigma_{\hat{X}',\hat{Y}'}^{-1}\bar{\boldsymbol{z}} \geq c \mid \boldsymbol{F}^\top \boldsymbol{L}'^{-1}\bar{\boldsymbol{z}} < 0\right] \geq \Pr\left[\frac{m'n'}{m'+n'}\bar{\boldsymbol{z}}^\top \Sigma_{\hat{X}',\hat{Y}'}^{-1}\bar{\boldsymbol{z}} \geq c\right] ,$$

**Lemma 14.** *If $\boldsymbol{F}^\top \boldsymbol{L}_{\hat{\mathbb{P}},\hat{\mathbb{Q}}}^{-1}\boldsymbol{\mu}_{\hat{\mathbb{P}}} > \boldsymbol{F}^\top \boldsymbol{L}_{\hat{\mathbb{P}},\hat{\mathbb{Q}}}^{-1}\boldsymbol{\mu}_{\hat{\mathbb{Q}}}$, then the test power of our bi-directional hypothesis can be lower bounded by*

$$q(\ell, \nu - \ell, \lambda, \beta\alpha) \cdot \xi + \Phi\left(-(\chi_{\ell,(2-\beta)\alpha}^2)^{1/2} - (\omega/\ell)^{1/2}\boldsymbol{F}^\top \boldsymbol{L}_{\hat{\mathbb{P}},\hat{\mathbb{Q}}}^{-1}(\boldsymbol{\mu}_{\hat{\mathbb{P}}} - \boldsymbol{\mu}_{\hat{\mathbb{Q}}})\right) ,$$

*where $\omega = m'n'/(m'+n')$, $\nu = m' + n' - 1$ and $\lambda = \omega\|\boldsymbol{L}_{\hat{\mathbb{P}},\hat{\mathbb{Q}}}^{-1}(\boldsymbol{\mu}_{\hat{\mathbb{P}}} - \boldsymbol{\mu}_{\hat{\mathbb{Q}}})\|_2^2$.*

*Proof.* From Lemma 12, we first have, for $\boldsymbol{F}^\top \boldsymbol{L}_{\hat{\mathbb{P}},\hat{\mathbb{Q}}}^{-1}\boldsymbol{\mu}_{\hat{\mathbb{P}}} > \boldsymbol{F}^\top \boldsymbol{L}_{\hat{\mathbb{P}},\hat{\mathbb{Q}}}^{-1}\boldsymbol{\mu}_{\hat{\mathbb{Q}}}$,

$$\begin{aligned}
\Pr[h = 1 \mid \boldsymbol{F}^\top \boldsymbol{L}'^{-1}\bar{\boldsymbol{z}} \geq 0] &= \Pr\left[\frac{m'n'}{m'+n'}\bar{\boldsymbol{z}}^\top \Sigma_{\hat{X}',\hat{Y}'}^{-1}\bar{\boldsymbol{z}} \geq \chi_{\ell,\beta\alpha}^2 \mid \boldsymbol{F}^\top \boldsymbol{L}'^{-1}\bar{\boldsymbol{z}} \geq 0\right] \\
&> \Pr\left[\frac{m'n'}{m'+n'}\bar{\boldsymbol{z}}^\top \Sigma_{\hat{X}',\hat{Y}'}^{-1}\bar{\boldsymbol{z}} \geq \chi_{\ell,\beta\alpha}^2\right] , \tag{17}
\end{aligned}$$

by substituting $c = \chi_{\ell,\beta\alpha}^2$ into Eqn. (14). From $\boldsymbol{\mu}_{\hat{\mathbb{P}}} \neq \boldsymbol{\mu}_{\hat{\mathbb{Q}}}$, we also have, from the work of [84],

$$\frac{m'+n'-\ell-1}{(m'+n'-2)\ell}\mathcal{T}(\hat{X}', \hat{Y}') = \frac{m'+n'-\ell-1}{(m'+n'-2)\ell} \times \frac{m'n'}{m'+n'}\bar{\boldsymbol{z}}^\top \Sigma_{\hat{X}',\hat{Y}'}^{-1}\bar{\boldsymbol{z}} \sim F(\ell, m'+n'-1-\ell, \lambda) ,$$

where $\lambda = m'n'(\boldsymbol{\mu}_{\hat{\mathbb{P}}} - \boldsymbol{\mu}_{\hat{\mathbb{Q}}})^\top \Sigma_{\hat{\mathbb{P}},\hat{\mathbb{Q}}}^{-1}(\boldsymbol{\mu}_{\hat{\mathbb{P}}} - \boldsymbol{\mu}_{\hat{\mathbb{Q}}})/(m'+n')$. Given the significance level $\beta\alpha$, we have

$$\Pr\left[\frac{m'+n'-\ell-1}{(m'+n'-2)\ell}\mathcal{T}(\hat{X}', \hat{Y}') \geq F_{\ell,m'+n'-1-\ell,\beta\alpha}\right] = q(\ell, m'+n'-1-\ell, \lambda, \beta\alpha) , \tag{18}$$

and this follows that

$$\Pr\left[\frac{m'n'}{m'+n'}\bar{\boldsymbol{z}}^\top \Sigma_{\hat{X}',\hat{Y}'}^{-1}\bar{\boldsymbol{z}} \geq \chi_{\ell,\beta\alpha}^2\right] = q(\ell, m'+n'-1-\ell, \lambda, \beta\alpha) .$$

Recall $\boldsymbol{B} = \sqrt{m'n'/(m'+n')}\boldsymbol{L}'^{-1}\mathrm{diag}(\boldsymbol{F})\bar{\boldsymbol{z}}$ and $\boldsymbol{F}^\top \boldsymbol{L}'^{-1}\bar{\boldsymbol{z}} = \sqrt{(m'+n')/m'n'}\mathbf{1}^\top \boldsymbol{B}$ in the proof of Lemma 3, and that $\bar{B} = \mathbf{1}^\top \boldsymbol{B}/\ell$ is normally distributed with mean $\sqrt{m'n'/(m'+n')}\boldsymbol{F}^\top \boldsymbol{L}_{\hat{\mathbb{P}},\hat{\mathbb{Q}}}^{-1}(\boldsymbol{\mu}_{\hat{\mathbb{P}}} - \boldsymbol{\mu}_{\hat{\mathbb{Q}}})/\ell$ and variance $1/\ell$. We have

$$\Pr[\boldsymbol{F}^\top \boldsymbol{L}'^{-1}\bar{\boldsymbol{z}} > 0] = \Pr[\bar{B} \geq 0] = 1 - \Phi\left(-\sqrt{\frac{m'n'}{(m'+n')\ell}}\boldsymbol{F}^\top \boldsymbol{L}_{\hat{\mathbb{P}},\hat{\mathbb{Q}}}^{-1}(\boldsymbol{\mu}_{\hat{\mathbb{P}}} - \boldsymbol{\mu}_{\hat{\mathbb{Q}}})\right) , \tag{19}$$

since $\bar{B}$ is normally distributed with mean $\sqrt{m'n'/(m'+n')}\boldsymbol{F}^\top \boldsymbol{L}_{\hat{\mathbb{P}},\hat{\mathbb{Q}}}^{-1}(\boldsymbol{\mu}_{\hat{\mathbb{P}}}-\boldsymbol{\mu}_{\hat{\mathbb{Q}}})/\ell$ and variance $1/\ell$. Combining with Eqns. (17)-(19), we have

$$\Pr\left[\frac{m'n'}{m'+n'}\bar{\boldsymbol{z}}^\top \Sigma_{\hat{X}',\hat{Y}'}^{-1}\bar{\boldsymbol{z}} \geq \chi_{\ell,\beta\alpha}^2 \cap \boldsymbol{F}^\top \boldsymbol{L}'^{-1}\bar{\boldsymbol{z}} \geq 0\right]$$

$$= \Pr\left[\frac{m'n'}{m'+n'}\bar{\boldsymbol{z}}^\top \Sigma_{\hat{X}',\hat{Y}'}^{-1}\bar{\boldsymbol{z}} \geq \chi_{\ell,\beta\alpha}^2 \mid \boldsymbol{F}^\top \boldsymbol{L}'^{-1}\bar{\boldsymbol{z}} \geq 0\right]\Pr[\boldsymbol{F}^\top \boldsymbol{L}'^{-1}\bar{\boldsymbol{z}} \geq 0]$$

$$> \Pr\left[\frac{m'n'}{m'+n'}\bar{\boldsymbol{z}}^\top \Sigma_{\hat{X}',\hat{Y}'}^{-1}\bar{\boldsymbol{z}} \geq \chi_{\ell,\beta\alpha}^2\right]\Pr[\boldsymbol{F}^\top \boldsymbol{L}'^{-1}\bar{\boldsymbol{z}} \geq 0]$$

$$= q(\ell, m'+n'-1-\ell, \lambda, \beta\alpha)\left(1-\Phi(-\sqrt{\frac{m'n'}{(m'+n')\ell}}\boldsymbol{F}^\top \boldsymbol{L}_{\hat{\mathbb{P}},\hat{\mathbb{Q}}}^{-1}(\boldsymbol{\mu}_{\hat{\mathbb{P}}}-\boldsymbol{\mu}_{\hat{\mathbb{Q}}}))\right). \quad (20)$$

For $\boldsymbol{F}^\top \boldsymbol{L}_{\hat{\mathbb{P}},\hat{\mathbb{Q}}}^{-1}\boldsymbol{\mu}_{\hat{\mathbb{P}}} > \boldsymbol{F}^\top \boldsymbol{L}_{\hat{\mathbb{P}},\hat{\mathbb{Q}}}^{-1}\boldsymbol{\mu}_{\hat{\mathbb{Q}}}$, we substitute $c = \chi_{\ell,(2-\beta)\alpha}^2$ into Eqn. (14), and it holds that

$$\Pr\left[\frac{m'n'}{m'+n'}\bar{\boldsymbol{z}}^\top \Sigma_{\hat{X}',\hat{Y}'}^{-1}\bar{\boldsymbol{z}} \geq \chi_{\ell,(2-\beta)\alpha}^2 \cap \boldsymbol{F}^\top \boldsymbol{L}'^{-1}\bar{\boldsymbol{z}} < 0\right]$$

$$= \Pr\left[\bar{B}^2 \geq (\chi_{\ell,(2-\beta)\alpha}^2 - S^2)/\ell \cap \bar{B} < 0\right]$$

$$= \Pr\left[\bar{B} \leq -\sqrt{(\chi_{\ell,(2-\beta)\alpha}^2 - S^2)/\ell}\right]$$

$$= \Phi\left[-\sqrt{\chi_{\ell,(2-\beta)\alpha}^2 - S^2} - \sqrt{\frac{m'n'}{(m'+n')\ell}}\boldsymbol{F}^\top \boldsymbol{L}_{\hat{\mathbb{P}},\hat{\mathbb{Q}}}^{-1}(\boldsymbol{\mu}_{\hat{\mathbb{P}}}-\boldsymbol{\mu}_{\hat{\mathbb{Q}}})\right]$$

$$> \Phi\left[-\sqrt{\chi_{\ell,(2-\beta)\alpha}^2} - \sqrt{\frac{m'n'}{(m'+n')\ell}}\boldsymbol{F}^\top \boldsymbol{L}_{\hat{\mathbb{P}},\hat{\mathbb{Q}}}^{-1}(\boldsymbol{\mu}_{\hat{\mathbb{P}}}-\boldsymbol{\mu}_{\hat{\mathbb{Q}}})\right]. \quad (21)$$

Combining with Eqns. (20)-(21), we give a lower bound for the test power of bi-directional hypothesis

$$\Pr\left[h=1\right] = \Pr\left[h=1 \cap \boldsymbol{F}^\top \boldsymbol{L}'^{-1}\bar{\boldsymbol{z}} > 0\right] + \Pr\left[h=1 \cap \boldsymbol{F}^\top \boldsymbol{L}'^{-1}\bar{\boldsymbol{z}} < 0\right]$$

$$= \Pr\left[\frac{m'n'}{m'+n'}\bar{\boldsymbol{z}}^\top \Sigma_{\hat{X}',\hat{Y}'}^{-1}\bar{\boldsymbol{z}} \geq \chi_{\ell,\beta\alpha}^2 \cap \boldsymbol{F}^\top \boldsymbol{L}'^{-1}\bar{\boldsymbol{z}} \geq 0\right]$$

$$+ \Pr\left[\frac{m'n'}{m'+n'}\bar{\boldsymbol{z}}^\top \Sigma_{\hat{X}',\hat{Y}'}^{-1}\bar{\boldsymbol{z}} \geq \chi_{\ell,(2-\beta)\alpha}^2 \cap \boldsymbol{F}^\top \boldsymbol{L}'^{-1}\bar{\boldsymbol{z}} < 0\right]$$

$$> q(\ell, m'+n'-1-\ell, \lambda, \beta\alpha)\left(1-\Phi(-\sqrt{\frac{m'n'}{(m'+n')\ell}}\boldsymbol{F}^\top \boldsymbol{L}_{\hat{\mathbb{P}},\hat{\mathbb{Q}}}^{-1}(\boldsymbol{\mu}_{\hat{\mathbb{P}}}-\boldsymbol{\mu}_{\hat{\mathbb{Q}}}))\right)$$

$$+ \Phi\left(-\sqrt{\chi_{\ell,(2-\beta)\alpha}^2} - \sqrt{\frac{m'n'}{(m'+n')\ell}}\boldsymbol{F}^\top \boldsymbol{L}_{\hat{\mathbb{P}},\hat{\mathbb{Q}}}^{-1}(\boldsymbol{\mu}_{\hat{\mathbb{P}}}-\boldsymbol{\mu}_{\hat{\mathbb{Q}}})\right).$$

$$\square$$

**Lemma 15.** *If $\boldsymbol{F}^\top \boldsymbol{L}_{\hat{\mathbb{P}},\hat{\mathbb{Q}}}^{-1}\boldsymbol{\mu}_{\hat{\mathbb{P}}} < \boldsymbol{F}^\top \boldsymbol{L}_{\hat{\mathbb{P}},\hat{\mathbb{Q}}}^{-1}\boldsymbol{\mu}_{\hat{\mathbb{Q}}}$, then the test power of our bi-directional hypothesis can be lower bounded by*

$$q(\ell, \nu - \ell, \lambda, (2-\beta)\alpha)(1-\xi) + 1 - \Phi\left((\chi_{\ell,\beta\alpha}^2)^{1/2} - (\omega/\ell)^{1/2}\boldsymbol{F}^\top \boldsymbol{L}_{\hat{\mathbb{P}},\hat{\mathbb{Q}}}^{-1}(\boldsymbol{\mu}_{\hat{\mathbb{P}}}-\boldsymbol{\mu}_{\hat{\mathbb{Q}}})\right),$$

*where $\omega = m'n'/(m'+n')$, $\nu = m'+n'-1$ and $\lambda = \omega\|\boldsymbol{L}_{\hat{\mathbb{P}},\hat{\mathbb{Q}}}^{-1}(\boldsymbol{\mu}_{\hat{\mathbb{P}}}-\boldsymbol{\mu}_{\hat{\mathbb{Q}}})\|_2^2$.*

*Proof.* From Corollary 13, we have, for $\boldsymbol{F}^\top \boldsymbol{L}_{\hat{\mathbb{P}},\hat{\mathbb{Q}}}^{-1}\boldsymbol{\mu}_{\hat{\mathbb{P}}} < \boldsymbol{F}^\top \boldsymbol{L}_{\hat{\mathbb{P}},\hat{\mathbb{Q}}}^{-1}\boldsymbol{\mu}_{\hat{\mathbb{Q}}}$,

$$\Pr[h=1 \mid \boldsymbol{F}^\top \boldsymbol{L}'^{-1}\bar{\boldsymbol{z}} \leq 0] = \Pr\left[\frac{m'n'}{m'+n'}\bar{\boldsymbol{z}}^\top \Sigma_{\hat{X}',\hat{Y}'}^{-1}\bar{\boldsymbol{z}} \geq \chi_{\ell,(2-\beta)\alpha}^2 \mid \boldsymbol{F}^\top \boldsymbol{L}'^{-1}\bar{\boldsymbol{z}} < 0\right]$$

$$> \Pr\left[\frac{m'n'}{m'+n'}\bar{\boldsymbol{z}}^\top \Sigma_{\hat{X}',\hat{Y}'}^{-1}\bar{\boldsymbol{z}} \geq \chi_{\ell,(2-\beta)\alpha}^2\right],$$

Recall that
$$\Pr\left[\frac{m'n'}{m'+n'}\bar{z}^\top \Sigma^{-1}_{\hat{X}',\hat{Y}'}\bar{z} \geq \chi^2_{\ell,(2-\beta)\alpha}\right] = q(\ell, m'+n'-1-\ell, \lambda, (2-\beta)\alpha)$$
in the proof of Lemma 14, and this follows that

$$
\Pr\left[\frac{m'n'}{m'+n'}\bar{z}^\top \Sigma^{-1}_{\hat{X}',\hat{Y}'}\bar{z} \geq \chi^2_{\ell,(2-\beta)\alpha} \cap \boldsymbol{F}^\top \boldsymbol{L}'^{-1}\bar{z} < 0\right]
$$
$$
= \Pr\left[\frac{m'n'}{m'+n'}\bar{z}^\top \Sigma^{-1}_{\hat{X}',\hat{Y}'}\bar{z} \geq \chi^2_{\ell,(2-\beta)\alpha} \mid \boldsymbol{F}^\top \boldsymbol{L}'^{-1}\bar{z} < 0\right]\Pr[\boldsymbol{F}^\top \boldsymbol{L}'^{-1}\bar{z} < 0]
$$
$$
> \Pr\left[\frac{m'n'}{m'+n'}\bar{z}^\top \Sigma^{-1}_{\hat{X}',\hat{Y}'}\bar{z} \geq \chi^2_{\ell,(2-\beta)\alpha}\right]\Pr[\boldsymbol{F}^\top \boldsymbol{L}'^{-1}\bar{z} < 0]
$$
$$
= q(\ell, m'+n'-1-\ell, \lambda, (2-\beta)\alpha)\,\Phi\left(-\sqrt{\frac{m'n'}{(m'+n')\ell}}\boldsymbol{F}^\top \boldsymbol{L}^{-1}_{\hat{\mathbb{P}},\hat{\mathbb{Q}}}(\boldsymbol{\mu}_{\hat{\mathbb{P}}} - \boldsymbol{\mu}_{\hat{\mathbb{Q}}})\right). \quad (22)
$$

Recall $\boldsymbol{B} = \sqrt{m'n'/(m'+n')}\boldsymbol{L}'^{-1}\mathrm{diag}(\boldsymbol{F})\bar{z}$ and $\boldsymbol{F}^\top \boldsymbol{L}'^{-1}\bar{z} = \sqrt{(m'+n')/m'n'}\mathbf{1}^\top \boldsymbol{B}$ in the proof of Lemma 3, and that $\bar{B} = \mathbf{1}^\top \boldsymbol{B}/\ell$ is normally distributed with mean $\sqrt{m'n'/(m'+n')}\boldsymbol{F}^\top \boldsymbol{L}^{-1}_{\hat{\mathbb{P}},\hat{\mathbb{Q}}}(\boldsymbol{\mu}_{\hat{\mathbb{P}}} - \boldsymbol{\mu}_{\hat{\mathbb{Q}}})/\ell$ and variance $1/\ell$. We have, if $\boldsymbol{F}^\top \boldsymbol{L}^{-1}_{\hat{\mathbb{P}},\hat{\mathbb{Q}}}\boldsymbol{\mu}_{\hat{\mathbb{P}}} < \boldsymbol{F}^\top \boldsymbol{L}^{-1}_{\hat{\mathbb{P}},\hat{\mathbb{Q}}}\boldsymbol{\mu}_{\hat{\mathbb{Q}}}$,

$$
\Pr\left[\frac{m'n'}{m'+n'}\bar{z}^\top \Sigma^{-1}_{\hat{X}',\hat{Y}'}\bar{z} \geq \chi^2_{\ell,\beta\alpha} \cap \boldsymbol{F}^\top \boldsymbol{L}'^{-1}\bar{z} \geq 0\right]
$$
$$
= \Pr\left[\bar{B}^2 \geq (\chi^2_{\ell,\beta\alpha} - S^2)/\ell \cap \bar{B} \geq 0\right]
$$
$$
= \Pr\left[\bar{B} \geq \sqrt{(\chi^2_{\ell,\beta\alpha} - S^2)/\ell}\right]
$$
$$
= 1 - \Phi\left(\sqrt{\chi^2_{\ell,\beta\alpha} - S^2} - \sqrt{\frac{m'n'}{(m'+n')\ell}}\boldsymbol{F}^\top \boldsymbol{L}^{-1}_{\hat{\mathbb{P}},\hat{\mathbb{Q}}}(\boldsymbol{\mu}_{\hat{\mathbb{P}}} - \boldsymbol{\mu}_{\hat{\mathbb{Q}}})\right)
$$
$$
> 1 - \Phi\left(\sqrt{\chi^2_{\ell,\beta\alpha}} - \sqrt{\frac{m'n'}{(m'+n')\ell}}\boldsymbol{F}^\top \boldsymbol{L}^{-1}_{\hat{\mathbb{P}},\hat{\mathbb{Q}}}(\boldsymbol{\mu}_{\hat{\mathbb{P}}} - \boldsymbol{\mu}_{\hat{\mathbb{Q}}})\right). \quad (23)
$$

Combining with Eqns. (22)-(23), we give a lower bound for the test power of bi-directional hypothesis
$$
\Pr[h=1] = \Pr\left[h=1 \cap \boldsymbol{F}^\top \boldsymbol{L}'^{-1}\bar{z} \geq 0\right] + \Pr\left[h=1 \cap \boldsymbol{F}^\top \boldsymbol{L}'^{-1}\bar{z} < 0\right]
$$
$$
= \Pr\left[\frac{m'n'}{m'+n'}\bar{z}^\top \Sigma^{-1}_{\hat{X}',\hat{Y}'}\bar{z} \geq \chi^2_{\ell,\beta\alpha} \cap \boldsymbol{F}^\top \boldsymbol{L}'^{-1}\bar{z} \geq 0\right]
$$
$$
+ \Pr\left[\frac{m'n'}{m'+n'}\bar{z}^\top \Sigma^{-1}_{\hat{X}',\hat{Y}'}\bar{z} \geq \chi^2_{\ell,(2-\beta)\alpha} \cap \boldsymbol{F}^\top \boldsymbol{L}'^{-1}\bar{z} < 0\right]
$$
$$
> q(\ell, m'+n'-1-\ell, \lambda, (2-\beta)\alpha)\Phi\left(-\sqrt{\frac{m'n'}{(m'+n')\ell}}\boldsymbol{F}^\top \boldsymbol{L}^{-1}_{\hat{\mathbb{P}},\hat{\mathbb{Q}}}(\boldsymbol{\mu}_{\hat{\mathbb{P}}} - \boldsymbol{\mu}_{\hat{\mathbb{Q}}})\right)
$$
$$
+ 1 - \Phi\left(\sqrt{\chi^2_{\ell,\beta\alpha}} - \sqrt{\frac{m'n'}{(m'+n')\ell}}\boldsymbol{F}^\top \boldsymbol{L}^{-1}_{\hat{\mathbb{P}},\hat{\mathbb{Q}}}(\boldsymbol{\mu}_{\hat{\mathbb{P}}} - \boldsymbol{\mu}_{\hat{\mathbb{Q}}})\right).
$$

This completes the proof. $\qquad\square$

**Lemma 16.** *For our bi-directional hypothesis, the type-I error rate is equal to $\alpha$ if $\boldsymbol{\mu}_{\hat{\mathbb{P}}} = \boldsymbol{\mu}_{\hat{\mathbb{Q}}}$.*

*Proof.* We first consider the case $\boldsymbol{F}^\top \boldsymbol{L}'^{-1}\bar{z} > 0$. By substituting $c = \chi^2_{\ell,\beta\alpha}$ into Eqn. (14), we have
$$
\Pr[h=1 | \boldsymbol{F}^\top \boldsymbol{L}'^{-1}\bar{z} \geq 0]
$$
$$
= \Pr\left[\frac{m'n'}{m'+n'}\bar{z}^\top \Sigma^{-1}_{\hat{X}',\hat{Y}'}\bar{z} \geq \chi^2_{\ell,\beta\alpha} \mid \boldsymbol{F}^\top \boldsymbol{L}'^{-1}\bar{z} \geq 0\right]
$$
$$
= \Pr\left[\frac{m'n'}{m'+n'}\bar{z}^\top \Sigma^{-1}_{\hat{X}',\hat{Y}'}\bar{z} \geq \chi^2_{\ell,\beta\alpha}\right]
$$
$$
= \beta\alpha.
$$

Recall $\boldsymbol{B} = \sqrt{m'n'/(m'+n')}\boldsymbol{L}'^{-1}\mathrm{diag}(\boldsymbol{F})\bar{\boldsymbol{z}}$ and $\boldsymbol{F}^\top \boldsymbol{L}'^{-1}\bar{\boldsymbol{z}} = \sqrt{(m'+n')/m'n'}\mathbf{1}^\top \boldsymbol{B}$ in the proof of Lemma 3, and that $\bar{B} = \mathbf{1}^\top \boldsymbol{B}/\ell$ is normally distributed with mean $\sqrt{m'n'/(m'+n')}\boldsymbol{F}^\top \boldsymbol{L}_{\hat{\mathbb{P}},\hat{\mathbb{Q}}}^{-1}(\boldsymbol{\mu}_{\hat{\mathbb{P}}} - \boldsymbol{\mu}_{\hat{\mathbb{Q}}})/\ell$ and variance $1/\ell$. We have

$$\Pr[\boldsymbol{F}^\top \boldsymbol{L}'^{-1}\bar{\boldsymbol{z}} \geq 0] = \Pr[\bar{B} \geq 0] = 1/2\,,$$

and we have

$$P\left[\frac{m'n'}{m'+n'}\bar{\boldsymbol{z}}^\top \Sigma_{\hat{X}',\hat{Y}'}^{-1}\bar{\boldsymbol{z}} \geq \chi_{\ell,\beta\alpha}^2 \cap \boldsymbol{F}^\top \boldsymbol{L}'^{-1}\bar{\boldsymbol{z}} \geq 0\right]$$

$$= \Pr\left[\frac{m'n'}{m'+n'}\bar{\boldsymbol{z}}^\top \Sigma_{\hat{X}',\hat{Y}'}^{-1}\bar{\boldsymbol{z}} \geq \chi_{\ell,\beta\alpha}^2 \mid \boldsymbol{F}^\top \boldsymbol{L}'^{-1}\bar{\boldsymbol{z}} \geq 0\right]\Pr[\boldsymbol{F}^\top \boldsymbol{L}'^{-1}\bar{\boldsymbol{z}} \geq 0] = \frac{\beta\alpha}{2}\,, \quad (24)$$

since $\bar{B}$ is normally distributed with mean 0.

For the case $\boldsymbol{F}^\top \boldsymbol{L}'^{-1}\bar{\boldsymbol{z}} < 0$, we similarly substitute $c = \chi_{\ell,(2-\beta)\alpha}^2$ into Eqn. (14), and it follows that

$$\Pr\left[\frac{m'n'}{m'+n'}\bar{\boldsymbol{z}}^\top \Sigma_{\hat{X}',\hat{Y}'}^{-1}\bar{\boldsymbol{z}} \geq \chi_{\ell,(2-\beta)\alpha}^2 \cap \boldsymbol{F}^\top \boldsymbol{L}'^{-1}\bar{\boldsymbol{z}} < 0\right]$$

$$= \Pr\left[\frac{m'n'}{m'+n'}\bar{\boldsymbol{z}}^\top \Sigma_{\hat{X}',\hat{Y}'}^{-1}\bar{\boldsymbol{z}} \geq \chi_{\ell,(2-\beta)\alpha}^2\right]$$

$$\quad - \Pr\left[\frac{m'n'}{m'+n'}\bar{\boldsymbol{z}}^\top \Sigma_{\hat{X}',\hat{Y}'}^{-1}\bar{\boldsymbol{z}} \geq \chi_{\ell,(2-\beta)\alpha}^2 \cap \boldsymbol{F}^\top \boldsymbol{L}'^{-1}\bar{\boldsymbol{z}} \geq 0\right]$$

$$= \Pr\left[\frac{m'n'}{m'+n'}\bar{\boldsymbol{z}}^\top \Sigma_{\hat{X}',\hat{Y}'}^{-1}\bar{\boldsymbol{z}} \geq \chi_{\ell,(2-\beta)\alpha}^2\right]$$

$$\quad - \Pr\left[\frac{m'n'}{m'+n'}\bar{\boldsymbol{z}}^\top \Sigma_{\hat{X}',\hat{Y}'}^{-1}\bar{\boldsymbol{z}} \geq \chi_{\ell,(2-\beta)\alpha}^2\right]\Pr[\boldsymbol{F}^\top \boldsymbol{L}'^{-1}\bar{\boldsymbol{z}} \geq 0]$$

$$= \Pr\left[\frac{m'n'}{m'+n'}\bar{\boldsymbol{z}}^\top \Sigma_{\hat{X}',\hat{Y}'}^{-1}\bar{\boldsymbol{z}} \geq \chi_{\ell,(2-\beta)\alpha}^2\right] - \frac{1}{2}\Pr\left[\frac{m'n'}{m'+n'}\bar{\boldsymbol{z}}^\top \Sigma_{\hat{X}',\hat{Y}'}^{-1}\bar{\boldsymbol{z}} \geq \chi_{\ell,(2-\beta)\alpha}^2\right]$$

$$= \frac{1}{2}\Pr\left[\frac{m'n'}{m'+n'}\bar{\boldsymbol{z}}^\top \Sigma_{\hat{X}',\hat{Y}'}^{-1}\bar{\boldsymbol{z}} \geq \chi_{\ell,(2-\beta)\alpha}^2\right] = (1-\beta/2)\alpha\,. \quad (25)$$

From Eqns. (24)-(25), we have

$$\Pr[h=1] = \Pr\left[h=1 \cap \boldsymbol{F}^\top \boldsymbol{L}'^{-1}\bar{\boldsymbol{z}} > 0\right] + \Pr\left[h=1 \cap \boldsymbol{F}^\top \boldsymbol{L}'^{-1}\bar{\boldsymbol{z}} < 0\right]$$

$$= \Pr\left[\frac{m'n'}{m'+n'}\bar{\boldsymbol{z}}^\top \Sigma_{\hat{X}',\hat{Y}'}^{-1}\bar{\boldsymbol{z}} \geq \chi_{\ell,\beta\alpha}^2 \cap \boldsymbol{F}^\top \boldsymbol{L}'^{-1}\bar{\boldsymbol{z}} \geq 0\right]$$

$$\quad + \Pr\left[\frac{m'n'}{m'+n'}\bar{\boldsymbol{z}}^\top \Sigma_{\hat{X}',\hat{Y}'}^{-1}\bar{\boldsymbol{z}} \geq \chi_{\ell,(2-\beta)\alpha}^2 \cap \boldsymbol{F}^\top \boldsymbol{L}'^{-1}\bar{\boldsymbol{z}} < 0\right]$$

$$= \frac{\beta\alpha}{2} + (1-\beta/2)\alpha = \alpha\,.$$

This completes the proof. $\qquad\square$

Theorem 4 follows from Lemmas 14-16.

## A.5   Proof of Theorem 5

Recall that $\mathcal{B}_{\langle 1\rangle}, \mathcal{B}_{\langle 2\rangle}, \ldots, \mathcal{B}_{\langle s\rangle}$ are rectangle regions of a non-increasing order w.r.t. $g(\cdot,\cdot)$. For each rectangle region $\mathcal{B}_{\langle i\rangle}$, we could define its local null hypothesis

$$H_{0,\langle i\rangle}: \boldsymbol{\mu}_{\hat{\mathbb{P}}_{\mathcal{B}_{\langle i\rangle}}} = \boldsymbol{\mu}_{\hat{\mathbb{Q}}_{\mathcal{B}_{\langle i\rangle}}} \quad \text{with} \quad \boldsymbol{\mu}_{\hat{\mathbb{P}}_{\mathcal{B}_{\langle i\rangle}}} = E_{\hat{\boldsymbol{x}}'\sim\hat{\mathbb{P}}_{\mathcal{B}_{\langle i\rangle}}}[\hat{\boldsymbol{x}}'] \quad \text{and} \quad \boldsymbol{\mu}_{\hat{\mathbb{Q}}_{\mathcal{B}_{\langle i\rangle}}} = E_{\hat{\boldsymbol{y}}'\sim\hat{\mathbb{Q}}_{\mathcal{B}_{\langle i\rangle}}}[\hat{\boldsymbol{y}}']\,.$$

From Theorem 2, the testing statistic $\mathcal{T}(X'_{\mathcal{B}_{\langle i\rangle}}, Y'_{\mathcal{B}_{\langle i\rangle}})$ follows the $\chi^2$ distribution with freedom of $\ell$ degrees under the local null hypothesis $H_{0,\langle i\rangle}$. Denote by $\chi_\ell^2(\mathcal{T}(\hat{X}'_{\mathcal{B}_i}, \hat{Y}'_{\mathcal{B}_i}))$ the $p$-value, and we have

**Lemma 17.** *[68] The p-value $\chi_\ell^2(\mathcal{T}(\hat{X}'_{\mathcal{B},\langle i \rangle}, \hat{Y}'_{\mathcal{B},\langle i \rangle}))$ follows a uniform distribution $\mathcal{U}[0,1]$ under the local null hypothesis $H_{0,\langle i \rangle} \colon \boldsymbol{\mu}_{\hat{\mathbb{P}}_{\mathcal{B}_{\langle i \rangle}}} = \boldsymbol{\mu}_{\hat{\mathbb{Q}}_{\mathcal{B}_{\langle i \rangle}}}.$*

We take an interactive multi-step testing procedure to identify the index set of rectangle regions of local significant differences. Define the candidate rejection set $\mathcal{R}(t) = \{H_{0,\langle i \rangle}\}_{i=1}^t$ for $1 \leq t \leq s$ with $\mathcal{R}(0) = \emptyset$, and exclude one null hypothesis $H_{0,\langle t \rangle}$ at the $t$-th step. We could generate a sequence as follows:

$$\{H_{0,\langle i \rangle}\}_{i=1}^s = \mathcal{R}(s) \supseteq \mathcal{R}(s-1) \supseteq \mathcal{R}(s-2) \supseteq \cdots \supseteq \mathcal{R}(0) = \emptyset \, . \tag{26}$$

and it holds that $H_{0,\langle i \rangle} = \mathcal{R}(i) \setminus \mathcal{R}(i-1)$. Recall the local bi-directional hypothesis

$$h_{\langle i \rangle} = h(\hat{X}'_{\mathcal{B}_{\langle i \rangle}}, \hat{Y}'_{\mathcal{B}_{\langle i \rangle}}) \quad \text{for} \quad i \in [s] \, .$$

Denote by $p_*$ the parameter of significant level for the local two-sample test and masked $p$-value. From Theorem 4, we have $\Pr[h_{\langle i \rangle} = 1] = p_*$ under the null local hypothesis $H_{0,\langle i \rangle}$. We further present some useful lemmas as follows.

**Lemma 18.** *[67] We have $E[h_{\langle i \rangle}] = p_*$ for $i \in [s]$ and $h_{\langle 1 \rangle}, h_{\langle 2 \rangle}, \cdots, h_{\langle s \rangle}$ are mutually independent, if $\boldsymbol{\mu}_{\hat{\mathbb{P}}_{\mathcal{B}_{\langle i \rangle}}} = \boldsymbol{\mu}_{\hat{\mathbb{Q}}_{\mathcal{B}_{\langle i \rangle}}}$ for every $i \in [s]$ and the p-values are uniformly distributed.*

Denote by $\widetilde{\mathcal{B}}$ the set of rectangle regions that the local two samples $\hat{X}'_{\mathcal{B}_{\langle i \rangle}}$ and $\hat{Y}'_{\mathcal{B}_{\langle i \rangle}}$ are actually drawn from one identical distribution, and we define

$$\mathcal{H}_0 = \left\{ H_{0,\langle i \rangle} \colon \mathcal{B}_{\langle i \rangle} \in \widetilde{\mathcal{B}} \right\} \, .$$

**Lemma 19.** *[67] If there is some rectangle region such that $\boldsymbol{\mu}_{\hat{\mathbb{P}}_{\mathcal{B}_{\langle \cdot \rangle}}} \neq \boldsymbol{\mu}_{\hat{\mathbb{Q}}_{\mathcal{B}_{\langle \cdot \rangle}}}$, and if the p-value follows a uniform distribution under null hypothesis $H_{0,\langle \cdot \rangle}$, then we have*

$$E\left[ h_{\langle i \rangle} \mid \{h_{\langle k \rangle}\}_{k=i+1}^s, \left\{\mathbb{I}\left(H_{0,\langle k \rangle} \in \mathcal{H}_0\right)\right\}_{k=i+1}^s, H_{0,\langle i \rangle} \in \mathcal{H}_0 \right] = p_* \quad \text{for} \quad i \in [s] \, .$$

We also define the *weighted mirror-conservativeness*, motivated from [85, 67], as follows.

**Definition 20.** We say that a density function $f(\cdot)$ satisfies the *weighted mirror-conservativeness* if it holds that, for some given $p \in (0, 1]$,

$$f(aw) \leq f(1 - (1 - wp)a/p) \text{ for every } w \in [1, 2], \ a \in [0, p] \, .$$

Here, we introduce an additional parameter $w$ to incorporate two directions for our method, which is different from the previous mirror-conservativeness [85, 67]. We could also present two sufficient conditions for weighted mirror-conservativeness from [85, 67]: i) the non-decrease of $f$ and ii) the convexity of cumulative density function of $p$-value.

**Lemma 21.** *If the density function of the p-value satisfies the weighted mirror-conservativeness under local null hypothesis $H_{0,\langle \cdot \rangle}$, and if there is some rectangle region such that $\boldsymbol{\mu}_{\hat{\mathbb{P}}_{\mathcal{B}_{\langle \cdot \rangle}}} \neq \boldsymbol{\mu}_{\hat{\mathbb{Q}}_{\mathcal{B}_{\langle \cdot \rangle}}}$, then we have, for every $i \in [s]$,*

$$E\left[ h_{\langle i \rangle} \mid \{h_{\langle k \rangle}\}_{k=i+1}^s, \left\{\mathbb{I}(h_{\langle k \rangle} \in \mathcal{H}_0)\right\}_{k=i+1}^s, H_{0,\langle i \rangle} \in \mathcal{H}_0, \{g_{\langle k \rangle}\}_{k=1}^s \right] \leq p_* \, , \tag{27}$$

*where $g_{\langle k \rangle} = g(\hat{X}'_{\mathcal{B}_{\langle k \rangle}}, \hat{Y}'_{\mathcal{B}_{\langle k \rangle}})$.*

*Proof.* We first prove

$$E[h_{\langle i \rangle} \mid g_{\langle i \rangle} = a] \leq p_* \quad \text{for} \quad H_{0,\langle i \rangle} \in \mathcal{H}_0 \, , \tag{28}$$

from our bi-directional hypothesis and masked $p$-value with $\beta \in [1, 2]$ in Eqn. (7), and it is sufficient to consider the following two cases:

- For $\boldsymbol{F}_{\mathcal{B}_{\langle i\rangle}}^{\top}\boldsymbol{L}_{\mathcal{B}_{\langle i\rangle}}^{\prime -1}(\boldsymbol{c}_{\hat{X}_{\mathcal{B}_{\langle i\rangle}}^{\prime}}-\boldsymbol{c}_{\hat{Y}_{\mathcal{B}_{\langle i\rangle}}^{\prime}})\geq 0$, we have

$$E\left[h_{\langle i\rangle}\mid g_{\langle i\rangle}=a,\boldsymbol{F}_{\mathcal{B}_{\langle i\rangle}}^{\top}\boldsymbol{L}_{\mathcal{B}_{\langle i\rangle}}^{\prime -1}(\boldsymbol{c}_{\hat{X}_{\mathcal{B}_{\langle i\rangle}}^{\prime}}-\boldsymbol{c}_{\hat{Y}_{\mathcal{B}_{\langle i\rangle}}^{\prime}})>0\right]$$

$$=\frac{p_*f(\beta a)}{p_*f(\beta a)+(1-p_*)f(1-\frac{1-\beta p_*}{p_*}a)}$$

$$=\frac{p_*}{p_*+(1-p_*)f(1-\frac{1-\beta p_*}{p_*}a)/f(\beta a)}$$

$$\leq\quad p_*\,,$$

where the last inequality holds from the weighted mirror-conservativeness with $p$-value's uniform distribution $\mathcal{U}[0,1]$.

- For $\boldsymbol{F}_{\mathcal{B}_{\langle i\rangle}}^{\top}\boldsymbol{L}_{\mathcal{B}_{\langle i\rangle}}^{\prime -1}(\boldsymbol{c}_{\hat{X}_{\mathcal{B}_{\langle i\rangle}}^{\prime}}-\boldsymbol{c}_{\hat{Y}_{\mathcal{B}_{\langle i\rangle}}^{\prime}})<0$, we similarly have

$$E\left[h_{\langle i\rangle}\mid g_{\langle i\rangle}=a,\boldsymbol{F}_{\mathcal{B}_{\langle i\rangle}}^{\top}\boldsymbol{L}_{\mathcal{B}_{\langle i\rangle}}^{\prime -1}(\boldsymbol{c}_{\hat{X}_{\mathcal{B}_{\langle i\rangle}}^{\prime}}-\boldsymbol{c}_{\hat{Y}_{\mathcal{B}_{\langle i\rangle}}^{\prime}})<0\right]$$

$$=\frac{p_*f((2-\beta)a)}{p_*f((2-\beta)a)+(1-p_*)f(1-\frac{1-(2-\beta)p_*}{p_*}a)}$$

$$=\frac{p_*}{p_*+(1-p_*)f(1-\frac{1-(2-\beta)p_*}{p_*}a)/f((2-\beta)a)}$$

$$\leq\quad p_*\,.$$

We define the information available for choosing $H_{0,\langle i\rangle}$ as a filtration (sequence of nested $\sigma$-fields)

$$\mathcal{F}_{\langle i\rangle}=\sigma\left(\{\mathcal{B}_{\langle k\rangle},g_{\langle k\rangle}\}_{k=1}^{s},\{\chi_\ell^2(\mathcal{T}(\hat{X}_{\mathcal{B}_{\langle k\rangle}}^{\prime},\hat{Y}_{\mathcal{B}_{\langle k\rangle}}^{\prime}))\}_{k=i+1}^{s}\right),$$

and also define the filtration

$$\mathcal{F}_{\langle i\rangle}^{h}=\sigma\left(\{h_{\langle k\rangle}\}_{k=i+1}^{s},\{\mathbb{I}(H_{0,\langle k\rangle}\in\mathcal{H}_0)\}_{k=i+1}^{s}\right).$$

This follows that

$$E\left[h_{\langle i\rangle}\mid\{h_{\langle k\rangle}\}_{k=i+1}^{s},\{\mathbb{I}(h_{\langle k\rangle}\in\mathcal{H}_0)\}_{k=i+1}^{s},H_{0,\langle i\rangle}\in\mathcal{H}_0,\{g_{\langle k\rangle}\}_{k=1}^{s}\right]$$

$$=\quad E\left[h_{\langle i\rangle}\mid\mathcal{F}_{\langle i\rangle}^{h},H_{0,\langle i\rangle}\in\mathcal{H}_0,\{g_{\langle k\rangle}\}_{k=1}^{s}\right]$$

$$=\quad E\left[E\left[h_{\langle i\rangle}\mid\mathcal{F}_{\langle i\rangle},\mathcal{F}_{\langle i\rangle}^{h},H_{0,\langle i\rangle}\in\mathcal{H}_0,\{g_{\langle k\rangle}\}_{k=1}^{s}\right]\Big|\mathcal{F}_{\langle i\rangle}^{h},H_{0,\langle i\rangle}\in\mathcal{H}_0,\{g_{\langle k\rangle}\}_{k=1}^{s}\right]$$

where the last equality holds from the law of total expectation. We have

$$E\left[h_{\langle i\rangle}\mid\mathcal{F}_{\langle i\rangle},\mathcal{F}_{\langle i\rangle}^{h},H_{0,\langle i\rangle}\in\mathcal{H}_0,\{g_{\langle k\rangle}\}_{k=1}^{s}\right]$$

$$=\quad\sum_{H_{0,j}\in\mathcal{R}(i)\cap\mathcal{H}_0}E[h_{\langle i\rangle}\mid\mathcal{F}_{\langle i\rangle},\mathcal{F}_{\langle i\rangle}^{h},H_{0,\langle i\rangle}\in\mathcal{H}_0,\{g_{\langle k\rangle}\}_{k=1}^{s}]$$

$$\times\Pr\left[H_{0,\langle i\rangle}=H_{0,j}\mid\mathcal{F}_{\langle i\rangle},\mathcal{F}_{\langle i\rangle}^{h},H_{0,\langle i\rangle}\in\mathcal{H}_0,\{g_{\langle k\rangle}\}_{k=1}^{s}\right]$$

$$=\quad\sum_{H_{0,j}\in\mathcal{R}(i)\cap\mathcal{H}_0}E\left[h_{\langle i\rangle}\mid\mathcal{F}_{\langle i\rangle}\right]\Pr\left[H_{0,\langle i\rangle}=H_{0,j}\mid\mathcal{F}_{\langle i\rangle},\mathcal{F}_{\langle i\rangle}^{h},H_{0,\langle i\rangle}\in\mathcal{H}_0,\{g_{\langle k\rangle}\}_{k=1}^{s}\right],$$

where the last equation holds from the fact that $\{\mathcal{F}_{\langle i\rangle}^{h},H_{0,\langle i\rangle}\in\mathcal{H}_0,\{g_{\langle k\rangle}\}_{k=1}^{s}\}$ is a subset of $\mathcal{F}_{\langle i\rangle}$. We further have, since $h_{\langle i\rangle}$ is independent of other information in $\mathcal{F}_{\langle i\rangle}$,

$$E\left[h_{\langle i\rangle}\mid\mathcal{F}_{\langle i\rangle},\mathcal{F}_{\langle i\rangle}^{h},H_{0,\langle i\rangle}\in\mathcal{H}_0,\{g_{\langle k\rangle}\}_{k=1}^{s}\right]$$

$$=\quad\sum_{H_{0,j}\in\mathcal{R}(i)\cap\mathcal{H}_0}E\left[h_{\langle i\rangle}\mid g_{\langle i\rangle}\right]\Pr\left[H_{0,\langle i\rangle}=H_{0,j}\mid\mathcal{F}_{\langle i\rangle},\mathcal{F}_{\langle i\rangle}^{h},H_{0,\langle i\rangle}\in\mathcal{H}_0,\{g_{\langle k\rangle}\}_{k=1}^{s}\right]$$

$$\leq\quad p_*\sum_{H_{0,j}\in\mathcal{R}(i)\cap\mathcal{H}_0}\Pr\left[H_{0,\langle i\rangle}=H_{0,j}\mid\mathcal{F}_{\langle i\rangle},\mathcal{F}_{\langle i\rangle}^{h},H_{0,\langle i\rangle}\in\mathcal{H}_0,\{g_{\langle k\rangle}\}_{k=1}^{s}\right]$$

$$=\quad p_*\,,$$

which completes the proof. $\qquad\square$

We say that a random variable $Z$ follows a Bernoulli distribution with parameter $p$, denoted by $Z \sim \mathcal{B}ern(p)$, if
$$\Pr[Z = 1] = p \quad \text{and} \quad \Pr[Z = 0] = 1 - p .$$

We also say that a random variable $Z$ follows a negative binomial distribution with parameters $r$ and $p$, denoted by $Z \sim \mathcal{NB}(r, p)$, if
$$\Pr[Z = k] = \binom{k + r - 1}{k}(1 - p)^r p^k .$$

It is necessary to introduce a definition as follows:

**Definition 22.** We say that a random variable $Z$ is stochastically dominated by a distribution $\mathbb{G}$, denoted by
$$Z \preceq \mathbb{G} ,$$
if for random variable $X \sim \mathbb{G}$, it holds that
$$\Pr[Z \geq x] \leq \Pr[X \geq x] \quad \text{for} \quad x \in (-\infty, +\infty).$$

We further introduce some useful lemmas as follows:

**Lemma 23.** *[67] Let $Z_1, \cdots, Z_s$ be i.i.d random variables with $Z_i \sim \mathcal{B}ern(p_*)$ for some $p_* > 0$, and write $\widetilde{N}_t = \sum_{j=1}^{t} Z_j$ and $\widetilde{\mathcal{G}}_t = \sigma(\widetilde{N}_t, \{Z_j\}_{j=t+1}^{s})$ for $t \in [s]$. We have*
$$\widetilde{N}_{\widetilde{\tau}} \preceq \mathcal{NB}(v, p_*) ,$$
*where the stopping index $\widetilde{\tau}$ is parameterized by some constant $v(\geq 1)$, defined by*
$$\widetilde{\tau} = \max\left\{0 < t \leq s : t - \widetilde{N}_t < v \text{ or } t = 1\right\} .$$

We further introduce a weighted version of Lemma 23 as follows:

**Lemma 24.** *[67] Let $\{W_j\}_{j=1}^{s}$ be a sequence of weights, drawn from a Bernoulli distribution, s.t. $\sum_{j=1}^{s} W_j = u$ for fixed constant $u \leq s$; and $Z_j \mid \sigma(\{Z_k, W_k\}_{k=j+1}^{s}, W_j = 1) \sim \mathcal{B}ern(p_*)$. Write $N_t^w = \sum_{j=1}^{t} W_j Z_j$, and we have*
$$N_{\tau^w}^w \preceq \mathcal{NB}(v, p_*) ,$$
*where the stopping index $\tau^w$ is parameterized by some constant $v(\geq 1)$, defined by*
$$\tau^w = \max\left\{0 < t \leq s : \sum_{j=1}^{t} W_j - N_t^w < v \text{ or } t = 1\right\} .$$

We now introduce a different version of Lemma 24 by considering different parameter for Bernoulli distribution as follows:

**Lemma 25.** *[67] Let $Z_j \mid \sigma(\{Z_k, W_k\}_{k=j+1}^{s}, W_j = 1)$ follow a Bernoulli distribution with parameter $p(\{Z_k, W_k\}_{k=j+1}^{s})$ for $j \in [s]$, respectively. We have*
$$N_{\tau^w}^w \preceq \mathcal{NB}\left(v, p(\{Z_k, W_k\}_{k=j+1}^{s})\right) \preceq \mathcal{NB}(v, p_*) ,$$
*if $p(\{Z_k, W_k\}_{k=j+1}^{s}) \leq p_*$ for every $j \in [s]$.*

We now present the detailed proof of Theorem 5 as follows.

**Proof of Theorem 5.** We first consider $\boldsymbol{\mu}_{\hat{\mathbb{P}}_{\mathcal{B}_{\langle i\rangle}}} = \boldsymbol{\mu}_{\hat{\mathbb{Q}}_{\mathcal{B}_{\langle i\rangle}}}$ for every $i \in [s]$. From Lemma 18, $\{h_{\langle i\rangle}\}_{i=1}^{s}$ are $s$ i.i.d. random variables with $h_{\langle i\rangle} \sim \mathcal{B}ern(p_*)$. Recall that the stopping rule in our testing, i.e., Eqn. (11), which is equivalent to
$$1 - (1 - p_*)^{t - |\mathcal{I}(t)| + 1} \leq \alpha_* ,$$

where $\alpha_*$ is a parameter to control familywise error rate and $\mathcal{I}(t) = \{i \in [t] : h(\hat{X}'_{\mathcal{B}_{\langle i \rangle}}, \hat{Y}'_{\mathcal{B}_{\langle i \rangle}}) = 1\}$. The stopping rule can be rewritten as $t - |\mathcal{I}(t)| < v$ with

$$v = \lfloor \ln(1 - \alpha_*)/\ln(1 - p_*) \rfloor . \tag{29}$$

Let $Z_j = h_{\langle j \rangle}$ and $\widetilde{N}_t = \sum_{j=1}^{t} Z_j$. We define the stopping index $\widetilde{\tau}$ as follows

$$\widetilde{\tau} = \max \left\{ 0 < t \le s : t - \widetilde{N}_t < v \text{ or } t = 1 \right\} .$$

From Lemma 23, the number of rejections at the stopping index is given by

$$|\mathcal{I}(\widetilde{\tau})| = \sum_{j=1}^{\widetilde{\tau}} h_{\langle j \rangle} = \widetilde{N}_{\widetilde{\tau}} \preceq \mathcal{NB}(v, p_*) .$$

If $\boldsymbol{\mu}_{\hat{\mathbb{P}}_{\mathcal{B}_{\langle i \rangle}}} = \boldsymbol{\mu}_{\hat{\mathbb{Q}}_{\mathcal{B}_{\langle i \rangle}}}$ for every $i \in [s]$, then the number of false rejections is

$$|\mathcal{I}(\widetilde{\tau}) \cap \mathcal{H}_0| = |\mathcal{I}(\widetilde{\tau})| \preceq \mathcal{NB}(v, p_*) ,$$

and hence the familywise error rate (FWER) is upper bounded by

$$\Pr\left[ |\mathcal{I}(\widetilde{\tau}) \cap \mathcal{H}_0| \ge 1 \right] \le 1 - (1 - p_*)^v \le \alpha_* ,$$

where the last inequality follows from Eqn. (29).

We now consider that there is some rectangle region with $\boldsymbol{\mu}_{\hat{\mathbb{P}}_{\mathcal{B}_{\langle \cdot \rangle}}} \ne \boldsymbol{\mu}_{\hat{\mathbb{Q}}_{\mathcal{B}_{\langle \cdot \rangle}}}$. In such case, we provide an upper bound for familywise error rate without the information of masked $p$-values, and prove that the number of false rejections is stochastically dominated by $\mathcal{NB}(v, p_*)$.

Let $Z_j = h_{\langle j \rangle}$ and $W_j = \mathbb{I}(H_{0, \langle j \rangle} \in \mathcal{H}_0)$. We define the stopping index $\tau^w$ as follows:

$$\tau^w = \max \left\{ 0 < t \le s : \sum_{j=1}^{t} \mathbb{I}[h_{\langle j \rangle} = 0 \cap H_{0, \langle j \rangle} \in \mathcal{H}_0] = \sum_{j=1}^{t} W_j(1 - Z_j) < v \text{ or } t = 1 \right\} ,$$

where $v$ is given in Eqn. (29). It is easy to see that

$$Z_j \mid \sigma\left( \{Z_k, W_k\}_{k=j+1}^{s}, W_j = 1 \right) \sim \mathcal{B}ern(p_*)$$

from Lemma 19. Denote by $u = |\mathcal{H}_0|$, and we have $\sum_{j=1}^{s} W_j = u$ and $u \le s$. From Lemma 24, we have the number of false rejections

$$\sum_{j=1}^{\tau^w} \mathbb{I}[h_{\langle j \rangle} = 1 \cap H_{0, \langle j \rangle} \in \mathcal{H}_0] = \sum_{j=1}^{\tau^w} W_j Z_j = N_{\tau^w}^w \preceq \mathcal{NB}(v, p_*) . \tag{30}$$

Recall that $t - |\mathcal{I}(t)| < v$ is our stopping rule on the exploration of local significant differences. Denote by $\tau_T^w$ the stopping index in our exploration, and we have

$$\sum_{j=1}^{\tau_T^w} \mathbb{I}[h_{\langle j \rangle} = 0 \cap H_{0, \langle j \rangle} \in \mathcal{H}_0]$$

$$\le \sum_{j=1}^{\tau_T^w} \mathbb{I}[h_{\langle j \rangle} = 0] = \tau_T^w - \sum_{j=1}^{\tau_T^w} \mathbb{I}[h_{\langle j \rangle} = 1] = \tau_T^w - \mathcal{I}(\tau_T^w) < v .$$

Since $N_t^w$ is non-decreasing with respect to $t$, it is easy to obtain

$$\tau_T^w \le \tau^w \quad \text{and} \quad N_{\tau_T^w}^w \le N_{\tau^w}^w ,$$

and we have the number of false rejections

$$|\mathcal{I}(\tau_T^w) \cap \mathcal{H}_0| = \sum_{j=1}^{\tau_T^w} \mathbb{I}[h_{\langle j \rangle} = 1 \cap H_{0, \langle j \rangle} \in \mathcal{H}_0] = N_{\tau_T^w}^w \le N_{\tau^w}^w \preceq \mathcal{NB}(v, p_*) . \tag{31}$$

We upper bound the familywise error rate without considering the masked $p$-values as follows:

$$\Pr\left[|\mathcal{I}(\tau_T^w) \cap \mathcal{H}_0| \geq 1\right] \leq \Pr\left[|\mathcal{I}(\tau^w) \cap \mathcal{H}_0| \geq 1\right] \leq 1 - (1 - p_*)^v \leq \alpha_* \ .$$

We finally take masked $p$-values $\{g_{\langle k \rangle}\}_{k=1}^s$ into consideration. From Lemma 21, it is easy to observe

$$Z_j \mid \sigma\left(\{Z_k, W_k\}_{k=j+1}^s, W_j = 1\right) \ \sim \ \mathcal{B}ern(p(\{Z_k, W_k\}_{k=j+1}^s)) \ ,$$

where

$$p(\{Z_k, W_k\}_{k=j+1}^s) = E\left[h_{\langle j \rangle} \mid \{h_{\langle k \rangle}\}_{k=j+1}^s, \{\mathbb{I}(h_{\langle k \rangle} \in \mathcal{H}_0)\}_{k=j+1}^s, H_{0,\langle j \rangle} \in \mathcal{H}_0, \{g_{\langle k \rangle}\}_{k=1}^s\right] \leq p_* \ .$$

This follows that

$$\mathcal{NB}\left(v, p\left(\{Z_k, W_k\}_{k=j+1}^s\right)\right) \preceq \mathcal{NB}(v, p_*) \ ,$$

and we further have, from Lemma 25,

$$|\mathcal{I}(\tau_T^w) \cap \mathcal{H}_0| = \sum_{j=1}^{\tau^w} W_j Z_j = N_{\tau^w}^w \preceq \mathcal{NB}\left(v, p\left(\{Z_k, W_k\}_{k=j+1}^s\right)\right) \preceq \mathcal{NB}(v, p_*) \ .$$

We finally upper bound the familywise error rate by considering the masked $p$-values $\left\{g_{\langle k \rangle}\right\}_{k=1}^s$ as

$$\begin{aligned}
\Pr\left[|\mathcal{I}(\tau_T^w) \cap \mathcal{H}_0| \geq 1 \mid \{g_{\langle k \rangle}\}_{k=1}^s\right] &\leq \ \mathbb{P}\left(|\mathcal{I}(\tau^w) \cap \mathcal{H}_0| \geq 1 \mid \{g_{\langle k \rangle}\}_{k=1}^s\right) \\
&\leq \ 1 - (1 - p_*)^v \leq \alpha_* \ .
\end{aligned}$$

This completes the proof. $\qquad\qquad\qquad\qquad\qquad\qquad\qquad\qquad\qquad\qquad\qquad\qquad\square$

## B  Optimization for Test Locations and Mahalanobis Kernels

We take gradient method [43] for the optimization of Eqn. (5) as in the work of [10]. Specifically, we calculate gradients, and update test locations and Mahalanobis kernels iteratively. In the following of this section, we present the calculation of some crucial gradients in optimization.

For test location $\boldsymbol{v}_j$ with $j \in [\ell]$, we have

$$\nabla_{\boldsymbol{v}_j} \mathcal{T}(\hat{X}, \hat{Y}) = \left(\frac{\partial \mathcal{T}(\hat{X}, \hat{Y})}{\partial \boldsymbol{v}_{j,1}}, \frac{\partial \mathcal{T}(\hat{X}, \hat{Y})}{\partial \boldsymbol{v}_{j,2}}, \ldots, \frac{\partial \mathcal{T}(\hat{X}, \hat{Y})}{\partial \boldsymbol{v}_{j,\ell}}\right)^\top \ , \tag{32}$$

where, for $i \in [\ell]$,

$$\frac{\partial \mathcal{T}(\hat{X}, \hat{Y})}{\partial \boldsymbol{v}_{j,i}} = \frac{\partial \mathcal{T}(\hat{X}, \hat{Y})}{\partial \boldsymbol{c}_{\hat{X}}} \frac{\partial \boldsymbol{c}_{\hat{X}}}{\partial \boldsymbol{v}_{j,i}} + \frac{\partial \mathcal{T}(\hat{X}, \hat{Y})}{\partial \boldsymbol{c}_{\hat{Y}}} \frac{\partial \boldsymbol{c}_{\hat{Y}}}{\partial \boldsymbol{v}_{j,i}} + \text{Tr}\left[\frac{\partial \mathcal{T}(\hat{X}, \hat{Y})}{\partial \Sigma_{\hat{X}, \hat{Y}}} \frac{\partial \Sigma_{\hat{X}, \hat{Y}}}{\partial \boldsymbol{v}_{j,i}}\right] \ , \tag{33}$$

where $\text{Tr}[\cdot]$ denotes the trace.

We further have

$$\begin{aligned}
\frac{\partial \mathcal{T}(\hat{X}, \hat{Y})}{\partial \boldsymbol{c}_{\hat{X}}} &= \ 2mn\Sigma_{\hat{X}, \hat{Y}}^{-1}(\boldsymbol{c}_{\hat{X}} - \boldsymbol{c}_{\hat{Y}})/(m + n) \ , \\
\frac{\partial \boldsymbol{c}_{\hat{X}}}{\partial \boldsymbol{v}_{j,i}} &= \ \frac{1}{m} \sum_{r=1}^m \frac{\partial \hat{\boldsymbol{x}}_r}{\partial \boldsymbol{v}_{j,i}} \quad \text{with} \quad \frac{\partial \hat{\boldsymbol{x}}_r}{\partial \boldsymbol{v}_{j,i}} = \left(0, \cdots, 0, \frac{\partial \kappa_j(\hat{\boldsymbol{x}}_r, \boldsymbol{v}_j)}{\partial \boldsymbol{v}_{j,i}}, 0, \cdots, 0\right)^\top \ ,
\end{aligned}$$

where all elements are zeros except for the $j$-th element. We also have

$$\begin{aligned}
\frac{\partial \kappa_j(\boldsymbol{x}_r, \boldsymbol{v}_j)}{\partial \boldsymbol{v}_{j,i}} &= \ \frac{\partial}{\partial \boldsymbol{v}_{j,i}} \left\{\exp\left(-(\boldsymbol{x}_r - \boldsymbol{v}_j)^\top M_j(\boldsymbol{x}_r - \boldsymbol{v}_j)/2\gamma_j^2\right)\right\} \\
&= \ \kappa_j(\hat{\boldsymbol{x}}_r, \boldsymbol{v}_j)(M_j(\hat{\boldsymbol{x}}_r - \boldsymbol{v}_j))^\top (0, \ldots, 1, \ldots, 0)^\top / \gamma_j^2 \ ,
\end{aligned}$$

where the $i$-th element is 1. We similarly calculate $\partial \mathcal{T}(\hat{X}, \hat{Y})/\partial \boldsymbol{c}_{\hat{Y}} \times \partial \boldsymbol{c}_{\hat{Y}}/\partial \boldsymbol{v}_{j,i}$.

For the third term in Eqn. (33), we have

$$\frac{\partial \mathcal{T}(\hat{X}, \hat{Y})}{\partial \Sigma_{\hat{X}, \hat{Y}}} = \Sigma_{\hat{X}, \hat{Y}}^{-1}(\boldsymbol{c}_{\hat{X}} - \boldsymbol{c}_{\hat{Y}})(\boldsymbol{c}_{\hat{X}} - \boldsymbol{c}_{\hat{Y}})^{\top}\Sigma_{\hat{X}, \hat{Y}}^{-1},$$

$$\frac{\partial \Sigma_{\hat{X}, \hat{Y}}}{\partial \boldsymbol{v}_{j,i}} = \frac{1}{m+n-2}\left(\sum_{r=1}^{m}\frac{\partial(\hat{\boldsymbol{x}}_r - \boldsymbol{c}_{\hat{X}})(\hat{\boldsymbol{x}}_r - \boldsymbol{c}_{\hat{X}})^{\top}}{\partial \boldsymbol{v}_{j,i}} + \sum_{r=1}^{n}\frac{\partial(\hat{\boldsymbol{y}}_r - \boldsymbol{c}_{\hat{Y}})(\hat{\boldsymbol{y}}_r - \boldsymbol{c}_{\hat{Y}})^{\top}}{\partial \boldsymbol{v}_{j,i}}\right),$$

where

$$\frac{\partial(\hat{\boldsymbol{x}}_r - \boldsymbol{c}_{\hat{X}})(\hat{\boldsymbol{x}}_r - \boldsymbol{c}_{\hat{X}})^{\top}}{\partial \boldsymbol{v}_{j,i}}$$

$$= \begin{pmatrix} 0 & \cdots & \frac{\partial(\hat{\boldsymbol{x}}_r - \boldsymbol{c}_{\hat{X}})(\hat{\boldsymbol{x}}_r - \boldsymbol{c}_{\hat{X}})_{1,j}^{\top}}{\partial \boldsymbol{v}_{j,i}} & \cdots & 0 \\ \vdots & \ddots & \vdots & \ddots & \vdots \\ \frac{\partial(\hat{\boldsymbol{x}}_r - \boldsymbol{c}_{\hat{X}})(\hat{\boldsymbol{x}}_r - \boldsymbol{c}_{\hat{X}})_{j,1}^{\top}}{\partial \boldsymbol{v}_{j,i}} & \cdots & \frac{\partial(\hat{\boldsymbol{x}}_r - \boldsymbol{c}_{\hat{X}})(\hat{\boldsymbol{x}}_r - \boldsymbol{c}_{\hat{X}})_{j,j}^{\top}}{\partial \boldsymbol{v}_{j,i}} & \cdots & \frac{\partial(\hat{\boldsymbol{x}}_r - \boldsymbol{c}_{\hat{X}})(\hat{\boldsymbol{x}}_r - \boldsymbol{c}_{\hat{X}})_{j,\ell}^{\top}}{\partial \boldsymbol{v}_{j,i}} \\ \vdots & \ddots & \vdots & \ddots & \vdots \\ 0 & \cdots & \frac{\partial(\hat{\boldsymbol{x}}_r - \boldsymbol{c}_{\hat{X}})(\hat{\boldsymbol{x}}_r - \boldsymbol{c}_{\hat{X}})_{\ell,j}^{\top}}{\partial \boldsymbol{v}_{j,i}} & \cdots & 0 \end{pmatrix}.$$

Here, $(\hat{\boldsymbol{x}}_r - \boldsymbol{c}_{\hat{X}})(\hat{\boldsymbol{x}}_r - \boldsymbol{c}_{\hat{X}})_{j,t}^{\top}$ denotes the element in $j$-th row and $t$-th column, and we have

$$(\hat{\boldsymbol{x}}_r - \boldsymbol{c}_{\hat{X}})(\hat{\boldsymbol{x}}_r - \boldsymbol{c}_{\hat{X}})_{j,t}^{\top}$$

$$= \left(\frac{m-1}{m}\kappa_t(\boldsymbol{x}_r, \boldsymbol{v}_j) - \frac{1}{m}\sum_{s\neq r}^{m}\kappa_t(\boldsymbol{x}_s, \boldsymbol{v}_j)\right)\left(\frac{m-1}{m}\kappa_t(\boldsymbol{x}_r, \boldsymbol{v}_t) - \frac{1}{m}\sum_{s\neq r}^{m}\kappa_t(\boldsymbol{x}_s, \boldsymbol{v}_t)\right),$$

and this follows that

$$\frac{\partial(\hat{\boldsymbol{x}}_r - \boldsymbol{c}_{\hat{X}})(\hat{\boldsymbol{x}}_r - \boldsymbol{c}_{\hat{X}})_{j,t}^{\top}}{\partial \boldsymbol{v}_{j,i}}$$

$$= \left(\frac{m-1}{m}\kappa_j(\boldsymbol{x}_r, \boldsymbol{v}_t) - \frac{1}{m}\sum_{s\neq r}^{m}\kappa_j(\boldsymbol{x}_s, \boldsymbol{v}_t)\right)\left(\frac{m-1}{m}\frac{\partial\kappa_j(\boldsymbol{x}_r, \boldsymbol{v}_j)}{\partial \boldsymbol{v}_{j,i}} - \frac{1}{m}\sum_{s\neq r}^{m}\frac{\partial\kappa_j(\boldsymbol{x}_s, \boldsymbol{v}_j)}{\partial \boldsymbol{v}_{j,i}}\right).$$

We similarly have

$$\frac{\partial(\hat{\boldsymbol{x}}_r - \boldsymbol{c}_{\hat{X}})(\hat{\boldsymbol{x}}_r - \boldsymbol{c}_{\hat{X}})_{j,j}^{\top}}{\partial \boldsymbol{v}_{j,i}}$$

$$= 2\left(\frac{m-1}{m}\kappa_j(\boldsymbol{x}_r, \boldsymbol{v}_j) - \frac{1}{m}\sum_{s\neq r}^{m}\kappa_j(\boldsymbol{x}_s, \boldsymbol{v}_j)\right)\left(\frac{m-1}{m}\frac{\partial\kappa_j(\boldsymbol{x}_r, \boldsymbol{v}_j)}{\partial \boldsymbol{v}_{j,i}} - \frac{1}{m}\sum_{s=r}^{m}\frac{\partial\kappa_j(\boldsymbol{x}_s, \boldsymbol{v}_j)}{\partial \boldsymbol{v}_{j,i}}\right).$$

For gamma parameter $\gamma_j$ with $j \in [\ell]$, we have

$$\nabla_{\gamma_j}\mathcal{T}(\hat{X}, \hat{Y}) = \frac{\partial \mathcal{T}(\hat{X}, \hat{Y})}{\partial \boldsymbol{c}_{\hat{X}}}\frac{\partial \boldsymbol{c}_{\hat{X}}}{\partial \gamma_j} + \frac{\partial \mathcal{T}(\hat{X}, \hat{Y})}{\partial \boldsymbol{c}_{\hat{Y}}}\frac{\partial \boldsymbol{c}_{\hat{Y}}}{\partial \gamma_j} + \text{Tr}\left[\frac{\partial \mathcal{T}(\hat{X}, \hat{Y})}{\partial \Sigma_{\hat{X}, \hat{Y}}}\frac{\partial \Sigma_{\hat{X}, \hat{Y}}}{\partial \gamma_j}\right]. \qquad (34)$$

We further have

$$\frac{\partial \boldsymbol{c}_{\hat{X}}}{\partial \gamma_j} = \frac{1}{m}\sum_{r=1}^{m}\frac{\partial \hat{\boldsymbol{x}}_r}{\partial \gamma_j} \quad \text{with} \quad \frac{\partial \hat{\boldsymbol{x}}_r}{\partial \gamma_j} = \left(0, \ldots, \frac{\partial\kappa_j(\hat{\boldsymbol{x}}_r, \boldsymbol{v}_j)}{\partial \gamma_j}, \ldots, 0\right)^{\top},$$

where

$$\frac{\partial\kappa_j(\boldsymbol{x}_r, \boldsymbol{v}_j)}{\partial \gamma_j} = \frac{\partial \exp\left(-(\boldsymbol{x}_r - \boldsymbol{v}_j)^{\top}M_j(\boldsymbol{x}_r - \boldsymbol{v}_j)/2\gamma_j^2\right)}{\partial \gamma_j}$$

$$= \kappa_j(\hat{\boldsymbol{x}}_r, \boldsymbol{v}_j)(\boldsymbol{x}_r - \boldsymbol{v}_j)^{\top}M_j(\boldsymbol{x}_r - \boldsymbol{v}_j)\gamma_j^{-3}.$$

We similarly calculate

$$\frac{\partial \Sigma_{\hat{X},\hat{Y}}}{\partial \gamma_j} = \frac{1}{m+n-2} \left( \sum_{r=1}^{m} \frac{\partial(\hat{\boldsymbol{x}}_r - \boldsymbol{c}_{\hat{X}})(\hat{\boldsymbol{x}}_r - \boldsymbol{c}_{\hat{X}})^\top}{\partial \gamma_j} + \sum_{r=1}^{n} \frac{\partial(\hat{\boldsymbol{y}}_r - \boldsymbol{c}_{\hat{Y}})(\hat{\boldsymbol{y}}_r - \boldsymbol{c}_{\hat{Y}})^\top}{\partial \gamma_j} \right),$$

where

$$\frac{\partial(\hat{\boldsymbol{x}}_r - \boldsymbol{c}_{\hat{X}})(\hat{\boldsymbol{x}}_r - \boldsymbol{c}_{\hat{X}})^\top}{\partial \gamma_j}$$

$$= \begin{pmatrix} 0 & \cdots & \frac{\partial(\hat{\boldsymbol{x}}_r - \boldsymbol{c}_{\hat{X}})(\hat{\boldsymbol{x}}_r - \boldsymbol{c}_{\hat{X}})_{1,j}^\top}{\partial \gamma_j} & \cdots & 0 \\ \vdots & \ddots & \vdots & \ddots & \vdots \\ \frac{\partial(\hat{\boldsymbol{x}}_r - \boldsymbol{c}_{\hat{X}})(\hat{\boldsymbol{x}}_r - \boldsymbol{c}_{\hat{X}})_{j,1}^\top}{\partial \gamma_j} & \cdots & \frac{\partial(\hat{\boldsymbol{x}}_r - \boldsymbol{c}_{\hat{X}})(\hat{\boldsymbol{x}}_r - \boldsymbol{c}_{\hat{X}})_{j,j}^\top}{\partial \gamma_j} & \cdots & \frac{\partial(\hat{\boldsymbol{x}}_r - \boldsymbol{c}_{\hat{X}})(\hat{\boldsymbol{x}}_r - \boldsymbol{c}_{\hat{X}})_{j,\ell}^\top}{\partial \gamma_j} \\ \vdots & \ddots & \vdots & \ddots & \vdots \\ 0 & \cdots & \frac{\partial(\hat{\boldsymbol{x}}_r - \boldsymbol{c}_{\hat{X}})(\hat{\boldsymbol{x}}_r - \boldsymbol{c}_{\hat{X}})_{\ell,j}^\top}{\partial \gamma_j} & \cdots & 0 \end{pmatrix},$$

and this follows that

$$\frac{\partial(\hat{\boldsymbol{x}}_r - \boldsymbol{c}_{\hat{X}})(\hat{\boldsymbol{x}}_r - \boldsymbol{c}_{\hat{X}})_{j,t}^\top}{\partial \gamma_j}$$

$$= \left( \frac{m-1}{m} \kappa_j(\boldsymbol{x}_r, \boldsymbol{v}_t) - \frac{1}{m} \sum_{s \neq r}^{m} \kappa_j(\boldsymbol{x}_s, \boldsymbol{v}_t) \right) \left( \frac{m-1}{m} \frac{\partial \kappa_j(\boldsymbol{x}_r, \boldsymbol{v}_j)}{\partial \gamma_j} - \frac{1}{m} \sum_{s \neq r}^{m} \frac{\partial \kappa_j(\boldsymbol{x}_s, \boldsymbol{v}_j)}{\partial \gamma_j} \right).$$

We similarly have

$$\frac{\partial(\hat{\boldsymbol{x}}_r - \boldsymbol{c}_{\hat{X}})(\hat{\boldsymbol{x}}_r - \boldsymbol{c}_{\hat{X}})_{j,j}^\top}{\partial \gamma_j}$$

$$= 2 \left( \frac{m-1}{m} \kappa_j(\boldsymbol{x}_r, \boldsymbol{v}_j) - \frac{1}{m} \sum_{s \neq r}^{m} \kappa_j(\boldsymbol{x}_s, \boldsymbol{v}_j) \right) \left( \frac{m-1}{m} \frac{\partial \kappa_j(\boldsymbol{x}_r, \boldsymbol{v}_j)}{\partial \gamma_j} - \frac{1}{m} \sum_{s=r}^{m} \frac{\partial \kappa_j(\boldsymbol{x}_s, \boldsymbol{v}_j)}{\partial \gamma_j} \right).$$

For Mahalanobis matrix $M_j$ with $j \in [\ell]$, we have

$$\nabla_{M_j} \mathcal{T}(\hat{X}, \hat{Y}) = \begin{pmatrix} \frac{\partial \mathcal{T}(\hat{X}, \hat{Y})}{\partial M_{j,1,1}} & \cdots & \frac{\partial \mathcal{T}(\hat{X}, \hat{Y})}{\partial M_{j,1,\ell}} \\ \vdots & \ddots & \vdots \\ \frac{\partial \mathcal{T}(\hat{X}, \hat{Y})}{\partial M_{j,\ell,1}} & \cdots & \frac{\partial \mathcal{T}(\hat{X}, \hat{Y})}{\partial M_{j,\ell,\ell}} \end{pmatrix}, \tag{35}$$

where we denote by $M_{j,a,b}$ the element in $a$-th row and $b$-th column in $M_j$.

We further have

$$\frac{\partial \mathcal{T}(\hat{X}, \hat{Y})}{\partial M_{j,a,b}} = \frac{\partial \mathcal{T}(\hat{X}, \hat{Y})}{\partial \boldsymbol{c}_{\hat{X}}} \frac{\partial \boldsymbol{c}_{\hat{X}}}{\partial M_{j,a,b}} + \frac{\partial \mathcal{T}(\hat{X}, \hat{Y})}{\partial \boldsymbol{c}_{\hat{Y}}} \frac{\partial \boldsymbol{c}_{\hat{Y}}}{\partial M_{j,a,b}} + \mathrm{Tr}\left[ \frac{\partial \mathcal{T}(\hat{X}, \hat{Y})}{\partial \Sigma_{\hat{X},\hat{Y}}} \frac{\partial \Sigma_{\hat{X},\hat{Y}}}{\partial M_{j,a,b}} \right],$$

with

$$\frac{\partial \boldsymbol{c}_{\hat{X}}}{\partial M_{j,a,b}} = \frac{1}{m} \sum_{r=1}^{m} \frac{\partial \hat{\boldsymbol{x}}_r}{\partial M_{j,a,b}} \quad \text{and} \quad \frac{\partial \hat{\boldsymbol{x}}_r}{\partial M_{j,a,b}} = \left( 0, \dots, \frac{\partial \kappa_j(\hat{\boldsymbol{x}}_r, \boldsymbol{v}_j)}{\partial M_{j,a,b}}, \dots, 0 \right)^\top,$$

and

$$\begin{aligned} \frac{\partial \kappa_j(\boldsymbol{x}_r, \boldsymbol{v}_j)}{\partial M_{j,a,b}} &= \frac{\partial \exp\left(-(\boldsymbol{x}_r - \boldsymbol{v}_j)^\top M_j (\boldsymbol{x}_r - \boldsymbol{v}_j)/2\gamma_j^2\right)}{\partial M_{j,a,b}} \\ &= -\frac{\kappa_j(\boldsymbol{x}_r, \boldsymbol{v}_j)}{2\gamma_j^2} \mathrm{Tr}\left[ \frac{\partial(\boldsymbol{x}_r - \boldsymbol{v}_j)^\top M_j(\boldsymbol{x}_r - \boldsymbol{v}_j)}{\partial M_j} \frac{\partial M_j}{\partial M_{j,a,b}} \right] \\ &= -\frac{\kappa_j(\boldsymbol{x}_r, \boldsymbol{v}_j)}{2\gamma_j^2} \mathrm{Tr}\left[ (\boldsymbol{x}_r - \boldsymbol{v}_j)(\boldsymbol{x}_r - \boldsymbol{v}_j)^\top \boldsymbol{J}^{a,b} \right], \end{aligned}$$

where $\boldsymbol{J}^{a,b}$ is the single-entry matrix (1 at $(a,b)$ and zero elsewhere). We similarly calculate $\partial\boldsymbol{c}_{\hat{Y}}/\partial M_{j,a,b}$ and have

$$\frac{\partial\Sigma_{\hat{X},\hat{Y}}}{\partial M_{j,a,b}} = \frac{1}{m+n-2}\left(\sum_{r=1}^{m}\frac{\partial(\hat{\boldsymbol{x}}_r-\boldsymbol{c}_{\hat{X}})(\hat{\boldsymbol{x}}_r-\boldsymbol{c}_{\hat{X}})^\top}{\partial M_{j,a,b}} + \sum_{r=1}^{n}\frac{\partial(\hat{\boldsymbol{y}}_r-\boldsymbol{c}_{\hat{Y}})(\hat{\boldsymbol{y}}_r-\boldsymbol{c}_{\hat{Y}})^\top}{\partial M_{j,a,b}}\right),$$

where

$$\frac{\partial(\hat{\boldsymbol{x}}_r-\boldsymbol{c}_{\hat{X}})(\hat{\boldsymbol{x}}_r-\boldsymbol{c}_{\hat{X}})^\top}{\partial M_{j,a,b}}$$

$$= \begin{pmatrix} 0 & \cdots & \frac{\partial(\hat{\boldsymbol{x}}_r-\boldsymbol{c}_{\hat{X}})(\hat{\boldsymbol{x}}_r-\boldsymbol{c}_{\hat{X}})_{1,j}^\top}{\partial M_{j,a,b}} & \cdots & 0 \\ \vdots & \ddots & \vdots & \ddots & \vdots \\ \frac{\partial(\hat{\boldsymbol{x}}_r-\boldsymbol{c}_{\hat{X}})(\hat{\boldsymbol{x}}_r-\boldsymbol{c}_{\hat{X}})_{j,1}^\top}{\partial M_{j,a,b}} & \cdots & \frac{\partial(\hat{\boldsymbol{x}}_r-\boldsymbol{c}_{\hat{X}})(\hat{\boldsymbol{x}}_r-\boldsymbol{c}_{\hat{X}})_{j,j}^\top}{\partial M_{j,a,b}} & \cdots & \frac{\partial(\hat{\boldsymbol{x}}_r-\boldsymbol{c}_{\hat{X}})(\hat{\boldsymbol{x}}_r-\boldsymbol{c}_{\hat{X}})_{j,\ell}^\top}{\partial M_{j,a,b}} \\ \vdots & \ddots & \vdots & \ddots & \vdots \\ 0 & \cdots & \frac{\partial(\hat{\boldsymbol{x}}_r-\boldsymbol{c}_{\hat{X}})(\hat{\boldsymbol{x}}_r-\boldsymbol{c}_{\hat{X}})_{\ell,j}^\top}{\partial M_{j,a,b}} & \cdots & 0 \end{pmatrix}.$$

Here, we have

$$\frac{\partial(\hat{\boldsymbol{x}}_r-\boldsymbol{c}_{\hat{X}})(\hat{\boldsymbol{x}}_r-\boldsymbol{c}_{\hat{X}})_{j,t}^\top}{\partial M_{j,a,b}}$$

$$= \left(\frac{m-1}{m}\kappa_j(\boldsymbol{x}_r,\boldsymbol{v}_t) - \frac{1}{m}\sum_{s\neq r}^{m}\kappa_j(\boldsymbol{x}_s,\boldsymbol{v}_t)\right)\left(\frac{m-1}{m}\frac{\partial\kappa_j(\boldsymbol{x}_r,\boldsymbol{v}_j)}{\partial M_{j,a,b}} - \frac{1}{m}\sum_{s\neq r}^{m}\frac{\partial\kappa_j(\boldsymbol{x}_s,\boldsymbol{v}_j)}{\partial M_{j,a,b}}\right).$$

We similarly have $\partial(\hat{\boldsymbol{x}}_r-\boldsymbol{c}_{\hat{X}})(\hat{\boldsymbol{x}}_r-\boldsymbol{c}_{\hat{X}})_{t,j}^\top/\partial M_{j,a,b}$ and

$$\frac{\partial(\hat{\boldsymbol{x}}_r-\boldsymbol{c}_{\hat{X}})(\hat{\boldsymbol{x}}_r-\boldsymbol{c}_{\hat{X}})_{j,j}^\top}{\partial M_{j,a,b}}$$

$$= 2\left(\frac{m-1}{m}\kappa_j(\boldsymbol{x}_r,\boldsymbol{v}_j) - \frac{1}{m}\sum_{s\neq r}^{m}\kappa_j(\boldsymbol{x}_s,\boldsymbol{v}_j)\right)\left(\frac{m-1}{m}\frac{\partial\kappa_j(\boldsymbol{x}_r,\boldsymbol{v}_j)}{\partial M_{j,a,b}} - \frac{1}{m}\sum_{s=r}^{m}\frac{\partial\kappa_j(\boldsymbol{x}_s,\boldsymbol{v}_j)}{\partial M_{j,a,b}}\right).$$

**Project Mahalanobis matrix onto a positive definite cone**

We can not guarantee the positive-definiteness of Mahalanobis matrices during the optimization process via gradient ascend. Motivated from [43], we project Mahalanobis matrix onto a positive definite cone as follows:

- Present the spectral (eigenvalue) decomposition of a Mahalanobis matrix $M$ as

$$M = \sum_{i=1}^{d}\lambda_i\boldsymbol{p}_i\boldsymbol{p}_i^T.$$

  where $\lambda_1, \lambda_2, \cdots, \lambda_d$ are their eigenvalues with corresponding eigenvectors $\boldsymbol{p}_1, \boldsymbol{p}_2, \cdots, \boldsymbol{p}_d$.
- Project the Mahalanobis matrix $M$ onto a positive definite cone

$$M = \sum_{i=1}^{d}\max\{\lambda_i,\delta\}v_iv_i^T \quad\text{for small positive constant } \delta.$$

## C  Datasets and Parameter Setting

**Datasets**

We partition datasets into several disjoint subsets, and then randomly draw data elements from each subset based on the sample fraction, i.e., the proportion of samples to be selected from each

subset. We construct two different samples by using two different sample fractions in above stratified sampling process, and construct two samples drawn from one identical distribution by adapting a same sample fraction, as done in [60].

We provide the details of constructing two samples for each dataset as follows:

- blob is constructed as the mixture of nine Gaussian modes. We first write

$$u_1 = [0,0], \quad u_2 = [0,1], \quad u_3 = [0,2],$$
$$u_4 = [1,0], \quad u_5 = [1,1], \quad u_6 = [1,2],$$
$$u_7 = [2,0], \quad u_8 = [2,1], \quad u_9 = [2,2],$$

and

$$\Delta_i = \begin{cases} -0.02 - 0.002 \times (i-1) & \text{for} \quad i < 5 \\ 0 & \text{for} \quad i = 5 \\ 0.02 + 0.002 \times (i-6) & \text{for} \quad i > 5 \,. \end{cases}$$

To construct different distributions $\mathbb{P}$ and $\mathbb{Q}$, we adapt different covariance structures and

$$\mathbb{P} = \sum_{i=1}^{9} \frac{1}{9} \mathcal{N}\left(u_i, 0.03 \times \boldsymbol{I}_2\right) \quad \text{and} \quad \mathbb{Q} = \sum_{i=1}^{9} \frac{1}{9} \mathcal{N}\left(u_i, \begin{bmatrix} 0.03 & \Delta_i \\ \Delta_i & 0.03 \end{bmatrix}\right).$$

To construct identical distribution for two samples, i.e., $\mathbb{P} = \mathbb{Q}$, we have

$$\mathbb{P} = \sum_{i=1}^{9} \frac{1}{9} \mathcal{N}\left(u_i, 0.03 \times \boldsymbol{I}_2\right) \quad \text{and} \quad \mathbb{Q} = \sum_{i=1}^{9} \frac{1}{9} \mathcal{N}\left(u_i, 0.03 \times \boldsymbol{I}_2\right).$$

  We set the sample size for training to 900 and for testing to 224.

- dna is a categorical dataset with 3 classes. For constructing two different samples, we set the sample fraction for one sample to $[0.30, 0.35, 0.35]$ and for the other sample to $[0.45, 0.25, 0.3]$; To construct two samples with one identical distribution, we set the same sample fraction $[0.30, 0.35, 0.35]$ for two samples. We set the sample size for training to 1000 and for testing to 250.

- agnos (agnostic) is a categorical dataset with 2 classes. For constructing two different samples, we set the sample fraction for one sample to $[0.35, 0.65]$ and for the other sample to $[0.65, 0.35]$; To construct two samples with one identical distribution, we set the same sample fraction $[0.35, 0.65]$ for two samples. We set the sample size for training to 1000 and for testing to 250.

- topo21 is a regression dataset with continuous target variables. Based on the sorted target variables, we divide the data into 4 equal parts. For constructing two different samples, we set the sample fraction for one sample to $[0.1, 0.3, 0.2, 0.4]$ and for the other sample to $[0.5, 0.2, 0.1, 0.2]$; To construct two samples with one identical distribution, we set the same sample fraction $[0.1, 0.3, 0.2, 0.4]$ for two samples. We set the sample size for training to 2200 and for testing to 550.

- har is a categorical dataset with 6 classes. For constructing two different samples, we set the sample fraction for one sample to $[0.10, 0.20, 0.10, 0.20, 0.20, 0.10]$ and for the other sample to $[0.15, 0.15, 0.20, 0.15, 0.20, 0.15]$; To construct two samples with one identical distribution, we set the same sample fraction $[0.10, 0.20, 0.10, 0.20, 0.20, 0.10]$ for two samples. We set the sample size for training to 2200 and for testing to 550.

- kropt is a categorical dataset with 18 classes, where we only consider categories 13, 14, 15 and 16 which have the majority of the data.. For constructing two different samples, we set the sample fraction for one sample to $[0.15, 0.2, 0.3, 0.35]$ and for the other sample to $[0.35, 0.35, 0.15, 0.15]$; To construct two samples with one identical distribution, we set the same sample fraction $[0.25, 0.25, 0.25, 0.25]$ for two samples. We set the sample size for training to 2000 and for testing to 500.

- diamon (diamonds) is a regression dataset with continuous target variables. Based on the sorted target variables, we divide the data into 4 equal parts. For constructing two different samples, we set the sample fraction for one sample to $[0.35, 0.2, 0.2, 0.25]$ and for the other sample to $[0.2, 0.3, 0.3, 0.2]$; To construct two samples with one identical distribution, we set the same sample fraction $[0.25, 0.25, 0.25, 0.25]$ for two samples. We set the sample size for training to 2000 and for testing to 500.

**Table 4:** Optimization parameters of our ME$_{\text{MaBiD}}$ test for different datasets.

| Dataset | # Test Locations | Learning Rate | Optimization Epoch |
|---------|------------------|---------------|--------------------|
| blob    | 17               | 0.007         | 1000               |
| dna     | 15               | 0.0004        | 1000               |
| agnos   | 15               | 0.001         | 200                |
| topo21  | 15               | 0.001         | 1000               |
| har     | 15               | 0.001         | 100                |
| kropt   | 1                | 0.002         | 1000               |
| diamon  | 2                | 0.0013        | 1000               |
| cifar10 | 2                | 0.003         | 200                |
| mnist   | 2                | 0.001         | 500                |
| santan  | 10               | 0.001         | 1000               |
| codrna  | 2                | 0.01          | 1000               |
| sea50   | 1                | 0.01          | 1000               |
| adult   | 1                | 0.001         | 1000               |
| labor   | 1                | 0.001         | 1000               |
| poker   | 3                | 0.002         | 200                |
| higgs   | 15               | 0.001         | 1000               |

- Original cifar10 (samples $\mathbb{P}$) is compared to adversarial-cifar10 (samples $\mathbb{Q}$) following [19], where adversarial-cifar10 is constructed by [86] for distribution shift detection. To construct two samples with one identical distribution, we select randomly between original cifar10 and adversarial-cifar10 and drawn two samples from the same dataset. The image is scaled to $32 \times 32$ and we set the sample size for training to 200 and for testing to 50.

- mnist contains 70000 handwritten digit images, we compare true mnist data (samples $\mathbb{P}$) to Fake-mnist data (samples $\mathbb{Q}$) following [39], where the Fake-mnist data is drawn from a pre-trained deep convolutional generative adversarial network [87] and the image is scaled to $32 \times 32$. To construct two samples with one identical distribution, we select randomly between original mnist and Fake-mnist and drawn two samples from the same dataset. We set the sample size for training to 600 and for testing to 150.

- santan (santandercustomersatisfaction) is a categorical dataset with 2 classes. For constructing two different samples, we set the sample fraction for one sample to $[0.8, 0.2]$ and for the other sample to $[0.25, 0.75]$; To construct two samples with one identical distribution, we set the same sample fraction $[0.5, 0.5]$ for two samples. We set the sample size for training to 1000 and for testing to 250.

- codrna is a categorical dataset with 2 classes. For constructing two different samples, we set the sample fraction for one sample to $[0.7, 0.3]$ and for the other sample to $[0.35, 0.65]$; To construct two samples with one identical distribution, we set same sample fraction $[0.5, 0.5]$ for two samples. We set the sample size for training to 1000 and for testing to 250.

- sea50 is a categorical dataset with 2 classes. For constructing two different samples, we set the sample fraction for one sample to $[0.7, 0.3]$ and for the other sample to $[0.3, 0.7]$; To construct two samples with one identical distribution, we set same sample fraction $[0.5, 0.5]$ for two samples. We set the sample size for training to 5000 and for testing to 1250.

- adult is a categorical dataset with 2 classes. For constructing two different samples, we set the sample fraction for one sample to $[0.6, 0.4]$ and for the other sample to $[0.4, 0.6]$; To construct two samples with one identical distribution, we set same sample fraction $[0.5, 0.5]$ for two samples. We set the sample size for training to 5000 and for testing to 1250.

- labor is a categorical dataset with 2 classes. For constructing two different samples, we set the sample fraction for one sample to $[0.6, 0.4]$ and for the other sample to $[0.45, 0.55]$; To construct two samples with one identical distribution, we set same sample fraction $[0.5, 0.5]$ for two samples. We set the sample size for training to 5000 and for testing to 1250.

- poker is a categorical dataset with 2 classes. For constructing two different samples, we set the sample fraction for one sample to $[0.2, 0.8]$ and for the other sample to $[0.8, 0.2]$; To construct two samples with one identical distribution, we set same sample fraction $[0.5, 0.5]$ for two samples. We set the sample size for training to 3000 and for testing to 750.

---

**Algorithm 2** ME$_{\text{MaBiD}}$ training

---

**Input**: Two training samples $X$ and $Y$, number of test locations $\ell$, optimization epoch $K$, step size $\eta$
**Output**: $\{v_i\}_{i=1}^{\ell}$, $\{M_i\}_{i=1}^{\ell}$, $\{\gamma_i\}_{i=1}^{\ell}$ and $\boldsymbol{F}$

1: Initialize $\{v_i\}_{i=1}^{\ell}$, $\{M_i\}_{i=1}^{\ell} = \boldsymbol{I}_d$ and $\{\gamma_i\}_{i=1}^{\ell}$ by Eqn. (36)
2: **for** $k = 1, 2, \ldots, K$ **do**
3:     Calculate the embeddding training samples $\hat{X}$ and $\hat{Y}$ based on Eqn (2)
4:     Calculate the statistic $\mathcal{T}(\hat{X}, \hat{Y})$ based on Eqn. (4)
5:     Calculate gradient $\nabla_{\boldsymbol{v}_j}\mathcal{T}(\hat{X}, \hat{Y}), \nabla_{\gamma_j}\mathcal{T}(\hat{X}, \hat{Y})$ and $\nabla_{M_j}\mathcal{T}(\hat{X}, \hat{Y})$ via Eqns. (32), (34), (35)
6:     Gradient ascend based on Adam optimization method with step size $\eta$
7: **end for**
8: Calculate the embeddding training samples $\hat{X}$ and $\hat{Y}$ based on Eqn (2)
9: Calculate the pooled covariance matrix $\Sigma_{\hat{X}, \hat{Y}}$ based on Eqn. (3)
10: Calculate the Schur decomposition of the pooled covariance matrix: $\boldsymbol{LL} = \Sigma_{\hat{X}, \hat{Y}}$.
11: Calculate the inference direction $\boldsymbol{F}$ based on Eqn. (6)
12: **return** $\{v_i\}_{i=1}^{\ell}$, $\{M_i\}_{i=1}^{\ell}$, $\{\gamma_i\}_{i=1}^{\ell}$ and $\boldsymbol{F}$

---

- higgs is a categorical dataset with 2 classes. For constructing two different samples, we set the sample fraction for one sample to $[0.0, 1.0]$ and for the other sample to $[1.0, 0.0]$; To construct two samples with one identical distribution, we set same sample fraction $[1.0, 0.0]$ for two samples. We set the sample size for training to 16000 and for testing to 4000.

In optimization, we adapt Adam optimization method from the pytorch library in python [88, 89]. Table 4 presents the details of the hyperparameter settings for each dataset, including the number of test locations, learning rate.

**Experimental settings**

At initialization, we usually set the Mahalanobis matrices $\{M_i\}_{i=1}^{\ell}$ to identity matrices. We further provide an alternative initialization for the Mahalanobis matrices based on the correlation information of two training samples as follows:

$$M_j = \sum_{i=1}^{m+n}(\boldsymbol{S}_{j,i} - \boldsymbol{c}_{\boldsymbol{S}_j})(\boldsymbol{S}_{j,i} - \boldsymbol{c}_{\boldsymbol{S}_j})^{\top}/(m+n-1) + \delta\mathbf{I}_d\,,$$

where $\boldsymbol{S}_j = \{\boldsymbol{x}_1 - \boldsymbol{v}_j, ..., \boldsymbol{x}_m - \boldsymbol{v}_j, \boldsymbol{y}_1 - \boldsymbol{v}_j, ..., \boldsymbol{y}_n - \boldsymbol{v}_j\}$, $\boldsymbol{S}_{j,i}$ is the $i$-th element of $\boldsymbol{S}_j$, and $\boldsymbol{c}_{\boldsymbol{S}_j} = \sum_{i=1}^{m+n}\boldsymbol{S}_{j,i}/(m+n)$. $\mathbf{I}_d$ is an identity matrix of size $d \times d$ to guarantee the positive definiteness with small constant $\delta > 0$. Here, the elements $\boldsymbol{x}$ and $\boldsymbol{y}$ are come from training samples $X$ and $Y$, and $m$ and $n$ are the numbers of elements in $X$ and $Y$ respectively.

For initialization of test locations, we first fit a Gaussian distribution for each sample and draw half of the number of test locations from each distribution. This could be expensive for high dimensional dataset, and we can simplify the process by directly sampling test locations from original dataset.

For initialization of the bandwidth parameters of Mahalanobis kernels, we let $\gamma_j = \gamma$ for every $j \in [\ell]$ and then linearly search for $\gamma$ that maximize the statistic from a candidate list with fixed test locations and Mahalanobis matrices, which can be formalized as follows:

$$\gamma_* = \text{argmax}_{\gamma}\mathcal{T}(\hat{X}, \hat{Y}) \quad \text{for} \quad \gamma \in \{med^2 \cdot 2^{-4}, med^2 \cdot 2^{-3.8}, med^2 \cdot 2^{-3.6}, \ldots, med^2 \cdot 2^4\}\,, \quad (36)$$

where $med$ denotes the median of pairwise Euclidean distances of points in $X$ and $Y$.

**Compared methods**

We now present the details for our compared methods as follows:

- **ME**[2]: The mean embeddding method learns a set of test locations and a single Gaussian kernel, and then measures the difference between two mean embedddings with a statistic following $\chi^2$ distribution [9, 10];

---

[2]The code is downloaded from *github.com/wittawatj/interpretable-test*.

---
**Algorithm 3** $\text{ME}_{\text{MaBiD}}$ testing

---

**Input**: Two testing samples $X'$ and $Y'$, test locations $\{\boldsymbol{v}_i\}_{i=1}^{\ell}$, Mahalanobis kernels parameters $\{M_i\}_{i=1}^{\ell}$ and $\{\gamma_i\}_{i=1}^{\ell}$, inference direction $\boldsymbol{F}$, significance level $\alpha$ and parameter $\beta$

**Output**: $h$

1: Calculate the embeddding testing samples $\hat{X}'$ and $\hat{Y}'$ based on Eqn. (2)
2: Calculate mean embeddings $\boldsymbol{c}_{\hat{X}'} = \sum_{i=1}^{m'} \hat{\boldsymbol{x}}_i'/m'$ and $\boldsymbol{c}_{\hat{Y}'} = \sum_{j=1}^{n} \hat{\boldsymbol{y}}_i'/n'$
3: Calculate the pooled covariance matrix $\Sigma_{\hat{X}',\hat{Y}'}$ based on Eqn. (3)
4: Calculate the Schur decomposition of the pooled covariance matrix: $\boldsymbol{L}'\boldsymbol{L}' = \Sigma_{\hat{X}',\hat{Y}'}$.
5: **if** $\boldsymbol{F}^T(\boldsymbol{c}_{\hat{X}'} - \boldsymbol{c}_{\hat{Y}'}) \geq 0$ **then**
6:     $h = \mathbb{I}[\chi_\ell^2(\mathcal{T}(\hat{X}',\hat{Y}')) \leq \beta\alpha]$
7: **else**
8:     $h = \mathbb{I}[\chi_\ell^2(\mathcal{T}(\hat{X}',\hat{Y}')) \leq (2-\beta)\alpha]$
9: **end if**
10: **return** $h$

---

---
**Algorithm 4** Exploring the local significant differences

---

**Input**: Two testing samples $X, Y$, Mahalanobis kernels' parameters $\{M_i\}_{i=1}^{\ell}$ and $\{\gamma_i\}_{i=1}^{\ell}$, test locations $\{\boldsymbol{v}_i\}_{i=1}^{\ell}$, partition tree $T$

**Output**: $t^*$

1: Calculate the embeddding testing samples $\hat{X}'$ and $\hat{Y}'$ based on Eqn. (2)
2: Get the local two samples $\hat{X}'_{\mathcal{B}_i}$ and $\hat{Y}'_{\mathcal{B}_i}$ for every $i \in [s]$ based on partition Tree $T$
3: Calculate bi-directional masked $p$-value $g(\hat{X}'_{\mathcal{B}_i}, \hat{Y}'_{\mathcal{B}_i})$ for each rectangle region $\mathcal{B}_i$
4: Resort rectangle regions as $\mathcal{B}_{\langle 1\rangle}, \mathcal{B}_{\langle 2\rangle}, \ldots, \mathcal{B}_{\langle s\rangle}$ in a non-increasing order based on $g(\cdot,\cdot)$
5: Calculate bi-directional hypothesis $h(\hat{X}'_{\mathcal{B}_i}, \hat{Y}'_{\mathcal{B}_i})$ for each rectangle region $\mathcal{B}_i$ by Eqn. (7)
6: Calculate the index set $t^*$ based on Eqn. (11)
7: **return** $t^*$

---

- **C2ST-S**[3]: A binary classification neural network is trained and the statistic is computed as the accuracy over a hold-out set of two samples [13];

- **MMDAgg**[4]: A solution for the fundamental kernel selection problem involves the aggregation of a large number of kernels with several bandwidths, where the incomplete $U$-statistics are used to measure the difference between two samples Schrab et al. [75]. Notice that we set the testing sample size for MMDAgg to 5 times that of the other methods, since it does not require training;

- **MMD-D**[5]: A deep kernel approach for Maximum Mean Discrepancy (MMD), where the parameters of a neural network, two lengthscales of Gaussian kernels and a regularization parameter are optimized [39];

- **C2ST-L**[6]: A binary classification neural network is trained and the statistic is computed as the difference between outputs of the logit function corresponding to two samples [14, 18];

- **AutoML**[7]: A binary classifier is trained based on Automated Machine Learning techniques and the statistic is same to C2ST-L approach [19].

We implement methods for exploring local significant differences in Python, following their respective guidelines as follows:

- **FDG**: Partition the sample space based on probability binning and then compare the cardinalities of two samples over rectangle regions, where a normalized chi-squared value is computed for each bin to measure the local difference [24];

---

[3]The code is downloaded from *github.com/lopezpaz/classifier_tests*.

[4]The code is downloaded from *github.com/antoninschrab/agginc-paper*.

[5]The code is downloaded from *github.com/fengliu90/DK-for-TST*.

[6]The code is downloaded from *github.com/xycheng/net_logit_test*.

[7]The code is downloaded from *github.com/jmkuebler/autoML-TST-paper*.

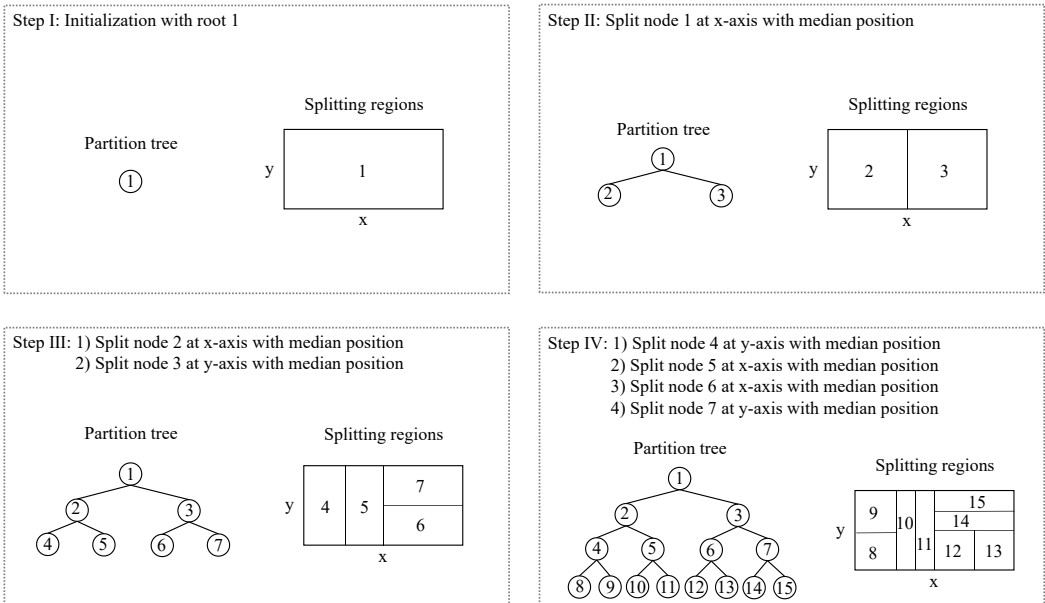

**Figure 9:** An illustration for our partition tree with splitting regions.

- **K-PRIM**: Partition the space based on patient rule induction method along with the information of kernel density estimation, and use a statistic following $\chi_1^2$ distribution to identify local distributional difference [28];

- **MRS**: Partition the sample space based on pólya tree method, which splits a leaf node based on the median of randomly selected feature, and then measure the difference between local two samples based on Binomial distribution [26];

- **TEAM**: Partition the sample space on the median of the feature with largest sample variance, and then measure differences between local two samples with Binomial distribution [27];

- **BTLDD**: Estimate the conditional probabilities of two samples based on a regression model and then cluster those elements with significant different conditional probabilities [77];

- **MMDT**: Partition the sample space into multiple equal grids based on quantile values of the features, and then estimate the kernel densities of two samples and measure local differences based on Welch's two-sample t-test statistic [29].

For the exploration of local significant differences, we take density differences [78] between two samples in a local region as an evaluation measure for local significant differences, and follow the works of [79, 80] based on $k$-NN density estimator with $k = 20$. Here, denote by $X$ and $Y$ the available data for density evaluation, we have estimated density functions as follows:

$$f_X(\boldsymbol{z}) := \frac{20}{N \cdot v_d \cdot r_X(\boldsymbol{z})^d} \quad \text{and} \quad f_Y(\boldsymbol{z}) := \frac{20}{N \cdot v_d \cdot r_Y(\boldsymbol{z})^d} \ ,$$

where $\boldsymbol{z} \in [0, 1]^d$, $N$ denotes sample size for density estimation and $v_d$ is the volume of a unit ball in $\mathbb{R}^d$. Denote by $r_X(\boldsymbol{z})^d$ and $r_Y(\boldsymbol{z})^d$ the distances from $\boldsymbol{z}$ to its 20-th nearest neighbors in $X$ and $Y$, respectively. For density estimation, we use all data of diamond, and $1,000,000$ data of other datasets due to the limitation of time complexity.

For our ME$_{\text{MaBiD}}$ test, we present the detailed training procedure in Algorithm 2 and testing procedure in Algorithm 3. Figure 9 is a pictorial illustration to present the region-splitting method and clarify that our partition tree is constructed iteratively: We initiate tree root with embedding space, and during each iteration, we select randomly one of those leaves with the largest size of training data points, and select the feature of the largest statistics value and with the median splitting position. We present the detailed description on the exploration of local significant differences in Algorithm 4.

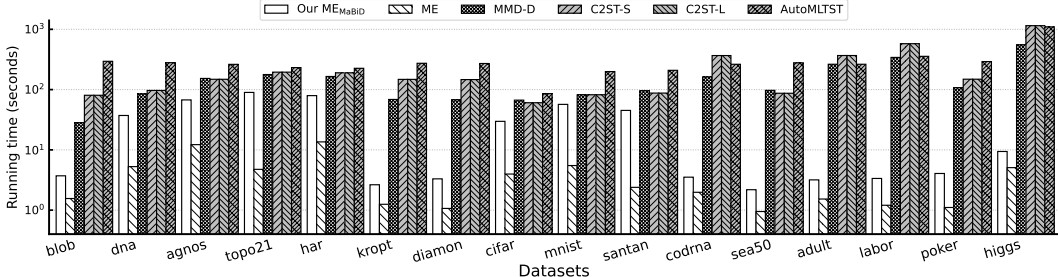

**Figure 10:** Comparisons of training time for different methods on two-sample test. Note that y-axis is in log-scale.

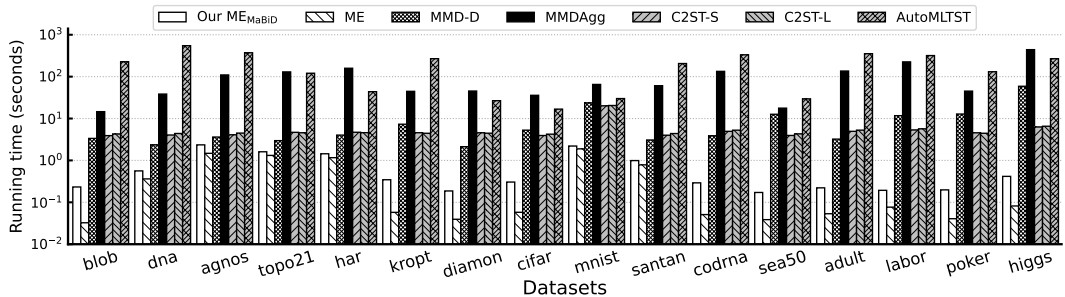

**Figure 11:** Comparisons of testing time for different methods on two-sample test. Note that y-axis is in log-scale.

## D Experimental comparisons

**Running time**

For fair comparisons, we run all experiments on a single core without parallel optimizations, and experiments are performed with Python on nodes of a computational cluster with a single CPU (Intel Core i9-10900X 3.7GHz) and a single GPU (GeForce RTX 2080 Ti), running Ubuntu with 128GB main memory. For these methods based on deep neural networks, such as C2ST-S, C2ST-L and MMD-D, we run experiments on GPU. For other methods except for AutoMLTST, we run experiments on CPU. For AutoMLTST, we run experiments on GPU for cifar and mnist, and run experiments on CPU for other datasets.

We further compare the average training time and testing time (in seconds) for different methods on two-sample test, as shown in Figure 10 and Figure 11. Notice that, the training time of MMDAgg is 0, since it has no training procedure. In testing procedure, our $ME_{MaBiD}$ and ME methods calculate the rejection threshold based on the asymptotic distribution $\chi^2_\ell$, whereas other methods perform permutation test or wild bootstrap.

**Additional experimental comparisons**

Table 5 presents the comparisons of type-I error for different methods. As can be seen, type-I error is limited about $\alpha = 0.05$ for all compared methods, which shows the effectiveness of test power in experiments for different methods.

Figure 12 shows the experimental comparisons between Mahalanobis kernels and Gaussian kernels, and it is obvious that Mahalanobis kernels achieves higher test power with better performance by exploiting local regions and directional information. Figure 13 presents experimental comparisons of test power with different training sample sizes. The testing sample size is set as 200 for blob, 1000 for higgs, 200 for codrna and 150 for sea50. As can be seen, our $ME_{MaBiD}$ test takes better test powers empirically than other methods w.r.t different training sample sizes by incorporating correlation and directional information of two samples.

**Table 5:** Comparisons of type-I error (mean±std).

| Dataset | Ours ME$_{MaBiD}$ | ME | MMDAgg | MMD-D | C2ST-L | C2ST-S | AutoMLTST |
|---------|-------------------|-----|--------|-------|--------|--------|-----------|
| blob | .043± .025 | .064± .056 | .061± .054 | .064± .018 | .037± .032 | .004± .006 | .019± .016 |
| dna | .045± .036 | .052± .068 | .061± .035 | .002± .004 | .037± .025 | .059± .054 | .033± .034 |
| agnostic | .047± .023 | .049± .034 | .051± .043 | .002± .004 | .042± .029 | .063± .045 | .046± .039 |
| topo21 | .033± .019 | .049± .032 | .038± .034 | .049± .039 | .019± .022 | .051± .024 | .038± .024 |
| har | .046± .033 | .033± .026 | .040± .030 | .045± .028 | .022± .014 | .061± .041 | .052± .041 |
| kropt | .033± .016 | .021± .006 | .013± .000 | .030± .029 | .046± .016 | .033± .012 | .033± .021 |
| diamon | .042± .024 | .029± .024 | .017± .012 | .046± .023 | .013± .010 | .021± .016 | .033± .031 |
| cifar | .015± .010 | .012± .011 | .025± .014 | .078± .068 | .021± .016 | .027± .017 | .020± .014 |
| mnist | .043± .038 | .013± .016 | .041± .028 | .092± .068 | .032± .041 | .057± .026 | .049± .039 |
| santan | .037± .031 | .092± .043 | .029± .006 | .058± .019 | .025± .018 | .037± .027 | .042± .036 |
| codrna | .013± .010 | .017± .016 | .075± .080 | .042± .028 | .046± .047 | .042± .012 | .008± .012 |
| sea50 | .032± .027 | .021± .012 | .058± .026 | .088± .059 | .029± .041 | .058± .056 | .050± .047 |
| adult | .024± .016 | .021± .016 | .029± .016 | .038± .010 | .033± .016 | .033± .026 | .062± .020 |
| labor | .017± .016 | .037± .010 | .054± .059 | .056± .029 | .029± .024 | .029± .026 | .042± .026 |
| poker | .029± .016 | .013± .010 | .054± .041 | .078± .047 | .029± .006 | .058± .031 | .050± .054 |
| higgs | .046± .011 | .024± .016 | .043± .031 | .066± .028 | .068± .098 | .048± .017 | .034± .039 |

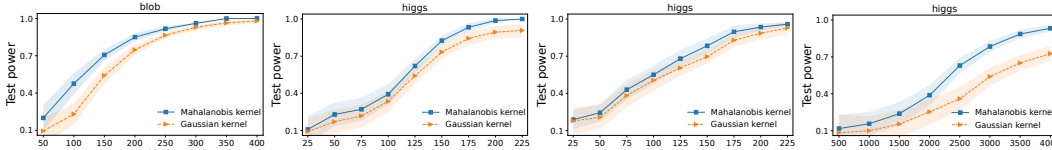

**Figure 12:** The comparisons of test power vs sample size for Mahalanobis kernel and Gaussian kernel.

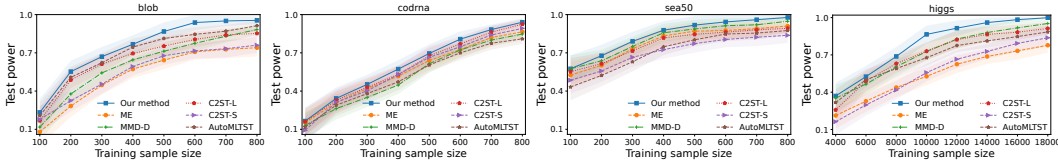

**Figure 13:** The comparisons of test power with different training sample sizes.

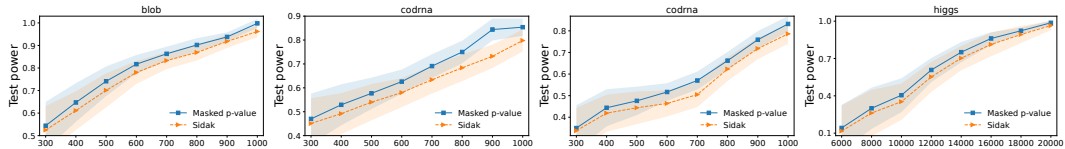

**Figure 14:** The comparisons of test power vs sample size for our bi-directional masked $p$-value and prior Sidak.

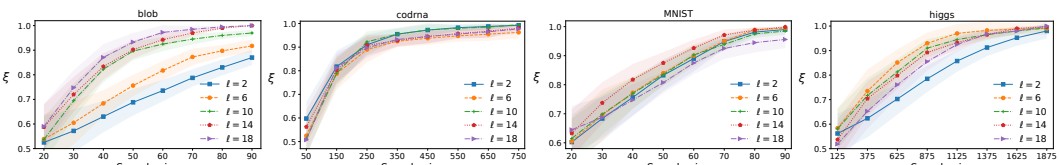

**Figure 15:** The comparisons of $\xi$ with different sample size and number of test locations.

Figure 14 presents experimental comparisons between our bi-directional masked $p$-value and previous Sidak [67]. As can be seen, our bi-directional masked $p$-value achieves higher test power by exploiting the significant level of local difference. Figure 15 analyzes the relationship between probability $\xi = \Pr[\text{sgn}(\boldsymbol{F}_{\mathcal{B}}^{\top} \boldsymbol{L}_{\mathcal{B}}'^{-1} (\boldsymbol{c}_{\hat{X}_{\mathcal{B}}'} - \boldsymbol{c}_{\hat{Y}_{\mathcal{B}}'})] \geq 1)$ and sample size, by considering different numbers of test locations $\ell \in \{2, 6, 10, 14, 18\}$. As can be seen, $\xi$ gradually increases as the sample size for different number of test locations, which shows the necessity of directional information in two-sample test.

Figure 16 analyzes the relationship between the number of local regions $s$ and probability $\xi = \Pr[\text{sgn}(\boldsymbol{F}_{\mathcal{B}}^{\top} \boldsymbol{L}_{\mathcal{B}}'^{-1} (\boldsymbol{c}_{\hat{X}_{\mathcal{B}}'} - \boldsymbol{c}_{\hat{Y}_{\mathcal{B}}'})] \geq 1)$. The probability $\xi$ increases with $s$, and hence can effectively partition the space into local regions with different inference directions. Figure 17 presents

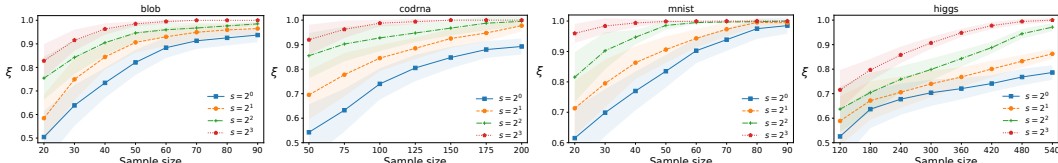

**Figure 16:** The comparisons of $\xi$ for partition trees with $s \in [2^0, 2^3]$. The value of $\xi$ increases as $s$ increases.

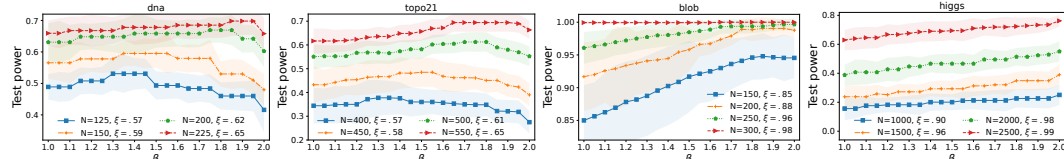

**Figure 17:** The correlation between the optimal parameter $\beta$ and probability $\xi = \Pr[\text{sgn}(\boldsymbol{F}_{\mathcal{B}}^\top \boldsymbol{L}_{\mathcal{B}}'^{-1} ( \boldsymbol{c}_{\hat{X}_{\mathcal{B}}'} - \boldsymbol{c}_{\hat{Y}_{\mathcal{B}}'} )] \geq 1)$. Here, $N$ denotes sample size.

additional experiments on the relationship between the optimal parameter $\beta$ and the probability $\xi = \Pr[\boldsymbol{F}^\top \boldsymbol{L}'^{-1}(\boldsymbol{c}_{\hat{X}'} - \boldsymbol{c}_{\hat{Y}'}) \geq 0]$, as shown in Figure 6.

