# OpenReview forum: "On the Exploration of Local Significant Differences For Two-Sample Test"
_NeurIPS.cc/2023/Conference — NeurIPS 2023 poster_

### Official Review · Reviewer_wXPw · 2023-06-21

**Soundness:** 3 good
**Presentation:** 3 good
**Contribution:** 3 good
**Rating:** 5
**Confidence:** 3

**Summary:**

This paper proposes a bidirectional two-sample test that can address both positive and negative gaps from one sample to the other. The test is proposed based on Hotelling's statistic, and the direction information is encoded via the mean difference. By searching test locations and an adequate Mahalanobis kernel, and more importantly, by detecting the local inference, the test exhibits a higher testing power than the baseline.

**Strengths:**

**Originality**\
The proposed test is novel.

**quality**\
The authors spent great effort delivering theorems and performing thorough experiments to validate the test.

**clarity**\
The writing and the structure of the paper are adequately clear, though it can be improved in some parts.

**significance**\
The proposed test has extensive applications in experimental design and observation studies, especially for the case where the global difference is small but the local difference is large.


**Weaknesses:**

**Tradeoff between local difference and overall sample complexity**\
The testing power typically depends on the problem's difficulty and the sample complexity. An easier problem and higher sample complexity lead to a higher testing power for a two-sample test. One important idea of the proposed test is to shift the testing focus to the local region instead of global, and hence potentially change the original testing problem to an easier one. The price to pay is the sample complexity used to compute the statistic for a region is certainly smaller than the sample complexity used for a global test. The paper seems not to address the tradeoff between the decreasing problem's difficulty and the decreasing sample complexity used to compute the statistic.

**Clarity**\
The authors could use pictorial illustrations for presenting the region-splitting method; right now it is not super clear to me how the method work. In addition, the paper could include a problem formulation section to present what's the null and alternative hypothesis here; the introduction of the local test sort of blurs the typical setting for the two-sample test.


**Questions:**

Do you split the dataset into a training set and a test set, optimize the kernel parameters and test locations and split the region base on the training set, and then perform the two-sample test based on the test set? If so, how does the size of the training set impact the testing power, and do you have experimental results for different sizes of training sets?

**Limitations:**

The following papers propose two-sample tests by detecting local differences inexplicitly.

1. Duan, Boyan, Aaditya Ramdas, and Larry Wasserman. "Interactive rank testing by betting." Conference on Causal Learning and Reasoning. PMLR. \
2. Li, Weizhi, et al. "A label efficient two-sample test." Uncertainty in Artificial Intelligence. PMLR, 2022.

---

> ### Author Rebuttal · Authors · 2023-08-08
>
> [Q1] Tradeoff between local difference and overall sample complexity … seems not to address the tradeoff between the decreasing problem's difficulty and the decreasing sample complexity used to compute the statistic.
>
> [A1] We will clarify that we take the number of Mahalanobis kernels and number of leaves of partition tree to balance problem's difficulty and sample complexity, as in (Soriano and Ma, 2017; Pura et al., 2023). For a difficult problem, we require more Mahalanobis kernels and more leaves of partition tree, which increases overall sample complexity. In experiments, we take cross validation for different datasets to select proper parameters on such tradeoff.
>
> &nbsp;
>
> [Q2] The authors could use pictorial illustrations for presenting the region-splitting method; right now it is not super clear to me how the method work.
>
> [A2] We will add a pictorial illustration to present the region-splitting method (Figure 1 in the uploaded PDF file), and clarify that our partition tree is constructed iteratively: We initiate tree root with embedding space, and during each iteration, we select randomly one of those leaves with the largest size of training data points, and select the feature of the largest statistics value and with the median splitting position. Algorithms 3 and 4 present the detailed descriptions for the region-splitting method in Appendix E.
>
> &nbsp;
>
> [Q3] Do you split the dataset into a training set and a test set …how does the size of the training set impact the testing power, and do you have experimental results for different sizes of training sets?
>
> [A3] We will clarify that we split every dataset into a training set (80%) and a test set (20%) as done in (Kübler et al., NeurIPS2020), and the details are given in Appendix B. We will add a figure (Figure 2 in the uploaded PDF file) to show the experimental results for different sizes on four datasets, and the trends are similar for other datasets. As can be seen, our method takes better test powers empirically.
>
> &nbsp;
>
> [Q4] The following papers propose two-sample tests by detecting local differences inexplicitly … 1. Duan, Boyan, Aaditya Ramdas, and Larry Wasserman … 2.Li, Weizhi, et al. …
>
> [A4] We will add two relevant papers (Duan et al. 2022; Li et al. 2022) by detecting local differences inexplicitly, and clarify that Duan et al. (2022) introduced the interactive rank test by betting, while Li et al. (2022) considered a different setting where sample features are easily measured yet sample labels are unknown and costly to obtain.
>
> &nbsp;
>
> We will add a problem formulation section to introduce local test, null and alternative hypothesis, etc. We will improve this work according to your suggestions, Thank you.

---

> > ### Comment · Reviewer_wXPw · 2023-08-14
> >
> > Thank you for the clarification. I will maintain my score as the tradeoff between local difference and overall sample complexity has not been analytically provided; it seems to me that extended efforts are needed to obtain quantitative results.

---

> > > ### Author Response · Authors · 2023-08-16
> > >
> > > Dear Reviewer wXPw,
> > >
> > > We will clarify that cross validation is used to select different parameters for Mahalanobis kernels and partition tree to balance between local difference and overall sample complexity empirically, and experiments show the superiority of our method.
> > >
> > > To the best of our knowledge, it still remains open to theoretically analyze the sample complexity on exploration of local significant difference (Roederer and Hardy, 2001; Cazáis and Lhéritier, 2015; Soriano and Ma, 2017; Dworkin and Linn, 2021; Pura et al., 2023). We tried to give theoretical analysis on this issue during the past three days but failed, and the main challenge lies in technical analysis of sample complexity for learning partition tree and multiple Mahalanobis kernels. We leave it as an open problem for further work.

---

### Official Review · Reviewer_CMa2 · 2023-07-04

**Soundness:** 2 fair
**Presentation:** 2 fair
**Contribution:** 3 good
**Rating:** 2
**Confidence:** 3

**Summary:**

The paper introduces a two-sample test that explores local variations in distribution. The authors provide a motivation for this specific formulation by highlighting two practical scenarios in cosmology and biology. They proceed with a literature review and conclude that none of the existing approaches is fully satisfactory. Guided by the problem's requirements and their intuition they incorporate space partitions, Mahalanobis kernels, and bi-directional hypothesis testing in developing the new test. The proposed algorithm is experimentally compared with existing methods, and it is found to empirically outperform them.


**Strengths:**

The paper is quite novel in both its motivation and the algorithm's construction. It is commendable that authors justify further specialization of two-sample tests ( "it is necessary to take one more step to find and understand local significant differences, rather than only two-sample test" ), by invoking two compelling practical problems.

I am impressed by the unorthodox design of this test, which incorporates methods from various sub-fields to create an algorithm that addresses the requirements and and seems to perform well in practice.

**Weaknesses:**

It is unclear which elements of the design (bi-directional p-values, tree partition, kernel design) contribute to overall performance of the algorithm.

Clarity of the writeup goes down starting from the "Construction of partition tree" section. A few examples:

* It is not explained why authors need masked p-value.
* Why is it important to guarantee equal probabilities in partitioned rectangle regions?
* Theorem 5 makes assumptions that I don't understand ".. if p-values are mutually independent for regions without difference, and are independent to regions with difference". Does your test control FWER?

Finally,  Bi-directional hypothesis for testing looks like designing non-centered rejection regions for the L^{-1}(c_{\hat x} - c_{\hat y}) (you don't have to that L2 norm of the embedding)


edit: I have read the rebuttal, and now I have less confidence that the clarity of the paper will improve. I am also more worried that the lack of clarity hides technical flaws. Moving  to 2.

**Questions:**

I suggest authors improve the quality of the writeup.

**Limitations:**

yes

---

> ### Author Rebuttal · Authors · 2023-08-08
>
> [Q1] It is unclear which elements of the design (bi-directional p-values, tree partition, kernel design) contribute to overall performance of the algorithm.
>
> [A1] We will clarify that the bi-directional p-values is proposed to improve test power by exploiting directional information between mean embeddings of two samples, as shown in Figure 4, and it is more adaptable to difference datasets than previous one-directional p-value (McIntosh, Amer. Stat. 2022) and non-directional p-value (Scetbon & Varoquaux, NeurIPS2019; Wynne & Duncan, JMLR2022), by selecting proper parameters.
>
> We will also clarify that we take partition tree to explore local significant difference following (Soriano and Ma, 2017; Pura et al., 2023), but with a new splitting criterion relevant to test power and data correlation; therefore, our method takes better performance on the exploration of local significant difference empirically, as shown in Table 4.
>
> We finally clarify that kernel has been a standard technique for two-sample test such as Gaussian Kernel (Gretton et al., NeurIPS2012; Jitkrittum et al., NeurIPS2016) and deep kernel (Liu et al., ICML2020), and our work takes multiple Mahanalobis kernels to exploit flexibly local differences from different neighborhoods and feature maps.
>
> Hence, the bi-directional p-values and kernel are helpful to improve the performance of two-sample test, and tree partition and kernel are helpful to improve the performance of exploration of local significant difference.
>
> &nbsp;
>
> [Q2] Clarity of the writeup … why authors need masked p-value. Why is it important to guarantee equal probabilities in partitioned rectangle regions?
>
> [A2] We will clarify that masked p-value is used to select the region of the largest local difference, and it is helpful to improve the test power of local hypothesis test. We will also clarify that equal probabilities could yield partition regions with balanced data points, and such partition takes better performance than regular grids, as shown empirically in (Boracchi et al., 2017; Liu et al., 2021).
>
> &nbsp;
>
> [Q3] Theorem 5 makes assumptions that I don't understand ". if p-values are mutually independent for regions without difference, and are independent to regions with difference". Does your test control FWER?
>
> [A3] We will clarify that the assumptions (Theorem 5) follow previous studies (Lei and Fithian, 2018; Duan et al., ICML2020), that is, 1) the p-values of local regions without differences are mutually independent; and 2) the p-values of local regions with differences are independent to those p-values of local regions without differences. We will also clarify that our test could upper bound the FWER with a given significant level under those assumptions, as shown in Figure 6 empirically.
>
> &nbsp;
>
> [Q4] Bi-directional hypothesis for testing looks like designing non-centered rejection regions for the L^{-1}(c_{\hat x} - c_{\hat y}) (you don't have to that L2 norm of the embedding)
>
> [A4] We will clarify that Bi-directional hypothesis essentially designs non-centered rejection regions in direction L^{-1}(c_{\hat x} - c_{\hat y}), and the L2 norm gives an alternative expression for optimization statistics T(\hat{X},\hat{Y}), and maximizing the L2 norm is essential to maximize the test power for two-sample test w.r.t. fixed sizes of two samples.
>
> &nbsp;
>
> We will improve the clarity of this work according to your suggestions. Thank you.

---

> ### Comment · Area_Chair_shj7 · 2023-08-17
>
> This is another friendly reminder from the AC that *you need to respond to the rebuttal*. The authors spent quite a lot of time preparing the rebuttal. As basic academic etiquette, you should also spend some time on it when you are available.

---

> > ### Comment · Reviewer_CMa2 · 2023-08-17
> >
> > I have read the rebuttal, and now I have less confidence that the clarity of the paper will improve. I am also more worried that the lack of clarity hides technical flaws.
> >
> > Let's consider the example of a "Bi-directional hypothesis." The authors state that they will explain how the Bi-directional hypothesis essentially creates non-centered rejection regions. If you are using a well-established, straightforward, perhaps even textbook, statistical procedure, you must not introduce a new name for it and simply include it in your paper. Would you accept a paper that reintroduces  OLS as  SNIPS (Simple Network with Interoperable, Practical, Simple features)?

---

> > > ### Comment · Area_Chair_shj7 · 2023-08-18
> > >
> > > Reviewer CMa2, **your sarcastic tone is problematic and you must stop doing it**.
> > >
> > > Can you be more specific what the *well-established, straightforward, perhaps even textbook* name for "Bi-directional hypothesis" with a proper reference? It's just a name and we can require the authors to change it whenever necessary. My comment applies to not only Bi-directional hypothesis but also similar terms in your mind.
> > >
> > > AC

---

> > > ### Author Response · Authors · 2023-08-18
> > >
> > > Dear Reviewer CMa2,
> > >
> > > We will clarify that this work studies on the exploration of local significant differences, and it is quite natural and intuitive to exploit local information from different local regions and local directions.
> > >
> > > For local regions, we propose multiple Mahalanobis kernels to exploit intrinsic structures and correlations from different neighborhoods and feature maps, which is different from previous Gaussian kernel (Jitkrittum et al., NeurIPS2016) and deep kernel (Liu et al., ICML2020).
> > >
> > > For local directions, we introduce the bi-directional hypothesis from two most discriminative directions to improve the sensitivity in detecting actual differences, which is more adaptable than previous one-directional hypothesis (McIntosh, Amer. Stat. 2022) and non-directional hypothesis (Scetbon & Varoquaux, NeurIPS2019; Wynne & Duncan, JMLR2022), by selecting proper parameters. The bi-directional hypothesis is partially motivated from directional hypothesis (Kaiser, 1960; Shaffer, 1972; Follmann, 1996; McIntosh, 2022).
> > >
> > > To our knowledge, it is the first time to introduce multiple Mahalanobis kernels and binary-directional hypothesis for two-sample test and exploration of local significant differences, with some new theoretical results and better experimental performance. Thank you.
> > >
> > > Yours sincerely,
> > >
> > > The authors.
> > >
> > > &nbsp;
> > >
> > > References:
> > >
> > > [1] F. Kaiser. Directional statistical decisions. Psychological Review, 67(3):160, 1960.
> > >
> > > [2] P. Shaffer. Directional statistical hypotheses and comparisons among means. Psychological Bulletin, 77(3):195, 1972
> > >
> > > [3] D. Follmann. A simple multivariate test for one-sided alternatives. Journal of the American Statistical Association, 91(434):854–861, 1996.
> > >
> > > [4] M. J. McIntosh. Calculating sample size for Follmann’s simple multivariate test for one-sided alternatives. American Statistician, 76(1):16–21, 2022.

---

> > > > ### Comment · Area_Chair_shj7 · 2023-08-18
> > > >
> > > > Authors, please clarify the following key point: Did you create the terms "one-directional hypothesis" and "non-directional hypothesis", or did the authors of your cited papers do so? I don't care other details because they are not directly related to the ongoing debate.

---

> > > > > ### Author Response · Authors · 2023-08-20
> > > > >
> > > > > Dear AC,
> > > > >
> > > > > We will clarify that the non-directional hypothesis was introduced to test the sum of differences of multiple variables without specific direction (Kaiser 1960; Dyba et al., 2006; Tavakol & Sandars 2014), and previous hypotheses (Scetbon & Varoquaux, NeurIPS2019; Wynne & Duncan, JMLR2022) for two-sample have been a case of non-directional hypothesis.
> > > > >
> > > > > We will also clarify that one-directional hypothesis (also called one-sided hypothesis) was introduced to test difference of multiple variables with one specific direction (Anselin 1998; Lachin & Bebu 2015); for example, previous hypothesis (Follmann, 1996; McIntosh 2022) focuses on one-directional hypothesis with the direction (1,...,1)'.
> > > > >
> > > > > Our bi-directional hypothesis is partially motivated from non-directional/one-directional hypothesis, but considers two most discriminative directions to improve the sensitivity in detecting actual differences for two-sample test. To our knowledge, we do not find similar definition and analysis before.
> > > > >
> > > > > Yours sincerely,
> > > > >
> > > > > The authors

---

> > > > > > ### Comment · Area_Chair_shj7 · 2023-08-20
> > > > > >
> > > > > > You don't get the point. You should know the difference between a concept and a name. The same concept can have different names in different research fields/areas.
> > > > > > See what the reviewer commented on your clarity issue:
> > > > > > > If you are using a well-established, straightforward, perhaps even textbook, statistical procedure, you must not introduce a new name for it and simply include it in your paper.
> > > > > >
> > > > > > Then answer my question directly.
> > > > > > > Did you create the terms "one-directional hypothesis" and "non-directional hypothesis", or did the authors of your cited papers do so?

---

> > > > > > > ### Comment · Area_Chair_shj7 · 2023-08-20
> > > > > > >
> > > > > > > If you are simply following the *naming convention* of previous researchers in this research area, it is no problem at all because this is how you respect previous researchers in this research area. If you changed the names of existing concepts by yourself, you should consider not introducing new names and simply including the old names in your paper, as required by this reviewer. Please understand the key is whether there is any renaming or rebranding of *well-established, straightforward, perhaps even textbook* concepts in your paper or not.

---

> > > > > > > > ### Author Response · Authors · 2023-08-20
> > > > > > > >
> > > > > > > > Dear AC,
> > > > > > > >
> > > > > > > > We will clarify that non-directional hypothesis was original proposed by Kaiser (1960), followed by statistical studies with the same name (Dyba et al., 2006; Tavakol & Sandars, 2014). It is also referred to as two-sided hypothesis (Cohen & Sackrowitz, 1998; Zhang & Gou, 2021) and two-tailed hypothesis (Ibe, 2014).
> > > > > > > >
> > > > > > > > We will also clarify that one-directional hypothesis has been introduced in (Anselin, 1998; Lachin & Bebu 2015), and it is also referred to as one-sided hypothesis (Lachin, 2014) and one-tailed hypothesis (Ibe, 2014).
> > > > > > > >
> > > > > > > > This work introduces new bi-directional hypothesis, which is partially motivated from non-directional/one-directional hypothesis, but considers two most discriminative directions to improve the sensitivity for two-sample test.
> > > > > > > >
> > > > > > > > Yours sincerely,
> > > > > > > >
> > > > > > > > The authors

---

> > > > > > > > > ### Comment · Reviewer_CMa2 · 2023-08-21
> > > > > > > > >
> > > > > > > > > >your sarcastic tone is problematic and you must stop doing it.
> > > > > > > > >
> > > > > > > > > I'm sorry it was perceived this way, I'll tone down my future responses.
> > > > > > > > >
> > > > > > > > > I think the "bi-directional masked p-value" can be understood narrowly as given by equation 7. Understood in such a narrow way, the "bi-directional masked p-value" is a new concept, but in my opinion, it has limited usefulness outside of this work.
> > > > > > > > >
> > > > > > > > > However, I think authors make a much broader claim. The contributions section states "[we] introduce bi-directional hypothesis for testing. Intuitively [...] the bi-directional hypothesis is beneficial to improve the sensitivity of two-sample test with proper parameter adaptation".
> > > > > > > > >
> > > > > > > > > The authors agreed that the concept of a "bi-directional hypothesis for testing" is essentially about designing a rejection region. Rejection regions are discussed in Chapter 10 of "All of Statistics: A Concise Course in Statistical Inference," particularly in the first few paragraphs. It is evident that rejection regions can take various forms and do not need to adhere to any specific structure. In my opinion, the paper as a whole gives the impression that the notion of custom rejection regions is a novel idea in hypothesis testing and warrants a distinct name. I would have simply written, "We consider a custom rejection region. It works well in our algorithm because of xyz."
> > > > > > > > >
> > > > > > > > > For what it's worth, I've given the "bi-directional hypothesis for testing" as an example of lack of clarity. Another example could be the author's rebuttal to another comment: "It is unclear which elements of the design (bi-directional p-values, tree partition, kernel design) contribute to the overall performance of the algorithm." After reading the rebuttal, I still don't know if all four components are necessary, or how I should think about ablations.
> > > > > > > > >
> > > > > > > > > The paper reads as an intriguing melange of techniques that work together, without a clear insight into why they do so.

---

> > > > > > > > > > ### Author Response · Authors · 2023-08-21
> > > > > > > > > >
> > > > > > > > > > Dear Reviwer CMa2,
> > > > > > > > > >
> > > > > > > > > > Thanks for your comments and suggestions. We will clarify the details as follows:
> > > > > > > > > >
> > > > > > > > > > [Q1] I think the "bi-directional masked p-value" can be understood narrowly as given by equation 7. Understood in such a narrow way, the "bi-directional masked p-value" is a new concept, but in my opinion, it has limited usefulness outside of this work.
> > > > > > > > > >
> > > > > > > > > > [A1] We will clarify that bi-directional masked p-value is proposed to reflect the significant level of local difference and improve the power of multiple hypothesis testing, which is an extension of previous masked p-value (Lei et al., 2020; Duan, 2021) (by setting $\beta = 1$). Our bi-directional masked p-value can also be applied to other multiple hypothesis testing scenarios (Lei & Fithian, 2018; Cai et al., 2019; Duan et al., ICML2020; Du et al., 2020).
> > > > > > > > > >
> > > > > > > > > > &nbsp;
> > > > > > > > > >
> > > > > > > > > > [Q2] The authors agreed that the concept of a "bi-directional hypothesis for testing" is essentially about designing a rejection region ... In my opinion, the paper as a whole gives the impression that the notion of custom rejection regions is a novel idea in hypothesis testing and warrants a distinct name…
> > > > > > > > > >
> > > > > > > > > > [A2] We clarify that our bi-directional hypothesis is essentially about designing a rejection region, and focuses on two most discriminative directions to improve the sensitivity for two-sample test. Our bi-directional hypothesis is a generalization of previous non-directional hypothesis (Kaiser, 1960; Tavakol & Sandars, 2014; Zhang & Gou, 2021) by setting $\beta=1$, and a generalization of previous one-directional hypothesis (Anselin 1998; Lachin & Bebu 2015) by setting $\beta=1$ and $F=(1,...,1)'$. We will consider a distinct name related to custom rejection regions according to your suggestion.
> > > > > > > > > >
> > > > > > > > > > &nbsp;
> > > > > > > > > >
> > > > > > > > > > [Q3] … I've given the "bi-directional hypothesis for testing" as an example of lack of clarity. Another example could be the author's rebuttal to another comment: "It is unclear which elements of the design (bi-directional p-values, tree partition, kernel design) contribute to the overall performance of the algorithm."…
> > > > > > > > > >
> > > > > > > > > > [A3] We will clarify that two components are necessary for two-sample test and the other two components are necessary for exploration of local significant difference, partially motivated from previous methods or empirical studies. We will present the details as follows:
> > > > > > > > > >
> > > > > > > > > > a) Kernel is a standard technique for two-sample test (Gretton et al., 2012; Chwialkowski et al., 2015; Liu et al., 2020; Schrab et al., 2022), while we design different Mahalanobis kernels to exploit more intrinsic structures for local differences (as shown in Figure 1).
> > > > > > > > > >
> > > > > > > > > > b) Tree partition is a basic technique for exploration of local significant difference (Soriano and Ma, 2017; Awaya and Ma, 2022; Pura et al., 2023), while we take new splitting criterion relevant to test power and data correlation.
> > > > > > > > > >
> > > > > > > > > > c) Bi-directional hypothesis is introduced for two-sample test to improve the sensitivity in detecting actual differences, by considering two most discriminative directions, which is partially observed from empirical studies (as shown in Figure 4).
> > > > > > > > > >
> > > > > > > > > > d) Bi-directional masked p-value is proposed for the exploation of local siginificant difference, which reflects the significant level of local difference and improves the power of multiple hypothesis testing, by exploiting directional information from different local regions. This is partially motivated from previous masked p-value of multiple hypothesis testing (Lei & Fithian, 2018; Cai et al., 2019; Du et al., 2020; Duan, 2021).
> > > > > > > > > >
> > > > > > > > > > &nbsp;
> > > > > > > > > >
> > > > > > > > > > Yours sincerely
> > > > > > > > > >
> > > > > > > > > > The authors

---

### Official Review · Reviewer_42S6 · 2023-07-06

**Soundness:** 3 good
**Presentation:** 3 good
**Contribution:** 3 good
**Rating:** 7
**Confidence:** 5

**Summary:**

In recent years, the focus on two-sample testing has significantly increased due to its wide range of practical applications. This study advances this exploration by highlighting local significant differences in two-sample testing. The authors introduce the ME, a robust test for two-sample testing that leverages local information through multiple Mahalanobis kernels and proposes a bi-directional hypothesis for testing. To pinpoint local significant differences, the embedding space is initially partitioned into several rectangular regions using a novel splitting criterion tied to test power and data correlation. Following this, local significant differences are identified using the bi-directional masked-value in conjunction with the ME test. The authors provide theoretical guarantees for their ME test, including lower bounds on test power, and effectively control the familywise error rate when investigating local significant differences. The effectiveness of the proposed methods in two-sample testing and the examination of local significant differences are then validated through extensive experiments.

**Strengths:**

1. This paper focuses on an important problem. Two-sample testing techniques have an important role in detecting distribution changes. T

2. The proposed method focuses on the local difference between two distributions, which is a promising research direction based on existing two-sample testing techniques. Recent testing techniques focus on learning good representations of data via maxing test power (Liu et al., 2020), which is another way to look at different areas with different weights but more flexible (bringing some issues). This paper proposes an effective way to implement this kind of technique.

3. The experiment is solid, verifying that the proposed method is good and effective.

**Weaknesses:**

1. There should be more discussion in the introduction. To show the difference between the deep-kernel-based method and your proposed method. I can understand the merit of the proposed method, but more discussions are needed in this paper.

2. The motivation to use Mahalanobis kernels is still not clear. Can we have more choices? What is your selection standard for kernels?

3. Procedures of permutation test should be introduced in preliminary. it is better for readers to understand the procedures of two-sample testing.

**Questions:**

See the weakness above.

---

> ### Author Rebuttal · Authors · 2023-08-08
>
> [Q1] To show the difference between the deep-kernel-based method and your proposed method, I can understand the merit of the proposed method, but more discussions are needed in this paper.
>
> [A1] We will add more discussions that Liu et al. (ICML2020) presented a deep kernel to adapt for variations in distribution smoothness and shape, with solid theoretical guarantees, and it is especially applicable to high dimensions and complex data, while our work takes a different way to exploit local distribution variations based on multiple Mahalanobis kernels from different neighborhoods and feature maps.
>
> &nbsp;
>
> [Q2] The motivation to use Mahalanobis kernels is still not clear. Can we have more choices? What is your selection standard for kernels?
>
> [A2] We will clarify that multiple Mahanalobis kernels flexibly exploit local differences from different neighborhoods and feature maps, and its analytic property (Theorem 2) ensures that the difference between two distributions can be detected almost surely at a finite number of randomly chosen locations (Chwialkowski et al., NeurIPS2015). There are different choices for kernels, such as Gaussian kernel, Laplacian kernel, deep kernel, etc., while our selection standard includes effectiveness, computational costs, kernel analyticity property, etc.
>
> &nbsp;
>
> We will introduce the permutation test (Sutherland et al., ICLR2017) in preliminary for readers, and improve this work according to your suggestions. Thank you.

---

> > ### Comment · Reviewer_42S6 · 2023-08-16
> > **Thanks for the responses**
> >
> > My concerns are addressed well. Please revise the paper accordingly if it is get in.

---

### Official Review · Reviewer_DcFi · 2023-07-06

**Soundness:** 3 good
**Presentation:** 2 fair
**Contribution:** 3 good
**Rating:** 6
**Confidence:** 4

**Summary:**

This paper investigates the exploration of local significant differences in the context of two-sample tests. The authors propose a new test called ME_{MaBiD}, which utilizes multiple Mahalanobis kernels and introduces a bi-directional hypothesis to exploit local information. The embedding space is partitioned into rectangle regions based on a new splitting criterion, and local significant differences are identified using the bi-directional masked p-value and the ME_{MaBiD} test. The paper provides theoretical guarantees for the ME_{MaBiD} test, including lower bounds on test power and control of familywise error rate. Extensive experiments validate the effectiveness and efficiency of the proposed methods for two-sample testing and the exploration of local significant differences.

**Strengths:**

Novel Approach: The paper introduces a new test, ME_{MaBiD}, which offers a novel approach to exploring local significant differences in two-sample tests. The utilization of multiple Mahalanobis kernels and the introduction of a bi-directional hypothesis provide a unique perspective for analyzing local information.

Rigorous Theoretical Analysis: The paper provides theoretical guarantees for the MEMaBiD test, including the asymptotic distribution of the proposed statistic with Mahalanobis kernels, lower bounds on test power, and control of familywise error rate. This demonstrates a strong foundation for the proposed method and enhances its credibility.

Exploration of Local Significant Differences: The paper addresses the important aspect of identifying local significant differences, going beyond traditional two-sample testing. By partitioning the embedding space into rectangle regions and introducing the bi-directional masked p-value, the paper offers a comprehensive framework for discovering and understanding local variations.

Extensive Experimental Validation: The paper conducts extensive experiments to validate the effectiveness and efficiency of the proposed methods. The superior performance of the ME_{MaBiD} test on various datasets for two-sample testing and the exploration of local significant differences demonstrates the practical value and applicability of the proposed approach.

------------------------after rebuttal------------------------
I'm leaning towards improving my evaluation due to the authors' commitment to enhancing clarity, which directly addresses my concern. Their responses have clarified the unique contributions of their work, particularly in exploring local significant differences, while their explanations on theoretical foundations underscore the method's credibility.

**Weaknesses:**

In general, I think the weakness lies mainly in clarity. It fails to give a formal definition of the input and output of the problem and defined further what is counted as local or not. Without such a formal definition, it would take more time to go into the details of the paper. Another issue with clarity is that the paper does not provide enough reasoning on why the proposed techniques are working towards the goal of the paper. For example, in Lines 67-68, why do Mahalanobis kernels exploit more information than the Gaussian kernels? In Eq.7 why the two directions are necessary, but they are not used in the exploration of local significance differences? For Lemma 1, would the learned \v_j satisfy the condition of drawn i.i.d. from some distribution? Why the alternative hypotheses H1 and H2 are important and practical? These factors are important to accurately evaluate the contribution of the paper but are not sufficiently discussed. Without such discussion,  the paper's main contributions seem to replace the Gaussian kernel with the Mahalanobis kernel and one direction test by two-directions, which is novel but may not be significant enough considering the motivation.

**Questions:**

In addition to what has been asked in the Weaknesses part. the type-I error rate in Theorem 4, and M_1, \ldots, M_\ell in Eq 5 are not defined. For the Construction of partition tree step one, is the leaf node randomly selected, or just select the largest? The paper seems to go with both.

---

> ### Author Rebuttal · Authors · 2023-08-08
>
> [Q1] I think the weakness lies mainly in clarity. It fails to give a formal definition of the input and output of the problem and defined further what is counted as local or not.
>
> [A1] We will present a formal definition of the problem, and clarify that the input includes two i.i.d. samples from two distributions, and the output for two-sample test is a boolean value to show whether two samples are from an identical distribution; and the output for local explorations includes some local regions with significant differences. We will also clarify that local explorations imply some local regions (i.e., leaves of partition tree) with significant differences, as in (Soriano and Ma, 2017).
>
> &nbsp;
>
> [Q2] in Lines 67-68, why do Mahalanobis kernels exploit more information than the Gaussian kernels?
>
> [A2] We will clarify that multiple Mahanalobis kernels flexibly exploit local differences from different neighborhoods and feature maps, and adapt to variations of distributional smoothness and shape, whereas Gaussian kernels focuses on  an isotropic scale in all directions, as shown in Figure 1.
>
> &nbsp;
>
> [Q3] In Eq.7 why the two directions are necessary, but they are not used in the exploration of local significance differences?
>
> [A3] We will clarify that two directions could improve test power by incorporating directional information between mean embeddings of two samples, as shown in Figure 4, and it is more adaptable to different datasets than previous one-directional test (McIntosh, Amer. Stat. 2022) and p-value test without direction (Scetbon & Varoquaux, NeurIPS2019; Wynne & Duncan, JMLR2022), by selecting parameters between two directions. We will also clarify that two directions are used for bi-directional masked p-value and local hypothesis test in the exploration of local significance differences.
>
> &nbsp;
>
> [Q4] For Lemma 1, would the learned \v_j satisfy the condition of drawn i.i.d. from some distribution? Why the alternative hypotheses H1 and H2 are important and practical?
>
> [A4] We will clarify that the i.i.d. condition for v_j has been a classical assumption in two-sample test (Chwialkowski et al., NeurIPS2015; Sectbon et al., NeurIPS2019), and the alternative hypotheses H1 and H2 are helpful to test the differences between two samples from two directions, and improve the lower bound of test power theoretically (Theorem 4).
>
> &nbsp;
>
> [Q5] … the type-I error rate in Theorem 4, and M_1, \ldots, M_\ell in Eq 5 are not defined. For the Construction of partition tree step one, is the leaf node randomly selected, or just select the largest?
>
> [A5] We will clarify that type-I error rate denotes the probability of falsely rejecting the true null hypothesis, and M_1, \ldots, M_\ell are positive-definite Mahalanobis kernels defined by Eqn. (1). We also clarify that we select randomly one of the largest leaves in the construction of partition tree.
>
> &nbsp;
>
> We will improve the clarity of this work with some defintions and discussions according to your suggestions. Thank you.

---

> > ### Comment · Reviewer_DcFi · 2023-08-13
> > **Re**
> >
> > Thank you for your detailed and thoughtful responses to my review. I think your explanations provide valuable insights into the motivations, methods, and contributions of your work. Given your detailed responses, I'm reassessing the contribution rating for your work. It's evident that your clarifications have addressed some of my initial concerns regarding novelty and significance. I commend your rigorous theoretical analysis, particularly the provision of lower bounds on test power and the control of familywise error rate, which enhance the credibility of your proposed method. I think I could raise my score in the current phase.
> >
> > I am now leaning toward acceptance. However, to further improve the evaluation, I have a few remaining questions:
> >
> > Could you provide further insights into the reasons Mahalanobis kernels are deemed more effective than Gaussian kernels for exploiting local differences?
> >
> > In Eq.7, while the use of two directions improves test power, could you elaborate on why these two directions are not employed in the exploration of local significant differences?
> >
> > I appreciate the clarification on Lemma 1. Could you also provide a brief explanation of why the alternative hypotheses H1 and H2 are practically important for your proposed approach?
> >
> > For the construction of the partition tree's first step, you mentioned selecting a leaf node randomly or selecting the largest. Could you clarify whether both methods are used, and under what circumstances?
> >
> > I appreciate your commitment to enhancing the clarity of the paper, which is crucial for ensuring its accessibility and comprehensibility. Your proposed clarifications, including formal definitions and explanations of terms, will significantly contribute to a better understanding of the problem, method, and results presented in the paper. This effort to improve clarity will undoubtedly benefit readers if the paper can be accepted.

---

> > > ### Author Response · Authors · 2023-08-15
> > >
> > > Dear Reviewer DcFi,
> > >
> > > Thanks for your evaluable comments and suggestions, and we will improve the paper accordingly.
> > > In the following we focus on technical questions.
> > >
> > > [Q1] Could you provide further insights into the reasons Mahalanobis kernels are deemed more effective than Gaussian kernels for exploiting local differences?
> > >
> > > [A1] We will clarify that Mahalanobis kernels could exploit more intrinsic structures and correlations for local differences from different directions and regions by using different Mahalanobis matrices, whereas Gaussian kernels focus on Euclidean distance with the same scale in all directions, and Guassian kernels have essentially been a special case of Mahalanobis kernels by selecting an identity matrix.
> > >
> > > [Q2] In Eq.7, while the use of two directions improves test power, could you elaborate on why these two directions are not employed in the exploration of local significant differences?
> > >
> > > [A2] We will clarify that we exploit local two directions by Eq. 8 for each local region, and the use of local two directions are twofold: i) calculations of bi-directional masked p-value, and ii) local bi-directional hypothesis test for each region, which are helpful to find local differences from distributional shapes.
> > >
> > > [Q3] I appreciate the clarification on Lemma 1. Could you also provide a brief explanation of why the alternative hypotheses H1 and H2 are practically important for your proposed approach?
> > >
> > > [A3] We clarify that alternative hypotheses H1 and H2 essentially test the difference between two samples from two most discriminative directions, which could improve the sensitivity in detecting actual differences. This is different from previous methods without direction (Scetbon & Varoquaux, NeurIPS2019; Wynne & Duncan, JMLR2022), where the potential assumption is that all directions have equal contribution to two-sample test. It is also different from previous one-directional method (Follmann, 1996; McIntosh 2022), which considers only one direction (1,...,1)' for every local region and every dataset without difference.
> > >
> > > [Q4] For the construction of the partition tree's first step, you mentioned selecting a leaf node randomly or selecting the largest. Could you clarify whether both methods are used, and under what circumstances?
> > >
> > > [A4] We will clarify that we randomly select one leaf node from the leaves nodes that contain the most training examples, and note that we may have one selection when there is only one leaf node that contains the most training examples.

---

### Author Rebuttal · Authors · 2023-08-08

We want to express our gratitudes to reviewers for their insightful comments, and we will improve the paper accordingly.

We present the detailed responses for each review (below), and upload a PDF file with two figures, as suggested by a reviewer.

---

### Comment · Area_Chair_shj7 · 2023-08-11
**Rebuttals are visible now**

Hi reviewers,

Please take a look at the rebuttals when you have some time. Thanks.

AC

---

> ### Author Response · Authors · 2023-08-13
>
> Dear Reviewers,
>
> We want to express our gratitudes to your helpful comments and suggestions, which will be of great importance on the improvement of this work.
>
> We have made efforts to address questions you raised and improve accordingly. We would like to double check to make sure that we have addressed all your concerns, in particular for Reviewer CMa2, and would you please let me know if you have any additional questions. Thank you.
>
> Best wishes,
>
> Authors.

---

### Decision · Program_Chairs · 2023-09-21

**Decision:**

Accept (poster)

**Comment:**

The paper studied two-sample test and focused on local significant differences to increase the test power. The paper has strong motivation and contributions. After the rebuttal and author-reviewer discussions, there were three positive (5, 6, 7) and one negative (2) reviewers. The rating of the negative reviewer was originally 3 before the rebuttal and then 2 after the rebuttal, which looks like an outlier to me. The major concern from this reviewer was clarity, which I think might be an issue but would not be a major issue. All the four reviewers recognized its novelty and impact. Thus, we should accept the paper for publication.

The only negative reviewer was especially unhappy about the name bi-directional hypothesis, but it is just a natural extension of the existing names non-directional hypothesis and one-directional hypothesis following previous research papers in the same research area. The authors should carefully introduce the research background: when a concept has two or more names (for example, one-directional hypothesis, one-sided hypothesis, and one-tailed hypothesis), all related names must be covered; then, you should explain why you adopt the naming convention of a specific paper but not that of another paper.